# IMPROVED ALGORITHM AND BOUNDS FOR SUCCESSIVE PROJECTION

**Jiashun Jin & Gabriel Moryoussef**
Department of Statistics
Carnegie Mellon University
Pittsburgh, PA 15213, USA
{jiashun, gmoryous}@andrew.cmu.edu

**Zheng Tracy Ke & Jiajun Tang & Jingming Wang**
Department of Statistics
Harvard University
Cambridge, MA 02138, USA
{zke,jiajuntang,jingmingwang}@fas.harvard.edu

## ABSTRACT

Given a $K$-vertex simplex in a $d$-dimensional space, suppose we measure $n$ points on the simplex with noise (hence, some of the observed points fall outside the simplex). Vertex hunting is the problem of estimating the $K$ vertices of the simplex. A popular vertex hunting algorithm is successive projection algorithm (SPA). However, SPA is observed to perform unsatisfactorily under strong noise or outliers. We propose pseudo-point SPA (pp-SPA). It uses a projection step and a denoise step to generate pseudo-points and feed them into SPA for vertex hunting. We derive error bounds for pp-SPA, leveraging on extreme value theory of (possibly) high-dimensional random vectors. The results suggest that pp-SPA has faster rates and better numerical performances than SPA. Our analysis includes an improved non-asymptotic bound for the original SPA, which is of independent interest.

## 1 INTRODUCTION

Fix $d \geq 1$ and suppose we observe $n$ vectors $X_1, X_2, \ldots, X_n$ in $\mathbb{R}^d$, where

$$X_i = r_i + \epsilon_i, \qquad \epsilon_i \stackrel{iid}{\sim} N(0, \sigma^2 I_d). \tag{1}$$

The Gaussian assumption is for technical simplicity and can be relaxed. For an integer $1 \leq K \leq d+1$, we assume that there is a simplex with $K$ vertices $\mathcal{S}_0$ on the hyperplane $\mathcal{H}_0$ such that each $r_i$ falls within the simplex (note that a simplex with $K$ vertices always falls on a $(K-1)$-dimensional hyperplane of $\mathbb{R}^d$). In other words, let $v_1, v_2, \ldots, v_K \in \mathbb{R}^d$ be the vertices of the simplex and let $V = [v_1, v_2, \ldots, v_K]$. We assume that for each $1 \leq i \leq n$, there is a $K$-dimensional weight vector $\pi_i$ (a weight vector is vector where all entries are non-negative with a unit sum) such that

$$r_i = \sum_{k=1}^{K} \pi_i(k) v_k = V \pi_i. \tag{2}$$

Here, $\pi_i$'s are unknown but are of major interest, and to estimate $\pi_i$, the key is vertex hunting (i.e., estimating the $K$ vertices of the simplex $\mathcal{S}_0$). In fact, once the vertices are estimated, we can estimate $\pi_1, \pi_2, \ldots, \pi_n$ by the relationship of $X_i \approx r_i = V \pi_i$. Motivated by these, the primary interest of this paper is vertex hunting (VH). The problem may arise in many application areas. *(1) Hyperspectral unmixing*: Hyperspectral unmixing (Bioucas-Dias et al., 2012) is the problem of separating the pixel spectra from a hyperspectral image into a collection of constituent spectra. $X_i$ contains the spectral measurements of pixel $i$ at $d$ different channels, $v_1, \ldots, v_K$ are the constituent spectra (called *endmembers*), and $\pi_i$ contains the fractional *abundances* of endmembers at pixel $i$. It is of great interest to identify the endmembers and estimate the abundances. *(2) Archetypal analysis*. Archytypal analysis (Cutler & Breiman, 1994) is a useful tool for representation learning. Take its application in genetics for example (Satija et al., 2015). Each $X_i$ is the gene expression of cell $i$, and each $v_k$ is an archetypal expression pattern. Identifying these archetypal expression patterns is useful for inferring a transcriptome-wide map of spatial patterning. *(3) Network membership estimation*. Let $A \in \mathbb{R}^{n,n}$ be the adjacency matrix of an undirected network with $n$ nodes and $K$ communities. Let $(\hat{\lambda}_k, \hat{\xi}_k)$ be the $k$-th eigenpair of $A$, and write $\hat{\Xi} = [\hat{\xi}_1, \hat{\xi}_2, \ldots, \hat{\xi}_K]$. Under certain network

models (e.g., Huang et al. (2023); Airoldi et al. (2008); Zhang et al. (2020); Ke & Jin (2023); Rubin-Delanchy et al. (2022)), there is a $K$-vertex simplex in $\mathbb{R}^K$ such that for each $1 \leq i \leq n$, the $i$-th row of $\widehat{\Xi}$ falls (up to noise corruption) inside the simplex, and vertex hunting is an important step in community analysis. *(4) Topic modeling.* Let $D \in \mathbb{R}^{n,p}$ be the frequency of word counts of $n$ text documents, where $p$ is the dictionary size. If $D$ follows the Hoffman's model with $K$ topics, then there is also simplex in the spectral domain (Ke & Wang, 2022)), so vertex hunting is useful.

Existing vertex hunting approaches can be roughly divided into two lines: constrained optimizations and stepwise algorithms. In the first line, one proposes an objective function and estimates the vertices by solving an optimization problem. The minimum volume transform (MVT) (Craig, 1994), archetypal analysis (AA) (Cutler & Breiman, 1994; Javadi & Montanari, 2020), and N-FINDER (Winter, 1999) are approaches of this line. In the second line, one uses a stepwise algorithm which iteratively identifies one vertex of the simplex at a time. This includes the popular successive projection algorithm (SPA) (Araújo et al., 2001). SPA is a stepwise greedy algorithm. It does not require an objective function (how to select the objective function may be a bit subjective), is computationally efficient, and has a theoretical guarantee. This makes SPA especially interesting.

**Our contributions**. Our primary interest is to improve SPA. Despite many good properties aforementioned, SPA is a greedy algorithm, which is vulnerable to noise and outliers, and may be significantly inaccurate. Below, we list two reasons why SPA may underperform. First, typically in the literature (e.g., Araújo et al. (2001)), one apply the SPA directly to the $d$-dimensional data points $X_1, X_2, \ldots, X_n$, regardless of what $(K, d)$ are. However, since the true vertices $v_1, \ldots, v_K$ lie on a $(K-1)$-dimensional hyperplane, if we directly apply SPA to $X_1, X_2, \ldots, X_n$, the resultant hyperplane formed by the estimated simplex vertices is likely to deviate from the true hyperplane, due to noise corruption. This will cause inefficiency of SPA. Second, since the SPA is a greedy algorithm, it tends to be biased outward bound. When we apply SPA, it is frequently found that most of the estimated vertices fall outside of true simplex (and some of them are faraway from the true simplex).

For illustration, Figure 1 presents an example, where $X_1, X_2, \ldots, X_n$ are generated from Model (1) with $(n, K, d, \sigma) = (1000, 3, 2, 1)$, and $r_i$ are uniform samples over $T$ ($T$ is the triangle with vertices $(1, 1)$, $(2, 4)$, and $(5, 2)$). In this example, the true vertices (large black points) form a triangle (dashed black lines) on a 2-dimensional hyperplane. The green and cyan-colored triangles are estimated by SPA and pp-SPA (our main algorithm to be introduced; since $d$ is equal to $K-1$, the hyperplane projection is skipped), respectively. In this example, the estimated simplex by SPA is significantly biased outward bound, suggesting a large room for improvement. Such outward bound bias of SPA is related to the design of the algorithm and is frequently observed (Gillis, 2019).

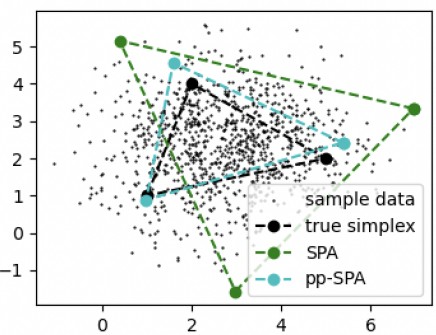

Figure 1: A numerical example ($d$=2, $K$=3).

To fix the issues, we propose pseudo-point SPA (pp-SPA) as a new approach to vertex hunting. It contains two novel ideas as follows. First, since the simplex $\mathcal{S}_0$ is on the hyperplane $\mathcal{H}_0$, we first use all data $X_1, \ldots, X_n$ to estimate the hyperplane, and then project all these points to the hyperplane. Second, since SPA is vulnerable to noise and outliers, a reasonable idea is to add a denoise step before we apply SPA. We propose a *pseudo-point (pp) approach* for denoising, where for each data point, we replace it by a pseudo point, computed as the average of *all of its neighbors within a radius of* $\Delta$. Utilizing information in the nearest neighborhood is a known idea in classification (Hastie et al., 2009), and the well-known $k$-nearest neighborhood (KNN) algorithm is such an approach. However, KNN or similar ideas were never used as a denoise step for vertex hunting. Compared with KNN, the idea of pseudo-point approach is motivated by the underlying geometry and is for a different purpose. For these reasons, the idea is new at least to some extent.

We have two theoretical contributions. First, Gillis & Vavasis (2013) derived a non-asymptotic error bound for SPA, but the bound is not always tight. Using a very different proof, we derive a sharper non-asymptotic bound for SPA. The improvement is substantial in the following case. Recall that $V = [v_1, v_2, \ldots, v_K]$ and let $s_k(V)$ be the $k$-th largest singular value of $V$. The bound in Gillis & Vavasis (2013) is proportional to $1/s_K^2(V)$, while our our bound is proportional to $1/s_{K-1}^2(V)$.

Since all vertices lie on a $(K-1)$-dimensional hyperplane, $s_{K-1}(V)$ is bounded away from 0, as long as the volume of true simplex is lower bounded. However, $s_K(V)$ may be 0 or nearly 0; in this case, the bound in Gillis & Vavasis (2013) is too conservative, but our bound is still valid. Second, we use our new non-asymptotic bound to derive the rate for pp-SPA, and show that the rate is much faster than the rate of SPA, especially when $d \gg K$. Even when $d = O(K)$, the bound we get for pp-SPA is still sharper than the bound of the original SPA. The main reason is that, for those points far away outside the true simplex, the corresponding pseudo-points we generate are much closer to the true simplex. This greatly reduces the *outward bound biases* of SPA (see Figure 1).

**Related literature**. It was observed that SPA is susceptible to outliers, motivating several variants of SPA (Gillis & Vavasis, 2015; Mizutani & Tanaka, 2018; Gillis, 2019). For example, Bhattacharyya & Kannan (2020); Bakshi et al. (2021); Nadisic et al. (2023) modified SPA by incorporating smoothing at each iteration. In contrast, our approach involves generating all pseudo points through neighborhood averaging before executing all successive projection steps. Additionally, we exploit the fact that the simplex resides in a low-dimensional hyperplane and apply a hyperplane projection step prior to the denoising and successive projection steps. Our theoretical results surpass those existing works for several reasons: (a) we propose a new variant of SPA; (b) our analyses build upon a better version of the non-asymptotic bound than the commonly-used one in Gillis & Vavasis (2013); and (c) we incorporate delicate random matrix and extreme value theory in our analysis.

## 2 A NEW VERTEX HUNTING ALGORITHM

The successive projection algorithm (SPA) (Araújo et al., 2001) is a popular vertex hunting method. This is an iterative algorithm that estimates one vertex at a time. At each iteration, it first projects all points to the orthogonal complement of those previously found vertices and then takes the point with the largest Euclidean norm as the next estimated vertex. See Algorithm 1 for a detailed description.

---

**Algorithm 1** The (orthodox) Successive Projection Algorithm (SPA)

---

**Input**: $X_1, X_2, \ldots, X_n$, and $K$.
Initialize $u = \mathbf{0}_p$ and $y_i = X_i$, for $1 \le i \le n$. For $k = 1, 2, \ldots, K$,

- Update $y_i$ to $(I_d - uu')y_i$. Obtain $i_k = \arg\max_{1 \le i \le n} \|y_i\|$. Update $u = \|y_{i_k}\|^{-1} y_{i_k}$.

**Output**: $\hat{v}_k = X_{i_k}$, for $1 \le k \le K$.

---

We propose pp-SPA as an improved version of the (orthodox) SPA, containing two main ideas: a *hyperplane projection* step and a *pseudo-point denoise* step. We now discuss the two steps separately.

Consider the *hyperplane projection* step first. In our model (2), the noiseless points $r_1, \ldots, r_n$ live in a $(K-1)$-dimensional hyperplane. However, with noise corruption, the observed data $X_1, \ldots, X_n$ are not exactly contained in a hyperplane. Our proposal is to first use data to find a 'best-fit' hyperplane and then project all data points to this hyperplane. Fix $d \ge K \ge 2$. Given a point $x_0 \in \mathbb{R}^d$ and a projection matrix $H \in \mathbb{R}^{d \times d}$ with rank $K-1$, the $(K-1)$-dimensional hyperplane associated with $(x_0, H)$ is $\mathcal{H} = \{x \in \mathbb{R}^d : (I_d - H)(x - x_0) = 0\}$. For any $x \in \mathbb{R}^d$, the Euclidean distance between $x$ and the hyperplane is equal to $\|(I_d - H)(x - x_0)\|$. Given $X_1, X_2, \ldots, X_n$, we aim to find a hyperplane to minimize the sum of square distances:

$$\min_{(x_0, H)} \{S(x_0, H)\}, \quad \text{where} \quad S(x_0, H) = \sum_{i=1}^{n} \|(I_d - H)(X_i - x_0)\|^2. \tag{3}$$

Let $Z = [Z_1, \ldots, Z_n]$, where $Z_i = X_i - \bar{X}$ and $\bar{X} = \frac{1}{n} \sum_{i=1}^{n} X_i$. For each $k$, let $u_k \in \mathbb{R}^d$ be the $k$th left singular vector of $Z$. Write $U = [u_1, \ldots, u_{K-1}]$. The next lemma is proved in the appendix.

**Lemma 1** $S(x_0, H)$ *is minimized by* $x_0 = \bar{X}$ *and* $H = UU'$.

For each $1 \le i \le n$, we first project each $X_i$ to $\tilde{X}_i$ and then transform $\tilde{X}_i$ to $Y_i$, where

$$\tilde{X}_i := \bar{X} + H(X_i - \bar{X}), \qquad Y_i := U'\tilde{X}_i; \qquad \text{note that } H = UU' \text{ and } Y_i \in \mathbb{R}^{K-1}. \tag{4}$$

These steps reduce noise. To see this, we note that the true simplex lives in a hyperplane with a projection matrix $H_0 = U_0 U_0'$. It can be shown that $U \approx U_0$ (up to a rotation) and $Y_i \approx r_i^* + U_0' \epsilon_i$,

with $r_i^* = U_0'\bar{X} + U_0'r_i$. These points $r_i^*$ still live in a simplex (in dimension $(K-1)$). Comparing this with the original model $X_i = r_i + \epsilon_i$, we see that $U_0'\epsilon_i$ are iid samples from $N(0, \sigma^2 I_{K-1})$, and $\epsilon_i$ are iid samples from $N(0, \sigma^2 I_d)$. Since $K - 1 \ll d$ in may applications, the projection may significantly reduce the dimension of the noise variable. Later in Section 4, we see that this implies a significant improvement in the convergence rate.

Next, consider the *neighborhood denoise* step. Fix an $\Delta > 0$ and an integer $N \geq 1$. Define the $\Delta$-neighborhood of $Y_i$ by $B_\Delta(Y_i) = \{x \in \mathbb{R}^{K-1} : \|x - Y_i\| \leq \Delta\}$. When there fewer than $N$ points in $B_\Delta(Y_i)$ (including $Y_i$ itself), remove $Y_i$ for the vertex hunting step next. Otherwise, replace $Y_i$ by the average of all points in $B_\Delta(Y_i)$ (denoted by $Y_i^*$). The main effect of the denoise effect is on the points that are *far outside the simplex*. For these points, we either delete them for the vertex hunting step (see below), or replace it by a point closer to the simplex. This way, we pull all these points "towards" the simplex, and thus reduce the estimation error in the subsequent vertex hunting step.

Finally, we apply the (orthodox) successive projection algorithm (SPA) to $Y_1^*, Y_2^*, \cdots, Y_n^*$ and let $\hat{v}_1, \hat{v}_2, \ldots, \hat{v}_K$ be the estimated vertices. Let $\hat{V} = [\hat{v}_1, \hat{v}_2, \ldots, \hat{v}_K]$. See Algorithm 2.

---

**Algorithm 2** Pseudo-Point Successive Projection Algorithm (pp-SPA)

---

**Input**: $X_1, X_2, \ldots, X_n \in \mathbb{R}^d$, the number of vertices $K$, and tuning parameters $(N, \Delta)$.

**Step 1** *(Projection)*. Obtain $\bar{X} = \frac{1}{n}\sum_{i=1}^n X_i$ and $Z = X - \bar{X}\mathbf{1}_n'$. Let $U = [u_1, \ldots, u_{K-1}]$ contain the first $(K-1)$ singular vectors of $Z$. For $1 \leq i \leq n$, let $Y_i = U'X_i \in \mathbb{R}^{K-1}$.

**Step 2** *(Denoise)*. Let $B_\Delta(Y_i) = \{x \in \mathbb{R}^{K-1} : \|x - Y_i\| \leq \Delta\}$ denote the $\Delta$-neighborhood of $Y_i$.

- If there are fewer than $N$ points (including $Y_i$ itself) in $B_\Delta(Y_i)$, delete this point.
- Otherwise, replace $Y_i$ by $Y_i^*$, which is the average of all points in $B_\Delta(Y_i)$.

**Step 3** *(VH)*. Let $\mathcal{J} \subset \{1, \ldots, n\}$ be the set of retained points in Step 2. Apply Algorithm 1 to $\{Y_i^*\}_{i \in \mathcal{J}}$ to get $\hat{v}_1^*, \hat{v}_2^*, \ldots, \hat{v}_K^* \in \mathbb{R}^{K-1}$. Let $\hat{v}_k = (I_d - H)\bar{X} + U\hat{v}_k^* \in \mathbb{R}^d$, $1 \leq k \leq K$.

**Output**: The estimated vertices $\hat{v}_1, \ldots, \hat{v}_K$.

---

**Remark 1**: The complexity of the orthodox SPA is $O(ndK)$. Regarding the complexity of pp-SPA, it applies SPA on $(K-1)$-dimensional pseudo-points, so the complexity is $O(nK^2)$. To obtain these pseudo points, we need a projection step and a denoise step. The projection step extracts the first $(K-1)$ singular vectors of a matrix $Z(n \times d)$. Performing the whole SVD decomposition would result in $O(\min(n^2 d, nd^2))$ time complexity. However, faster approach exists such as the truncated SVD which would decrease this complexity to $O(ndK)$. In the denoise step, we need to find the $\Delta$-neighborhoods for all $n$ points $Y_1, Y_2, \ldots, Y_n$. This can be made computationally efficient using the KD-Tree. The construction of KD-Tree takes $O(n \log n)$, and the search of neighbors typically takes $O\left(n^{(2 - \frac{1}{K-1})} + nm\right)$, where $m$ is the maximum number of points in a neighborhood.

**Remark 2**: Algorithm 2 has tuning parameters $(N, \Delta)$, where $\Delta$ is the radius of the neighborhood, and $N$ is used to prune out points far away from the simplex. For $N$, we typically take $N = \log(n)$ in theory and $N = 3$ in practice. Concerning $\Delta$, we use a heuristic choice $\Delta = \max_i \|Y_i - \bar{Y}\|/5$, where $\bar{Y} = \frac{1}{n}\sum_{i=1}^n Y_i$. It works satisfactorily in simulations.

**Remark 3** *(P-SPA and D-SPA)*: We can view pp-SPA as a generic algorithm, where we may either replace the projection step by a different dimension reduction step, or replace the denoise step by a different denoise idea, or both. In particular, it is interesting to consider two special cases: (i) *P-SPA*, which skips the denoise step and only uses the projection and VH steps; (ii) *D-SPA*, which skips the projection step and only uses the denoise and VH steps. We analyze these algorithms, together with pp-SPA (see Table 1 and Section C of the appendix). In this way, we can better understand the respective improvements of the projection step and the denoise step.

## 3 AN IMPROVED BOUND FOR SPA

Recall that $V = [v_1, v_2, \ldots, v_K]$, whose columns are the $K$ vertices of the true simplex $\mathcal{S}_0$. Let

$$\gamma(V) = \max_{1 \leq k \leq K}\{\|v_k\|\}, \qquad g(V) = 1 + 80\frac{\gamma^2(V)}{s_K^2(V)}, \qquad \beta(X) = \max_{1 \leq i \leq n}\{\|\epsilon_i\|\}. \tag{5}$$

**Lemma 2 (Gillis & Vavasis (2013), orthodox SPA)** *Consider $d$-dimensional vectors $X_1, \ldots, X_n$, where $X_i = r_i + \epsilon_i$, $1 \leq i \leq n$ and $r_i$ satisfy model (2). For each $1 \leq k \leq K$ there is an $i$ such that $\pi_i = e_k$. Suppose $\max_{1 \leq i \leq n} \|\epsilon_i\| \leq \frac{s_K(V)}{1 + 80\gamma^2(V)/s_K^2(V)} \min\{\frac{1}{2\sqrt{K-1}}, \frac{1}{4}\}$. Apply the orthodox SPA to $X_1, \ldots, X_n$ and let $\hat{v}_1, \hat{v}_2, \ldots, \hat{v}_K$ be the output. Up to a permutation of these $K$ vectors,*

$$\max_{1 \leq k \leq K}\{\|\hat{v}_k - v_k\|\} \leq \left[1 + 80\frac{\gamma^2(V)}{s_K^2(V)}\right] \max_{1 \leq i \leq n}\|\epsilon_i\| := g(V) \cdot \beta.$$

Lemma 2 is among the best known results for SPA, but this bound is still not satisfying. One issue is that $s_K(V)$ depends on the location (i.e., center) of $\mathcal{S}_0$, but how well we can do vertex hunting should not depend on its location. We expect that vertex hunting is difficult only if $\mathcal{S}_0$ has a small volume (so the simplex is nearly flat). To see how these insights connect to singular values of $V$, let $\bar{v} = K^{-1} \sum_{k=1}^K v_k$ be the center of $\mathcal{S}_0$, define $\tilde{V} = [v_1 - \bar{v}, \ldots, v_K - \bar{v}]$, and let $s_k(\tilde{V})$ be the $k$-th singular value of $\tilde{V}$. The next lemma is proved in the appendix:

**Lemma 3** $\text{Volume}(\mathcal{S}_0) = \frac{\sqrt{K}}{(K-1)!} \prod_{k=1}^{K-1} s_k(\tilde{V})$, $s_{K-1}(V) \geq s_{K-1}(\tilde{V})$, and $s_K(V) \leq \sqrt{K}\|\bar{v}\|$.

Lemma 3 yields several observations. First, as we shift the location of $\mathcal{S}_0$ so that its center gets close to the origin, $\|\bar{v}\| \approx 0$, and $s_K(V) \approx 0$. In this case, the bound in Lemma 2 becomes almost useless. Second, the volume of $\mathcal{S}_0$ is determined by the first $(K-1)$ singular values of $\tilde{V}$, irrelevant to the $K$th singular value. Finally, if the volume of $\mathcal{S}_0$ is lower bounded, then we immediately get a lower bound for $s_{K-1}(V)$. These observations motivate us to modify $g(V)$ in (5) to a new quantity that depends on $s_{K-1}(V)$ instead of $s_K(V)$; see (6) below.

Another issue of the bound in Lemma 2 is that $\beta(X)$ depends on the maximum of $\|\epsilon_i\|$, which is too conservative. Consider a toy example in Figure 2, where $\mathcal{S}_0$ is the dashed triangle, the red stars represent $r_i$'s and the black points are $X_i$'s. We observe that $X_2$ and $X_5$ are deeply in the interior of $\mathcal{S}_0$, and they should not affect the performance of SPA. We hope to modify $\beta(X)$ to a new quantity that does not depend on $\|\epsilon_2\|$ and $\|\epsilon_5\|$. One idea is to modify $\beta(X)$ to $\beta^*(X, V) = \max_i \text{Dist}(X_i, \mathcal{S}_0)$, where $\text{Dist}(\cdot, \mathcal{S}_0)$ is the Euclidean distance from a point to the simplex. For any point inside the simplex, this Euclidean distance is exactly zero. Hence, for this toy example, $\beta^*(X, V) \leq \max_{i \notin \{1,2,5\}}\|\epsilon_i\|$. However, we cannot simply replace $\beta(X)$ by $\beta^*(X, V)$, because $\|\epsilon_1\|$ also affects the performance of SPA and should not be left out. Note that $r_1$ is the only point located at the top vertex. When $X_1$ is far away from $r_1$, no matter whether $X_1$ is

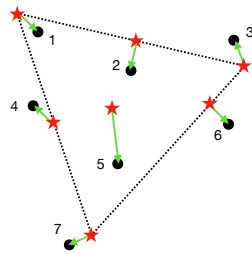

Figure 2: A toy example to show the difference between $\beta(X)$ and $\beta_{\text{new}}(X, V)$, where $\beta(X) = \max_i \|\epsilon_i\|$, and $\beta_{\text{new}}(X, V) \leq \max_{i \notin \{2,5\}} \|\epsilon_i\|$.

inside or outside $\mathcal{S}_0$, SPA still makes a large error in estimating this vertex. This inspires us to define $\beta^\dagger(X, V) = \max_k \min_{\{i:r_i=v_k\}} \|\epsilon_i\|$. When $\beta^\dagger(X, V)$ is small, it means for each $v_k$, there exists at least one $X_i$ that is close enough to $v_k$. To this end, let $\beta_{\text{new}}(X, V) = \max\{\beta^*(X, V), \beta^\dagger(X, V)\}$. Under this definition, $\beta_{\text{new}}(X) \leq \max_{i \notin \{2,5\}}\|\epsilon_i\|$, which is exactly as hoped.

Inspired by the above discussions, we introduce (for a point $x \in \mathbb{R}^d$, $\text{Dist}(x, \mathcal{S}_0)$ is the Euclidean distance from $x$ to $\mathcal{S}_0$; this distance is zero if $x \in \mathcal{S}_0$)

$$\begin{aligned} g_{\text{new}}(V) &= 1 + \frac{30\gamma(V)}{s_{K-1}(V)} \max\left\{1, \frac{\gamma(V)}{s_{K-1}(V)}\right\}, \\ \beta_{\text{new}}(X) &= \max\left\{\max_{1 \leq i \leq n} \text{Dist}(X_i, \mathcal{S}_0), \max_{1 \leq k \leq K} \min_{\{i:r_i=v_k\}} \|X_i - v_k\|\right\}. \end{aligned} \tag{6}$$

**Theorem 1** *Consider $d$-dimensional vectors $X_1, \ldots, X_n$, where $X_i = r_i + \epsilon_i$, $1 \leq i \leq n$ and $r_i$ satisfy model (2). For each $1 \leq k \leq K$ there is an $i$ such that $\pi_i = e_k$. Suppose for a properly small universal constant $c^* > 0$, $\max\{1, \frac{\gamma(V)}{\sigma_{K-1}(V)}\}\beta_{\text{new}}(X, V) \leq c^* \frac{s_{K-1}^2(V)}{\gamma(V)}$. Apply the orthodox SPA to $X_1, \ldots, X_n$ and let $\hat{v}_1, \hat{v}_2, \ldots, \hat{v}_K$ be the output. Up to a permutation of these $K$ vectors,*

$$\max_{1 \leq k \leq K}\{\|\hat{v}_k - v_k\|\} \leq g_{\text{new}}(V)\beta_{\text{new}}(X, V).$$

Note that $g_{\text{new}}(V) \leq g(V)$ and $\beta_{\text{new}}(X, V) \leq \beta(X)$. The non-asymptotic bound in Theorem 1 is always better than the bound in Lemma 2. We use an example to illustrate that the improvement can be substantial. Let $K = d = 3$, $v_1 = (20, 20, 10)$, $v_2 = (20, 30, 10)$, and $v_3 = (30, 22, 10)$. We put $r_1, r_2, r_3$ at each of the three vertices, $r_4, r_5, r_6$ at the mid-point of each edge, and $r_7$ at the center of the simplex (which is $\bar{v}$). We sample $\epsilon_1^*, \epsilon_2^*, \ldots, \epsilon_7^*$ i.i.d., from the unit sphere in $\mathbb{R}^3$. Let $\epsilon_i = 0.01\epsilon_i^*$, for $1 \leq i \leq 6$, and $\epsilon_7 = 0.05\epsilon_7^*$. By straightforward calculations, $g(V) = 4.3025 \times 10^4$, $g_{\text{new}}(V) = 6.577 \times 10^2$, $\beta(X) = 0.05$, $\beta_{new}(X, V) = 0.03$. Therefore, the bound in Lemma 2 gives $\max_k \|\hat{v}_k - v_k\| \leq 2151.3$, while the improved bound in Theorem 1 gives $\max_k \|\hat{v}_k - v_k\| \leq 18.7$. A more complicated version of this example can be found in Section D of the supplementary material.

The main reason we can achieve such a significant improvement is that our proof idea is completely different from the one in Gillis & Vavasis (2013). The proof in Gillis & Vavasis (2013) is driven by *matrix norm inequalities* and does not use any geometry. This is why they need to rely on quantities such as $s_K(V)$ and $\max_i \|\epsilon_i\|$ to control the norms of various matrices in their analysis. It is very difficult to modify their proof to obtain Theorem 1, as the quantities in (6) are insufficient to provide strong matrix norm inequalities. In contrast, our proof is guided by *geometric insights*. We construct a *simplicial neighborhood* near each true vertex and show that the estimate $\hat{v}_k$ in each step of SPA must fall into one of these simplicial neighborhoods.

## 4 THE BOUND FOR PP-SPA AND ITS IMPROVEMENT OVER SPA

We focus on the orthodox SPA in Section 3. In this section, we show that we can further improve the bound significantly if we use pp-SPA for vertex hunting. Recall that we have also introduced P-SPA and D-SPA in Section 2 as simplified versions of pp-SPA. We establish error bounds for P-SPA, D-SPA, and pp-SPA, under the Gaussian noise assumption in (1). A high-level summary is in Table 1. Recall that P-SPA, D-SPA, and pp-SPA all create pseudo-points and then feed them into SPA. Different ways of creating pseudo-points only affect the term $\beta_{\text{new}}(X, V)$ in the bound in Theorem 1. Assuming that $g_{\text{new}}(V) \geq C$, the order of $\beta_{\text{new}}(X, V)$ fully captures the error bound. Table 1 lists the sharp orders of $\beta_{\text{new}}(X, V)$ (including the constant).

Table 1: The sharp orders of $\beta_{\text{new}}(X, V)$ (settings: $K \geq 3$, $d$ satisfies (7), $s_{K-1}(V) > C$, and $m$ satisfies the condition in Theorem 3). P-SPA and D-SPA use *the projection only* and *the denoise only*, respectively. The constant $c_0 \in (0, 1)$ comes from $m$, and the constant $a_1 > 2$ is as in Lemma 5.

|  | $d \ll \log(n)$ | $d = a_0 \log(n)$ | $\log(n) \ll d \ll n^{1-\frac{2(1-c_0)}{K-1}}$ | $d \gg n^{1-\frac{2(1-c_0)}{K-1}}$ |
|---|---|---|---|---|
| SPA | $\sqrt{2\log(n)}$ | $\sqrt{a_1 \log(n)}$ | $\sqrt{d}$ | $\sqrt{d}$ |
| P-SPA | $\sqrt{2\log(n)}$ | $\sqrt{2\log(n)}$ | $\sqrt{2\log(n)}$ | $\sqrt{2\log(n)}$ |
| D-SPA | $\sqrt{2c_0 \log(n)}$ | NA | NA | NA |
| pp-SPA | $\sqrt{2c_0 \log(n)}$ | $\sqrt{2c_0 \log(n)}$ | $\sqrt{2c_0 \log(n)}$ | $\sqrt{2\log(n)}$ |

The results suggest that pp-SPA always has a strictly better error bound than SPA. When $d \gg \log(n)$, the improvement is a factor of $o(1)$; the larger $d$, the more improvement. When $d = O(\log(n))$, the improvement is a constant factor that is strictly smaller than 1. In addition, by comparing P-SPA and D-SPA with SPA, we have some interesting observations:

- *The projection effect.* From the first two rows of Table 1, the error bound of P-SPA is never worse than that of SPA. In many cases, P-SPA leads to a significant improvement. When $d \gg \log(n)$, the rate is faster by a factor of $\sqrt{\log(n)/d}$ (which is a huge improvement for high-dimensional data). When $d \asymp \log(n)$, there is still a constant factor of improvement.

- *The denoise effect.* We compare the error bounds for P-SPA and pp-SPA, where the difference is caused by the denoise step. In three out of the four cases of $d$ in Table 1, pp-SPA strictly improves P-SPA by a constant factor $c_0 < 1$.

  We note that pp-SPA applies denoise to the projected data in $\mathbb{R}^{K-1}$. We may also apply denoise to the original data in $\mathbb{R}^d$, which gives D-SPA. By Table 1, when $d \ll \sqrt{\log(n)}$, D-SPA improves SPA by a constant factor. However, for $d \gg \log(n)$, we always recommend applying denoise to the projected data. In such cases, the leading term in the extreme value of chi-square (see Lemma 5) is $d$, so the denoise is not effective if applied to original data.

Table 1 and the above discussions are for general settings. In a slightly more restrictive setting (see Theorem 2 below), both projection and denoise can improve the error bounds by a factor of $o(1)$.

We now present the rigorous statements. Owing to space constraint, we only state the error bounds of pp-SPA in the main text. The error bounds of P-SPA and D-SPA can be found in the appendix.

## 4.1 Some useful preliminary results

Recall that $V = [v_1, \ldots, v_K]$ and $r_i = V\pi_i$, $1 \le i \le n$. Let $\bar{v}$, $\bar{r}$, and $\bar{\pi}$ be the empirical means of $v_k$'s, $r_i$'s, and $\pi_i$'s, respectively. Introduce $\tilde{V} = [v_1 - \bar{v}, \ldots, v_K - \bar{v}]$, $R = n^{-1/2}[r_1 - \bar{r}, \ldots, r_n - \bar{r}]$, and $G = (1/n)\sum_{i=1}^{n}(\pi_i - \bar{\pi})(\pi_i - \bar{\pi})'$. Lemma 4 relates singular values of $R$ to those of $G$ and $V$ and is proved in the appendix ($A \preceq B$: $B - A$ is positive semi-definite. Also, $\lambda_k(G)$ is the $k$-th largest (absolute value) eigenvalue of $G$, $s_k(V)$ is the $k$-th largest singular value of $V$; same below).

**Lemma 4** *The following statements are true: (a) $RR' = VGV'$, (b) $\lambda_{K-1}(G) \cdot \tilde{V}\tilde{V}' \preceq VGV' \preceq \lambda_1(G) \cdot \tilde{V}\tilde{V}'$, and (c) $\lambda_{K-1}(G) \cdot s_{K-1}^2(\tilde{V}) \preceq \sigma_{K-1}^2(R) \preceq \lambda_1(G) \cdot s_{K-1}^2(\tilde{V})$.*

To analyze SPA and pp-SPA, we need precise results on the extreme values of chi-square variables. Lemma 5 is proved in the appendix.

**Lemma 5** *Let $M_n$ be the maximum of $n$ iid samples from $\chi_d^2(0)$. As $n \to \infty$, (a) if $d \ll \log(n)$, then $M_n/(2\log(n)) \to 1$, (b) if $d \gg \log(n)$, then $M_n/d \to 1$, and (c) if $d = a_0 \log(n)$ for a constant $a_0 > 0$, then $M_n/(a_1 \log(n)) \to 1$ where $a_1 > 2$ is unique solution of the equation $a_1 - a_0 \log(a_1) = 2 + a_0 - a_0 \log(a_0)$ (convergence in three cases are convergence in probability).*

## 4.2 Regularity conditions and main theorems

We assume
$$K = o(\log(n)/\log\log(n)), \qquad d = o(\sqrt{n}). \tag{7}$$
These are mild conditions. In fact, in practice, the dimension of the true simplex is usually relatively low, so the first condition is mild. Also, when the (low-dimensional) true simplex is embedded in a high dimensional space, it is not preferable to directly apply vertex hunting. Instead, one would use tools such as PCA to significantly reduce the dimension first and then perform vertex hunting. For this reason, the second condition is also mild. Moreover, recall that $G = n^{-1}\sum_{i=1}^{n}(\pi_i - \bar{\pi})(\pi_i - \bar{\pi})'$ is the empirical covariance matrix of the (weight vector) $\pi_i$ and $\gamma(V) = \max_{1 \le k \le K}\{\|v_k\|\}$. We assume for some constant $C > 0$,
$$\lambda_{K-1}(G) \ge C^{-1}, \qquad \lambda_1(G) \le C, \qquad \gamma(V) \le C. \tag{8}$$
The first two items are a mild balance condition on $\pi_i$ and the last one is a natural condition on $V$. Finally, in order for the (orthodox) SPA to perform well, we need
$$\sigma\sqrt{\log(n)}/s_{K-1}(\tilde{V}) \to 0. \tag{9}$$
In many applications, vertex hunting is used as a module in the main algorithm, and the data points fed into VH are from previous steps of some algorithm and satisfy $\sigma = o(1)$ (for example, see Jin et al. (2023); Ke & Wang (2022)). Hence, this condition is reasonable.

We present the main theorems (which are used to obtain Table 1). In what follows, Theorem 3 is for a general setting, and Theorem 2 concerns a slightly more restrictive setting. For each setting, we will specify explicitly the theoretically optimal choices of thresholds $(t_n, \epsilon_n)$ in pp-SPA.

For $1 \le k \le K$, let $J_k = \{i : r_i = v_k\}$ be the set of $r_i$ located at vertex $v_k$, and let $n_k = |J_k|$, for $1 \le k \le K$. Let $\Gamma(\cdot)$ denote the standard Gamma function. Define
$$m = \min\{n_1, n_2, \ldots, n_K\}, \qquad c_2 = 0.5(2e^2)^{-\frac{1}{K-1}}\sqrt{2/(K-1)}\big[\Gamma(\frac{K+1}{2})\big]^{\frac{1}{K-1}}. \tag{10}$$

Note that as $K \to \infty$, $c_2 \to 0.5/\sqrt{e}$. We also introduce
$$\alpha_n = \frac{\sqrt{d}}{\sqrt{n}s_{K-1}^2(\tilde{V})}\big(1 + \sigma\sqrt{\max\{d, 2\log(n)\}}\big), \qquad b_n = \frac{2\sigma}{\sqrt{n}}\sqrt{\max\{d, 2\log(n)\}}. \tag{11}$$

The following theorem is proved in the appendix.

**Theorem 2** *Suppose $X_1, X_2, \ldots, X_n$ are generated from model (1)-(2) where $m \geq c_1 n$ for a constant $c_1 > 0$ and conditions (7)-(9) hold. Fix $\delta_n$ such that $(K-1)/\log(n) \ll \delta_n \ll 1$, and let $t_n = \sqrt{K-1} \big( \frac{\log(n)}{n^{1-\delta_n}} \big)^{\frac{1}{K-1}}$. We apply pp-SPA to $X_1, X_2, \ldots, X_n$ with $(N, \Delta)$ to be determined below. Let $\hat{V} = [\hat{v}_1, \hat{v}_2, \ldots, \hat{v}_K]$, where $\hat{v}_1, \hat{v}_2, \ldots, \hat{v}_K$ are the estimated vertices.*

- *In the first case, $\alpha_n \ll t_n$. We take $N = \log(n)$ and $\Delta = c_3 t_n \sigma$ in pp-SPA, for a constant $c_3 \leq c_2$. Up to a permutation of $\hat{v}_1, \ldots, \hat{v}_K$, $\max_{1 \leq k \leq K}\{\|\hat{v}_k - v_k\|\} \leq \sigma g_{\text{new}}(V)[\sqrt{\delta_n} \cdot \sqrt{2\log(n)} + C\alpha_n] + b_n$.*

- *In the second case, $t_n \ll \alpha_n \ll 1$. We take $N = \log(n)$ and $\Delta = \sigma\alpha_n$ in pp-SPA. Up to a permutation of $\hat{v}_1, \ldots, \hat{v}_K$, $\max_{1 \leq k \leq K}\{\|\hat{v}_k - v_k\|\} \leq \sigma g_{\text{new}}(V) \cdot (1 + o_{\mathbb{P}}(1))\sqrt{2\log(n)}$.*

To interpret Theorem 2, we consider a special case where $K = O(1)$, $s_{K-1}(\tilde{V})$ is lower bounded by a constant, and we set $\delta_n = \log\log(n)/\log(n)$. By our assumption (7), $d = o(\sqrt{n})$. It follows that $\alpha_n \asymp \max\{d, \sqrt{d\log(n)}\}/\sqrt{n}$, $b_n \asymp \sigma\sqrt{\max\{d, \log(n)\}/n}$, and $t_n \asymp [\log(n)]^{\frac{1}{K-1}}/n^{\frac{1-o(1)}{K-1}}$. We observe that $\alpha_n$ always dominates $b_n/\sigma$. Whether $\alpha_n$ dominates $t_n$ is determined by $d/n$. When $d/n$ is properly small so that $\alpha_n \ll t_n$, using the first case in Theorem 2, we get $\max_k\{\|\hat{v}_k - v_k\|\} \leq C\big(\sqrt{\log(\log(n))} + \max\{d, \sqrt{d\log(n)}\}/\sqrt{n}\big) = O(\sqrt{\log\log(n)})$. When $d/n$ is properly large so that $\alpha_n \gg t_n$, using the second case in Theorem 2, we get $\max_k\{\|\hat{v}_k - v_k\|\} = O\big(\sqrt{\log(n)}\big)$. We then combine these two cases and further plug in the constants in Theorem 2. It yields

$$\max_{1 \leq k \leq K}\{\|\hat{v}_k^{\text{ppspa}} - v_k\|\} \leq \sigma g_{\text{new}}(V) \cdot \begin{cases} \sqrt{\log\log(n)} & \text{if } d/n \text{ is properly small;} \\ \sqrt{[2 + o(1)]\log(n)} & \text{if } d/n \text{ is properly large.} \end{cases} \quad (12)$$

It is worth comparing the error bound in Theorem 2 with that of the orthodox SPA (where we directly apply SPA on the original data points $X_1, X_2, \ldots, X_n$). Recall that $\beta(X)$ is as defined in (6). Note that $\beta(X) \leq \max_{1 \leq i \leq n}\|\epsilon_i\|$, where $\|\epsilon_i\|^2$ are i.i.d. variables from $\chi_d^2(0)$. Combining Lemma 5 and Theorem 1, we immediately obtain that for the (orthodox) SPA estimates $\hat{v}_1^{spa}, \hat{v}_2^{spa}, \ldots, \hat{v}_K^{spa}$, up to a permutation of these vectors (the constant $a_1$ is as in Lemma 5 and satisfies $a_1 > 2$):

$$\max_{1 \leq k \leq K}\{\|\hat{v}_k^{\text{spa}} - v_k\|\} \leq \sigma g_{\text{new}}(V) \cdot \begin{cases} \sqrt{\max\{d, 2\log(n)\}} & \text{if } d \ll \log(n) \text{ or } d \gg \log(n); \\ \sqrt{a_1 \log(n)} & \text{if } d = a_0 \log(n). \end{cases} \quad (13)$$

This bound is tight (e.g., when all $r_i$ fall into vertices). We compare (13) with Theorem 2. If $d \gg \log(n)$, the improvement is a factor of $\sqrt{\log(n)/d}$, which is huge when $d$ is large. If $d = O(\log(n))$, the improvement can still be a factor of $o(1)$ sometimes (e.g., in the first case of Theorem 2).

Theorem 2 assumes that there are a constant fraction of $r_i$ falling at each vertex. This can be greatly relaxed. The following theorem is proved in the appendix.

**Theorem 3** *Fix $0 < c_0 < 1$ and a sufficiently small constant $0 < \delta < c_0$. Suppose $X_1, X_2, \ldots, X_n$ are generated from model (1)-(2) where $m \geq n^{1-c_0+\delta}$ and conditions (7)-(9) hold. Let $t_n^* = \sqrt{K-1}\big( \frac{\log(n)}{n^{1-c_0}} \big)^{\frac{1}{K-1}}$. We apply pp-SPA to $X_1, X_2, \ldots, X_n$ with $(N, \Delta)$ to be determined below. Let $\hat{V} = [\hat{v}_1, \hat{v}_2, \ldots, \hat{v}_K]$, where $\hat{v}_1, \hat{v}_2, \ldots, \hat{v}_K$ are the estimated vertices.*

- *In the first case, $\alpha_n \ll t_n^*$. We take $N = \log(n)$ and $\Delta = c_3 t_n \sigma$ in pp-SPA, for a constant $c_3 \leq e^{c_0/(K-1)} c_2$. Up to a permutation of $\hat{v}_1, \ldots, \hat{v}_K$, $\max_{1 \leq k \leq K}\{\|\hat{v}_k - v_k\|\} \leq \sigma g_{\text{new}}(V)[\sqrt{c_0} \cdot \sqrt{2\log(n)} + C\alpha_n] + b_n$.*

- *In the second case, $\alpha_n \gg t_n^*$. Suppose $\alpha_n = o(1)$. We take $N = \log(n)$ and $\Delta = \alpha_n$ in pp-SPA. Up to a permutation of $\hat{v}_1, \ldots, \hat{v}_K$, $\max_{1 \leq k \leq K}\{\|\hat{v}_k - v_k\|\} \leq \sigma g_{\text{new}}(V) \cdot (1 + o_{\mathbb{P}}(1))\sqrt{2\log(n)}$.*

Comparing Theorem 3 with Theorem 2, the difference is in the first case, where the $o(1)$ factor of $\delta_n$ is replaced by a constant factor of $c_0 < 1$. Similarly as in (12), we obtain

$$\max_{1 \leq k \leq K}\{\|\hat{v}_k^{\text{ppspa}} - v_k\|\} \leq \sigma g_{\text{new}}(V) \cdot \begin{cases} \sqrt{2c_0 \log(n)} & \text{if } d/n \text{ is properly small;} \\ \sqrt{[2 + o(1)]\log(n)} & \text{if } d/n \text{ is properly large.} \end{cases} \quad (14)$$

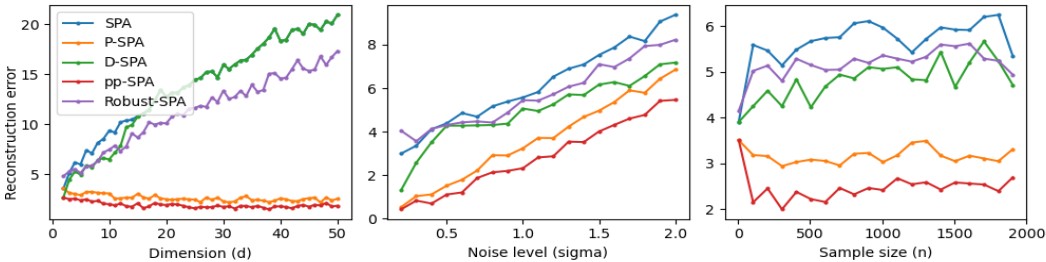

Figure 3: Performances of SPA, P-SPA, D-SPA, and pp-SPA in Experiment 1-3.

In this relaxed setting, we also compare Theorem 3 with (13): (a) When $d \gg \log(n)$, the improvement is a factor of $\sqrt{\log(n)/d}$. (b) When $d = O(\log(n))$, the improvement is at the constant order. It is interesting to further compare these "constants". Note that $g_{\mathrm{new}}(V)$ is the same for all methods. It suffices to compare the constants in the bound for $\beta_{\mathrm{new}}(V)$. In Case (b), the error bound of pp-SPA is smaller than that of SPA by a factor of $c_0 \in (0,1)$. For the practical purpose, even the improvement of a constant factor can have a huge impact, especially when the data contain strong noise and potential outliers. Our simulations in Section 5 further confirm this point.

## 5 NUMERICAL STUDY

We compare SPA, pp-SPA, and two simplified versions P-SPA and D-SPA (for illustration). We also compared these approaches with robust-SPA (Gillis, 2019) from `bit.ly/robustSPA` (with default tuning parameters). For pp-SPA and D-SPA, we need to specify tuning parameters $(N, \Delta)$. We use the heuristic choice in Remark 2. Fix $K = 3$ and three points $\{y_1, y_2, y_3\}$ in $\mathbb{R}^2$. Given $(n, d, \sigma)$, we first draw $(n-30)$ points uniformly from the 2-dimensional simplex whose vertices are $y_1, y_2, y_3$, and then put 10 points on each vertex of this simplex. Denote these points by $w_1, w_2, \ldots, w_n \in \mathbb{R}^2$. Next, we fix a matrix $A \in \mathbb{R}^{d \times 2}$, whose top $2 \times 2$ block is equal to $I_d$ and the remaining entries are zero. Let $r_i = Aw_i$, for all $i$. Finally, we generate $X_1, X_2, \ldots, X_n$ from model (1). We consider three experiments. In Experiment 1, we fix $(n, \sigma) = (1000, 1)$ and let $d$ range in $\{1, 2, \ldots, 49, 50\}$. In Experiment 2, we fix $(n, d) = (1000, 4)$ and let $\sigma$ range in $\{0.2, 0.3, \ldots, 2\}$. In Experiment 3, we fix $(d, \sigma) = (4, 1)$ and let $n$ range in $\{500, 600, \ldots, 1500\}$. We evaluate the vertex hunting error $\max_k\{\|\hat{v}_k - v_k\|\}$ (subject to a permutation of $\hat{v}_1, \ldots, \hat{v}_K$). For each set of parameters, we report the average error over 20 repetitions. The results are in Figure 3. They are consistent with our theoretical insights: The performances of P-SPA and D-SPA are both better than that of SPA, and the performance of pp-SPA is better than those of P-SPA and D-SPA. It suggests that both the projection and denoise steps are effective in reducing noise, and it is beneficial to combine them. When $d \leq 10$, pp-SPA, P-SPA and D-SPA all outperform robust-SPA; when $d > 10$, both pp-SPA and P-SPA outperform robust-SPA, and D-SPA (the simplified version without hyperplain projection) underperforms robust-SPA. The code to reproduce these experiments is available at `https://github.com/Gabriel78110/VertexHunting`.

## 6 DISCUSSION

Vertex hunting is a fundamental problem found in many applications. The Successive Projection algorithm (SPA) is a popular approach, but may behave unsatisfactorily in many settings. We propose pp-SPA as a new approach to vertex hunting. Compared to SPA, the new algorithm provides much improved theoretical bounds and encouraging improvements in a wide variety of numerical study. We also provide a sharper non-asymptotic bound for the orthodox SPA. For technical simplicity, our model assumes Gaussian noise, but our results are readily extendable to subGaussian noise. Also, our non-asymptotic bounds do not require any distributional assumption, and are directly applicable to different settings. For future work, we note that an improved bound on vertex hunting frequently implies improved bounds for methods that *contains vertex hunting as an important step*, such as Mixed-SCORE for network analysis (Jin et al., 2023; Bhattacharya et al., 2023), Topic-SCORE for text analysis (Ke & Wang, 2022), and state compression of Markov processes (Zhang & Wang, 2019), where vertex hunting plays a key role. Our algorithm and bounds may also be useful for related problems such as estimation of convex density support (Brunel, 2016).

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
