# Supplementary material for "Improved algorithm and bounds for successive projection"

**Jiashun Jin & Gabriel Moryoussef**
Department of Statistics
Carnegie Mellon University
Pittsburgh, PA 15213, USA
{jiashun, gmoryous}@andrew.cmu.edu

**Zheng Tracy Ke & Jiajun Tang & Jingming Wang**
Department of Statistics
Harvard University
Cambridge, MA 02138, USA
{zke,jiajuntang,jingmingwang}@fas.harvard.edu

## Contents

## A  PROOF OF PRELIMINARY LEMMAS

### A.1  PROOF OF LEMMA 1

This is a quite standard result, which can be found at tutorial materials (e.g., `https://people.math.wisc.edu/~roch/mmids/roch-mmids-llssvd-6svd.pdf`). We include a proof here only for convenience of readers.

We start by introducing some notation. Let $Z_i = X_i - \bar{X}$ and let $Z = [Z_1, \ldots, Z_n] \in \mathbb{R}^{d,n}$. Suppose the singular value decomposition of Z is given by $Z = U_Z D_Z V_Z'$. Since $H$ is a rank-$(K-1)$ projection matrix, we have $H = QQ'$, where $Q \in \mathbb{R}^{d,K-1}$ is such that $Q'Q = I_{K-1}$. Hence, we rewrite the optimization in (3) as follows:

$$\text{minimize} \sum_{i=1}^n (X_i - x_0)'(I_d - QQ')(X_i - x_0), \quad \text{subject to} \quad Q'Q = I_{K-1}.$$

For $\lambda \in \mathbb{R}$, consider the Lagrangian objective function

$$\widetilde{S}(x_0, Q, \lambda) = \sum_{i=1}^n (X_i - x_0)'(I_d - QQ')(X_i - x_0) + \lambda(Q'Q - I_{K-1}). \tag{A.1}$$

Setting its gradients w.r.t. $x_0$ and $Q$ to be 0 yields

$$\nabla_{x_0} \widetilde{S}(x_0, Q, \lambda) = -2(I_d - QQ') \sum_{i=1}^n (X_i - x_0) = 0, \tag{A.2}$$

$$\nabla_Q \widetilde{S}(x_0, Q, \lambda) = -2Q' \sum_{i=1}^n (X_i - x_0)(X_i - x_0)' + 2\lambda Q' = 0. \tag{A.3}$$

Firstly, we deduce from (A.2) that $\hat{x}_0 = \bar{X}$, which in view of (A.3) implies that $Q'(ZZ' - \lambda I_d) = 0$. The above equations also implies that the $(K-1)$ columns of $\widehat{Q}$ should be the distinct columns of $U_Z$. Now, the objective function in (A.1) is given by

$$\widetilde{S}(x_0, Q, \lambda) = \sum_{i=1}^n Z_i'(I_d - QQ')Z_i = \text{tr}[(I_d - QQ')ZZ'] = \text{tr}[(I_d - QQ')U_Z D_Z^2 U_Z']$$

$$= \text{tr}(D_Z)^2 - \text{tr}[Q'U_Z D_Z^2 U_Z'Q] = \text{tr}(D_Z^2) - \|D_Z U_Z'Q\|_{\text{F}}^2. \tag{A.4}$$

Note that for each column of $U_Z'Q \in \mathbb{R}^{d,K-1}$, it has exactly one entry being 1 and its other entries are all 0. Therefore, taking $\widehat{Q} = U$ maximizes $\|D_Z U_Z'Q\|_{\text{F}}^2$ and hence minimizes the objective function $\widetilde{S}$ in (A.1), that is, $\widehat{H} = UU'$. The proof is complete.

### A.2  PROOF OF LEMMA 3

For the simplex formed by $V \in \mathbb{R}^{d \times K}$, we can always find an orthogonal matrix $O \in \mathbb{R}^{d \times d}$ and a scalar $a$ such that

$$OV = \begin{pmatrix} x_1 & x_2 & \ldots & x_K \\ a & a & \ldots & a \\ 0 & 0 & \ldots & 0 \end{pmatrix}, \quad \text{where} \quad x_k \in \mathbb{R}^{K-1} \text{ for } k = 1, \ldots, K.$$

Denote $\bar{x} = K^{-1} \sum_{k=1}^K x_k$. Further we can represent

$$O\tilde{V} = \begin{pmatrix} x_1 - \bar{x} & x_2 - \bar{x} & \ldots & x_K - \bar{x} \\ 0 & 0 & \ldots & 0 \end{pmatrix}$$

We write $\tilde{X} := (x_1 - \bar{x}, x_2 - \bar{x}, \ldots, x_K - \bar{x})$. Since rotation and location do not change the volume,

$$\text{Volume}(\mathcal{S}_0) = \text{Volume}(\mathcal{S}(\tilde{X})).$$

where $\mathcal{S}(\tilde{X})$ represents the simplex formed by $\tilde{X}$. By Stein (1966), we have

$$\text{Volume}(\mathcal{S}_0) = \frac{\det(\tilde{A})}{(K-1)!}, \quad \text{with} \quad \tilde{A} = \begin{bmatrix} 1 & (x_1 - \bar{x})' \\ 1 & (x_2 - \bar{x})' \\ \vdots & \vdots \\ 1 & (x_K - \bar{x})' \end{bmatrix}$$

We also define

$$A = \begin{bmatrix} 1 & (v_1 - \bar{v})' \\ 1 & (v_2 - \bar{v})' \\ \vdots & \vdots \\ 1 & (v_K - \bar{v})' \end{bmatrix} = [\mathbf{1}_K, \tilde{V}'],$$

Since $(\tilde{A}, 0) = A \begin{pmatrix} 1 & 0 \\ 0 & O \end{pmatrix}$, it follows that $\tilde{A}\tilde{A}' = AA'$ and $\text{Volume}(\mathcal{S}_0) = \frac{\sqrt{\det(AA')}}{(K-1)!} = \frac{\sqrt{\det(A'A)}}{(K-1)!}$. Note that $A'A = \begin{pmatrix} K & 0 \\ 0 & \tilde{V}\tilde{V}' \end{pmatrix}$ by the fact that $\tilde{V}\mathbf{1}_K = 0$. Then $\det(A'A) = K\det(\tilde{V}\tilde{V}')$. Further notice that $\text{rank}(\tilde{V}\tilde{V}') = K - 1$. We thus conclude that

$$\text{Volume}(\mathcal{S}_0) = \frac{\sqrt{K}}{(K-1)!} \prod_{k=1}^{K-1} s_k(\tilde{V}).$$

This proves the first claim.

For the second and last claims, we first notice that $V = \tilde{V} - \bar{v}\mathbf{1}_K'$. Then $VV' = \tilde{V}\tilde{V}' + K\bar{v}\bar{v}'$ again by $\tilde{V}\mathbf{1}_K = 0$. Because both $\tilde{V}\tilde{V}'$ and $K\bar{v}\bar{v}'$ are positive semi-definite, by Weyl's inequality (see, for example Horn & Johnson (1985)), it follows that $s_{K-1}(V) \geq s_{K-1}(\tilde{V})$ and $s_K(V) = \sqrt{\lambda_{\min}(VV')} \leq \sqrt{K\|\bar{v}\|^2} = \sqrt{K}\|\bar{v}\|$.

### A.3 PROOF OF LEMMA 4

We first prove claim (a). Let $\Pi = [\pi_1 - \bar{\pi}, \ldots, \pi_n - \bar{\pi}] \in \mathbb{R}^{K,n}$. Recalling the definitions of $G$ and $V$, we have $G = n^{-1}\Pi\Pi'$ and $R = n^{-1/2}V\Pi$, so that $RR' = n^{-1}V\Pi\Pi'V' = VGV'$.

Next, we prove claim (b). Recall that $\tilde{V} = V - \bar{v}\mathbf{1}_K'$, so that $\tilde{V}\tilde{V}' = (V - \bar{v}\mathbf{1}_K')(V - \bar{v}\mathbf{1}_K')' = VV' - K\bar{v}\bar{v}'$. Note that Since $\pi_i'\mathbf{1}_K = \bar{\pi}'\mathbf{1}_K = 1$, we have $\Pi'\mathbf{1}_K = 0$, which implies that $G\mathbf{1}_K = n^{-1}\Pi(\Pi'\mathbf{1}_K) = 0$. We deduce from this observation that $\lambda_K(G) = 0$ and its associated eigenvector is $K^{-1/2}\mathbf{1}_K$. Therefore, $G - \lambda_{K-1}(G)I_K + K^{-1}\lambda_{K-1}(G)\mathbf{1}_K\mathbf{1}_K'$ is a positive semi-definite matrix, so that

$$VGV' - \lambda_{K-1}(G)\tilde{V}\tilde{V}' = VGV' - \lambda_{K-1}(G)VV' + \lambda_{K-1}(G)K\bar{v}\bar{v}'$$
$$= V[G - \lambda_{K-1}(G)I_K + K^{-1}\lambda_{K-1}(G)\mathbf{1}_K\mathbf{1}_K']V' \geq 0.$$

In addition, observing that $\Pi'\mathbf{1}_K = 0$ due to the fact that $\|\pi_i\|_1 = \|\bar{\pi}\|_1 = 1$, we obtain that

$$\tilde{V}G\tilde{V}' = (V - \bar{v}\mathbf{1}_K')G(V - \bar{v}\mathbf{1}_K')' = n^{-1}(V - \bar{v}\mathbf{1}_K')\Pi\Pi'(V - \bar{v}\mathbf{1}_K')' = VGV'.$$

Therefore,

$$\lambda_1(G)\tilde{V}\tilde{V}' - VGV' = \lambda_1(G)\tilde{V}\tilde{V}' - \tilde{V}G\tilde{V}' = \tilde{V}[\lambda_1(G)I_K - G]\tilde{V}' \geq 0,$$

which completes the proof of claim (b).

Finally, for claim (c), we obtain from (a) that $\sigma_{K-1}^2(R) = \lambda_{K-1}(RR') = \lambda_{K-1}(VGV')$, which by Weyl's inequality (see, for example, Horn & Johnson (1985)) and in view of claim (b) implies that $\lambda_{K-1}(G)\lambda_{K-1}(\tilde{V}\tilde{V}') \leq \sigma_{K-1}^2(R) \leq \lambda_1(G)\lambda_{K-1}(\tilde{V}\tilde{V}')$. The proof is therefore complete.

A.4 PROOF OF LEMMA 5

Recall that $z_1 \sim \chi_d^2(0)$. Let $b_n$ be the value such that

$$\mathbb{P}(z_1 \geq b_n) = 1/n.$$

By basic extreme value theory, it is known that

$$\frac{\max_{1 \leq i \leq n}\{z_i\}}{b_n} \to 1, \qquad \text{in probability.}$$

We now solve for $b_n$. It is seen that $b_n \geq d$. Recall that the density of $\chi_d^2(0)$ is

$$\frac{1}{2^{d/2}\Gamma(d/2)}x^{d/2-1}e^{-x/2}, \qquad x > 0.$$

Note that for any $x_0 \geq d$,

$$\int_{x_0}^{\infty} x^{d/2-1}e^{-x/2}dx = 2x_0^{d/2-1}e^{-x_0/2} + \int_{x_0}^{\infty}(d-2)x^{d/2-2}e^{-x/2}dx \qquad (A.5)$$

where the RHS is no greater than

$$\leq 2x_0^{d/2-1}e^{-x_0/2} + \frac{(d-2)}{x_0}\int_{x_0}^{\infty}x^{d/2-1}e^{-x/2}dx.$$

It follows that for all $x_0 \geq d$,

$$2x_0^{d/2-1}e^{-x_0/2} \leq \int_{x_0}^{\infty} x^{d/2-1}e^{-x/2}dx \leq x_0 \cdot x_0^{d/2-1}e^{-x_0/2}, \qquad (A.6)$$

where we have used

$$\frac{x_0}{x_0 - d + 2} \leq x_0/2.$$

It now follows that there is a term $a(x)$ such that when $x \geq d$,

$$1 \leq a(x) \leq x/2$$

and

$$\mathbb{P}(z_1 \geq x) = a(x)\frac{1}{2^{d/2}\gamma(d/2)}2x^{d/2-1}e^{-x/2}.$$

Combining these, $b_n$ is the solution of

$$a(x)\frac{1}{2^{d/2}\gamma(d/2)}2x^{d/2-1}e^{-x/2} = \frac{1}{n}. \qquad (A.7)$$

We now solve the equation in (A.7). Consider the case $d$ is even. The case where $d$ is odd is similar, so we omit it. When $d$ is even, using

$$\Gamma(d/2) = (d/2 - 1)! = (2/d)(d/2)! = (2/d)\theta(\frac{d}{2e})^{d/2},$$

where $\theta$ is the factor in the Stirling's formula which is $\leq C\sqrt{\log(d)}$. Plugging this into the left hand side of (A.7) and re-arrange, we have

$$\log(d/x) + (d/2)\log(\frac{ex}{d}) - x/2 = -\log(n) + o(\log(n)). \qquad (A.8)$$

We now consider three cases below separately.

- Case 1. $d \ll \log(n)$.
- Case 2. $d = a_0 \log(n)$ for a constant $a_0 > 0$.
- Case 3. $d \gg \log(n)$.

Consider Case 1. In this case, it is seen that when

$$x = O(\log(n)),$$

the LHS of (A.8) is

$$-x/2 + o(\log(n)).$$

Therefore, the solution of (A.8) is seen to be

$$b_n = (1 + o(1)) \cdot 2 \log(n).$$

Consider Case 2. In this case, $d = a_0 \log(n)$. Let $x = b_1 \log(n)$. Plugging these into (A.8) and rearranging,

$$a_1 - a_0 \log(a_1) = 2 + a_0 - a_0 \log(a_0) + o(1). \tag{A.9}$$

Now, consider the equation

$$a_1 - a_0 \log(a_1) = 2 + a_0 - a_0 \log(a_0).$$

It is seen that the equation has a unique solution (denoted by $b_0$) that is bigger than 2. Therefore, in this case,

$$b_n = (1 + o(1))b_0,$$

Consider Case 3. In this case, $d \gg \log(n)$. Consider again the equation

$$\log(d/x) + (d/2) \log(\frac{ex}{d}) - x/2 = -\log(n) + o(\log(n)).$$

Letting $y = x/d$ and rearranging, it follows that

$$y - \log(y) - 1 = o(1), \tag{A.10}$$

where for sufficiently large $n$, $o(1) > 0$ and $o(1) \to 0$. Note that the function $g(y) = y - \log(y) - 1$ is a convex function with a minimum of 0 reached at $y = 1$, it follows

$$y = 1 + o(1).$$

Recalling $y = x/d$, this shows

$$b_n = (1 + o(1))d.$$

This completes the proof of Lemma 5.

# B ANALYSIS OF THE SPA ALGORITHM

Fix $d \geq K - 1$. For any $V = [v_1, v_2, \ldots, v_K] \in \mathbb{R}^{d \times K}$, let $\sigma_k(V)$ denote the $k$th singular value of $V$, and define

$$\gamma(V) = \min_{v_0 \in \mathbb{R}^d} \max_{1 \leq k \leq K} \|v_k - v_0\|, \qquad d_{\max}(V) = \max_{x \in \mathcal{S}} \|x\|.$$

To capture the error bound for SPA, we introduce a useful quantity in the main paper:

$$\beta(X, V) := \max\left\{\max_{1 \leq i \leq n} \text{Dist}(X_i, \mathcal{S}), \quad \max_{1 \leq k \leq K} \min_{i: r_i = v_k} \|X_i - v_k\|\right\}. \tag{B.11}$$

We note that when $\max_i \text{Dist}(X_i, \mathcal{S})$ is small, no point is too far away from the simplex; and when $\max_k \min_{i:r_i = v_k} \|X_i - v_k\|$ is small, there is at least one point near each vertex.

Let's denote $\gamma = \gamma(V)$, $d_{\max} = d_{\max}(V)$, $\beta = \beta(X, V)$, and $\sigma_* = \sigma_{K-1}(V)$ for brevity. We shall prove the following theorem, which is a slightly stronger version of Theorem 1 in the main paper.

**Theorem A.** *Suppose for each $1 \leq k \leq K$, there exists $1 \leq i \leq n$ such that $\pi_i = e_k$. Suppose $\beta(X, V)$ satisfies that $450 d_{\max} \max\{1, \frac{d_{\max}}{\sigma_*}\}\beta \leq \sigma_*^2$. Let $\hat{v}_1, \hat{v}_2, \ldots, \hat{v}_r$ be the output of SPA. Up to a permutation of these $r$ vectors,*

$$\max_{1 \leq k \leq r} \|\hat{v}_k - v_k\| \leq \left(1 + \frac{30\gamma}{\sigma_*} \max\{1, \frac{d_{\max}}{\sigma_*}\}\right)\beta(X, V).$$

## B.1 SOME PRELIMINARY LEMMAS IN LINEAR ALGEBRA

To establish Theorem A, it is necessary to develop a few lemmas in linear algebra. First, we notice that the vertex matrix $V$ defines a mapping from the standard probability simplex $\mathcal{S}^*$ to the target simplex $\mathcal{S}$. The following lemma gives some properties of the mapping:

**Lemma A.** *Let $\mathcal{S}^* \subset \mathbb{R}^K$ be the standard probability simplex consisting of all weight vectors. Let $F : \mathcal{S}^* \to \mathcal{S}$ be the mapping with $F(\pi) = V\pi$. For any $\pi$ and $\tilde{\pi}$ in $\mathcal{S}^*$,*

$$\sigma_{K-1}(V) \cdot \|\pi - \tilde{\pi}\| \leq \|F(\pi) - F(\tilde{\pi})\| \leq \gamma(V) \cdot \|\pi - \tilde{\pi}\|_1. \tag{B.12}$$

*Fix $1 \leq s \leq K - 2$. If $\pi$ and $\tilde{\pi}$ share at least $s$ common entries, then*

$$\|F(\pi) - F(\tilde{\pi})\| \geq \sigma_{K-1-s}(V)\|\pi - \tilde{\pi}\|. \tag{B.13}$$

The first claim of Lemma A is about the case where $\mathcal{S}$ is non-degenerate. In this case,

$$\sigma_{K-1}(V) > 0.$$

Hence, we can upper/lower bound the distance between any two points in $\mathcal{S}$ by the distance between their barycentric coordinates. The second claim considers the case where $\mathcal{S}$ can be degenerate (i.e., $\sigma_{K-1}(V) = 0$ is possible) but

$$\sigma_{K-1-s}(V) > 0.$$

We can still use (B.12) to upper bound the distance between two points in $\mathcal{S}$ but the lower bound there is ineffective. Fortunately, if the two points share $s$ common entries in their barycentric coordinates (which implies that the two points are on the same face or edge), then we can still lower bound the distance between them.

Second, we study the Euclidean norm of a convex combination of $m$ points. Let $w_1, \ldots, w_m$ be the convex combination weights. By the triangle inequality,

$$\left\|\sum_{i=1}^m w_i x_i\right\| \leq \sum_{i=1}^m w_i \|x_i\| \leq \max_{1 \leq k \leq K} \|v_k\|.$$

This explains why $\max_{x \in \mathcal{S}} \|x\|$ is always attained at a vertex. Write

$$\delta := \sum_{i=1}^m w_i \|x_i\| - \left\|\sum_{i=1}^m w_i x_i\right\|.$$

Knowing $\delta \geq 0$ is not enough for showing Theorem A. We need to have an explicit lower bound for $\delta$, as given in the following lemma.

**Lemma B.** *Fix $m \geq 2$ and $x_1, \ldots, x_m \in \mathbb{R}^d$. Let $a = \min_{i \neq j} \|x_i - x_j\|$ and $b = \max_{i \neq j} |\|x_i\| - \|x_j\||$. For any $w_1, \ldots, w_m \geq 0$ such that $\sum_{i=1}^{m} w_i = 1$,*

$$\left\| \sum_{i=1}^{m} w_i x_i \right\| \leq L - \frac{a^2 - b^2}{4L} \sum_{i=1}^{m} w_i(1 - w_i), \quad with \ \ L := \sum_{i=1}^{m} w_i \|x_i\|. \tag{B.14}$$

By Lemma B, the lower bound for $\delta$ has the expression $\frac{a^2 - b^2}{4L} \sum_{i=1}^{m} w_i(1 - w_i)$. This lower bound is large if $a = \min_{i \neq j} \|x_i - x_j\|$ is properly large, and $b = \max_{i \neq j} |\|x_i\| - \|x_j\||$ is properly small, and $\sum_i w_i(1 - w_i)$ is properly large.

- A large $a$ means that these $m$ points are sufficiently 'different' from each other.
- A small $b$ means that the norms of these $m$ points are sufficiently close.
- A large $\sum_i w_i(1 - w_i)$ prevents each of $w_i$ from being too close to 1, implying that the convex combination is sufficiently 'mixed'.

Later in Section B.2, we will see that Lemma B plays a critical role in the proof of Theorem A.

Third, we explore the projection of $\mathcal{S}$ into a lower-dimensional space. Let $H \in \mathbb{R}^{d \times d}$ be an arbitrary projection matrix with rank $s$. We use $(I_d - H)$ to project $\mathcal{S}$ into the orthogonal complement of $H$, where the projected vertices are the columns of

$$V^{\perp} = (I_d - H)V.$$

Since the projected simplex is not guranteed to be non-degenerate, it is possible that $\sigma_{K-1}(V^{\perp}) = 0$. However, we have a lower bound for $\sigma_{K-1-s}(V^{\perp})$, as given in the following lemma:

**Lemma C.** *Fix $1 \leq s \leq K - 2$. For any projection matrix $H \in \mathbb{R}^{d \times d}$ with rank $s$,*

$$\sigma_{K-1-s}((I_d - H)V) \geq \sigma_{K-1}(V). \tag{B.15}$$

Finally, we present a lemma about

$$d_{\max} = \max_{x \in \mathcal{S}} \|x\| = \max_{1 \leq k \leq K} \|v_k\|.$$

In the analysis of SPA, it is not hard to get a lower bound for $d_{\max}$ in the first iteration. However, as the algorithm successively projects $\mathcal{S}$ into lower-dimensional subspaces, we need to keep track of this quantity for the projected simplex spanned by $V^{\perp}$. Lemma C shows that the singular values of $V^{\perp}$ can be lower bounded. It motivates us to have a lemma that provides a lower bound of $d_{\max}$ in terms of the singular values of $V$.

**Lemma D.** *Fix $0 \leq s \leq K - 2$. Suppose there are at least $s$ indices, $\{k_1, \ldots, k_s\} \subset \{1, 2, \ldots, K\}$, such that $\|v_k\| \leq \delta$. If $\sigma_{K-1-s}^2(V) \geq 2(K-2)\delta^2$, then*

$$\max_{1 \leq k \leq K} \|v_k\| \geq \frac{\sqrt{K - s - 1}}{\sqrt{2(K - s)}} \sigma_{K-1-s}(V) \geq \frac{1}{2} \sigma_{K-1-s}(V). \tag{B.16}$$

## B.2 THE SIMPLICIAL NEIGHBORHOODS AND A KEY LEMMA

We fix a simplex $\mathcal{S} \subset \mathbb{R}^d$ whose vertices are $v_1, v_2, \ldots, v_K$. Write $V = [v_1, v_2, \ldots, v_K] \in \mathbb{R}^{d \times K}$. Let $\mathcal{S}^*$ denote the standard probability simplex, and let $F : \mathcal{S}^* \to \mathcal{S}$ be the mapping in Lemma A. We introduce a local neighborhood for each vertex that has a "simplex shape":

**Definition B.1.** *Given $\epsilon \in (0, 1)$, for each $1 \leq k \leq K$, the $\epsilon$-simplicial-neighborhood of $v_k$ inside the simplex $\mathcal{S}$ is defined by*

$$\mathcal{V}_k(\epsilon) := \{F(\pi) : \pi \in \mathcal{S}^*, \ \pi(k) \geq 1 - \epsilon\}.$$

These simplicial neighborhoods are highlighted in blue in Figure 1.

First, we verify that each $\mathcal{V}_k(\epsilon)$ is indeed a "neighborhood" in the sense each $x \in \mathcal{V}_k(\epsilon)$ is sufficiently close to $v_k$. Note that $v_k = F(e_k)$, where $e_k$ is the $k$th standard basis vector of $\mathbb{R}^K$. For any $\pi \in \mathcal{S}^*$,

$$\|\pi - e_k\|_1 = 2[1 - \pi(k)].$$

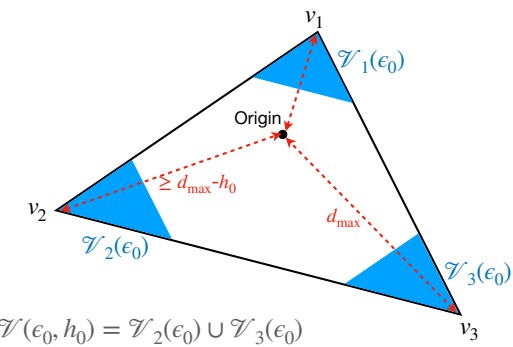

Figure 1: An illustration of the simplicial neighborhoods and $\mathcal{V}(\epsilon_0, h_0)$.

By Definition B.1, for any $x \in \mathcal{V}_k(\epsilon)$, its barycentric coordinate $\pi$ satisfies $1 - \pi(k) \leq \epsilon$. It follows by Lemma A that

$$\max_{x \in \mathcal{V}_k(\epsilon)} \|x - v_k\| = \max_{\pi \in \mathcal{S}^*:\pi(k) \leq 1-\epsilon} \|F(\pi) - F(e_k)\| \leq 2\gamma(V)\epsilon. \tag{B.17}$$

Hence, $\mathcal{V}_k(\epsilon)$ is within a ball centered at $v_k$ with a radius of $2\gamma(V)\epsilon$. However, we opt to utilize these simplex-shaped neighborhoods instead of standard balls, as this choice greatly simplifies proofs.

Next, we show that as long as $\epsilon < 1/2$, the $K$ neighborhoods $\mathcal{V}_1(\epsilon), \ldots, \mathcal{V}_K(\epsilon)$ are non-overlapping. By Lemma A,

$$\|v_k - v_\ell\| \geq \sigma_{K-1}(V)\|e_k - e_\ell\| \geq \sqrt{2}\sigma_{K-1}(V), \qquad \text{for } 1 \leq k \neq \ell \leq K. \tag{B.18}$$

When $x \in \mathcal{V}_k(\epsilon)$, the $k$th entry of $\pi := F^{-1}(x)$ is at least $1 - \epsilon > 1/2$. Since each $\pi \in \mathcal{S}^*$ cannot have two entries larger than $1/2$, these neighborhoods are disjoint:

$$\mathcal{V}_k(\epsilon) \cap \mathcal{V}_\ell(\epsilon) = \emptyset, \qquad \text{for any } 1 \leq k \neq \ell \leq K. \tag{B.19}$$

**An intuitive explanation of our proof ideas for Theorem A**: We outline our proof strategy using the example in Figure 1. The first step of SPA finds

$$i_1 = \text{argmax}_{1 \leq i \leq n} \|X_i\|.$$

The population counterpart of $X_{i_1}$ is denoted by $r_{i_1}$. We will explore the region of the simplex that $r_{i_1}$ falls into. In the noiseless case, $X_i = r_i$ for all $1 \leq i \leq n$. Since the maximum Euclidean norm over a simplex can only be attained at vertex, $r_{i_1}$ must equal to one of the vertices. In Figure 1, the vertex $v_3$ has the largest Euclidean norm, hence, $r_{i_1} = v_3$ in the noiseless case. In the noisy case, the index $i$ that maximizes $\|X_i\|$ may not maximize $\|r_i\|$; i.e., $r_{i_1}$ may not have the largest Euclidean norm among $r_i$'s. Noticing that $\|v_3\| > \|v_2\| > \|v_1\|$, we expect to see two possible cases:

- Possibility 1: $r_{i_1}$ is in the $\epsilon$-simplicial-neighborhood of $v_3$, for a small $\epsilon > 0$.
- Possibility 2 (when $\|v_2\|$ is close to $\|v_3\|$): $r_{i_1}$ is in the $\epsilon$-simplicial-neighborhood of $v_2$.

The focus of our proof will be showing that $r_{i_1}$ falls into $\mathcal{V}_2(\epsilon) \cup \mathcal{V}_3(\epsilon)$. No matter $r_i \in \mathcal{V}_2(\epsilon)$ holds or $r_i \in \mathcal{V}_3(\epsilon)$ holds, the corresponding $\hat{v}_1 = X_{i_1}$ is close to one of the vertices.

**Formalization of the above insights, and a key lemma**: Introduce the notation

$$\mathcal{K}^* = \{k : \|v_k\| = d_{\max}\}, \qquad \text{where} \quad d_{\max} := \max_{x \in \mathcal{S}} \|x\| = \max_k \|v_k\|. \tag{B.20}$$

Given any $h_0 > 0$ and $\epsilon_0 \in (0, 1/2)$, let $\mathcal{V}_k(\epsilon_0)$ be the same as in Definition B.1, and we define an index set $\mathcal{K}(h_0)$ and a region $\mathcal{V}(\epsilon_0, h_0) \subset \mathcal{S}$ as follows:

$$\mathcal{K}(h_0) = \{k : \|v_k\| \geq d_{\max} - h_0\}, \qquad \mathcal{V}(\epsilon_0, h_0) = \cup_{k \in \mathcal{K}(h_0)} \mathcal{V}_k(\epsilon_0), \tag{B.21}$$

For the example in Figure 1, $\mathcal{K}^* = \{3\}$, $\mathcal{K}(h_0) = \{2, 3\}$, and $\mathcal{V}(\epsilon_0, h_0) = \mathcal{V}_2(\epsilon_0) \cup \mathcal{V}_3(\epsilon_0)$.

In the proof of Theorem A, we will repeatedly use the following key lemma, which states that the Euclidean norm of any point in $\mathcal{S} \setminus \mathcal{V}(\epsilon_0, h_0)$ is strictly smaller than $d_{\max}$ by a certain amount:

**Lemma E.** *Fix a simplex $\mathcal{S} \subset \mathbb{R}^d$ with vertices $v_1, v_2, \ldots, v_K$. Write $d_{\max} = \max_{1 \leq k \leq K} \|v_k\|$. Suppose there exists $\sigma_* > 0$ such that*

$$d_{\max} \geq \sigma_*/2, \qquad \text{and} \qquad \min_{1 \leq k \neq \ell \leq K} \|v_k - v_\ell\| \geq \sqrt{2}\sigma_*. \tag{B.22}$$

*Let $\mathcal{K}(h_0)$ and $\mathcal{V}(\epsilon_0, h_0)$ be as defined in (B.21). Given any $t > 0$ such that $\max\{1, d_{\max}/\sigma_*\}t < 3\sigma_*$, if we set $(h_0, \epsilon_0)$ such that*

$$h_0 = \sigma_*/3, \qquad \text{and} \qquad 1/2 > \epsilon_0 \geq 6\sigma_*^{-1}\max\{1, d_{\max}/\sigma_*\}t, \tag{B.23}$$

*then*

$$\|x\| \leq d_{\max} - t, \qquad \text{for all } x \in \mathcal{S} \setminus \mathcal{V}(\epsilon_0, h_0). \tag{B.24}$$

Lemma E will be proved in Section B.4.5, where we invoke Lemma B to prove the claim here.

## B.3   PROOF OF THEOREM A (THEOREM 1 IN THE MAIN PAPER)

The proof consists of three steps. In Step 1, we study the first iteration of SPA and show that $\hat{v}_1$ falls in the neighborhood of a true vertex. In Steps 2-3, we recursively study the remaining iterations and show that, if $\hat{v}_1, \ldots, \hat{v}_{s-1}$ fall into the neighborhoods of $(s-1)$ true vertices, one per each, then $\hat{v}_s$ will also fall into the neighborhood of another true vertex. For clarity, we first study the second iteration in Step 2 (for which the notations are simpler), and then study the $s$th iteration for a general $s$ in Step 3.

Let's denote for brevity:

$$\gamma = \gamma(V), \qquad d_{\max} = d_{\max}(V), \qquad \sigma_* = \sigma_{K-1}(V), \qquad \beta = \beta(X, V).$$

Write $J_k = \{1 \leq i \leq n : \pi_i(k) = 1\}$, for $1 \leq k \leq K$. By the definition of $\beta(X, V)$,

$$\max_{1 \leq i \leq n} \text{Dist}(X_i, \mathcal{S}) \leq \beta, \qquad \max_{1 \leq k \leq K} \min_{i \in J_k} \|X_i - v_k\| \leq \beta. \tag{B.25}$$

**Step 1: Analysis of the first iteration of SPA**.

Applying Lemma D with $s = 0$, we have $d_{\max} \geq \sigma_*/2$. We then apply Lemma E. Let $\mathcal{V}(\epsilon_0, h_0)$ be as in (B.21), with

$$h_0 = \sigma_*/3, \qquad \text{and} \qquad \epsilon_0 = 15\max\{\sigma_*, \sigma_*^{-2}d_{\max}\}\beta. \tag{B.26}$$

Our assumptions yield $\epsilon_0 < 1/2$. Additionally, when $t = 7\beta/3$, $\epsilon_0 \geq 6\sigma_*^{-1}\max\{1, d_{\max}/\sigma^*\}t$, which satisfies (B.23). We apply Lemma E with $t = 7\beta/3$. It yields

$$\max_{x \in \mathcal{S} \setminus \mathcal{V}(\epsilon_0, h_0)} \|x\| \leq d_{\max} - 7\beta/3. \tag{B.27}$$

At the same time, let $\mathcal{K}^*$ be the same as in (B.20). For any $k \in \mathcal{K}^*$, it follows by (B.25) that

there exists at least one $i^* \in J_k$ such that $\|X_{i^*} - v_k\| \leq \beta$.

Note that $\|v_k\| = d_{\max}$ for $k \in \mathcal{K}^*$. It follows by the triangle inequality that

$$\|X_{i^*}\| \geq \|v_k\| - \beta \geq d_{\max} - \beta.$$

Since $\|X_{i_1}\| = \max_i \|X_i\|$, we immediately have:

$$\|X_{i_1}\| \geq \|X_{i^*}\| \geq d_{\max} - \beta. \tag{B.28}$$

Combining (B.27) and (B.28), we conclude that $X_{i_1} \notin \mathcal{S} \setminus \mathcal{V}(\epsilon_0, h_0)$; in other words,

$X_{i_1}$ can only be inside $\mathcal{V}(\epsilon_0, h_0)$ or outside $\mathcal{S}$. $\tag{B.29}$

Suppose $X_{i_1}$ is outside $\mathcal{S}$. Let $\text{proj}_{\mathcal{S}}(X_{i_1}) \in \mathbb{R}^d$ be the point in the simplex that is closest to $X_{i_1}$. In other words, $\|X_{i_1} - \text{proj}_{\mathcal{S}}(X_{i_1})\| = \min_{x \in \mathcal{S}} \|X_{i_1} - x\| = \text{Dist}(X_{i_1}, \mathcal{S})$. Using the first inequality in (B.25), we have

$$\|X_{i_1} - \text{proj}_{\mathcal{S}}(X_{i_1})\| \leq \beta. \tag{B.30}$$

It follows by the triangle inequality and (B.28) that

$$\|\text{proj}_{\mathcal{S}}(X_{i_1})\| \geq \|X_{i_1}\| - \beta \geq d_{\max} - 2\beta.$$

Combining it with (B.27), we conclude that $\mathrm{proj}_{\mathcal{S}}(X_{i_1})$ cannot be in $\mathcal{S} \setminus \mathcal{V}(\epsilon_0, h_0)$. So far, we have shown that one of the following cases must happen:

$$\text{Case 1: } X_{i_1} \in \mathcal{V}(\epsilon_0, h_0),$$
$$\text{Case 2: } X_{i_1} \notin \mathcal{S}, \text{ and } \mathrm{proj}_{\mathcal{S}}(X_{i_1}) \in \mathcal{V}(\epsilon_0, h_0). \tag{B.31}$$

In Case 1, since $\mathcal{V}_1(\epsilon_0), \dots, \mathcal{V}_K(\epsilon_0)$ are disjoint, there exists only one $k_1 \in \mathcal{K}(h_0)$ such that $X_{i_1} \in \mathcal{V}_{k_1}(\epsilon_0)$. It follows by (B.17) that

$$\|X_{i_1} - v_{k_1}\| \leq 2\gamma\epsilon_0, \qquad \text{in Case 1.} \tag{B.32}$$

In Case 2, similarly, there is only one $k_1 \in \mathcal{K}(h_0)$ such that $\mathrm{proj}_{\mathcal{S}}(X_{i_1}) \in \mathcal{V}_{k_1}(\epsilon_0)$. It follows by (B.17) again that

$$\|\mathrm{proj}_{\mathcal{S}}(X_{i_1}) - v_{k_1}\| \leq 2\gamma\epsilon_0.$$

Combining it with (B.30) gives

$$\|X_{i_1} - v_{k_1}\| \leq \|X_{i_1} - \mathrm{proj}_{\mathcal{S}}(X_{i_1})\| + \|\mathrm{proj}_{\mathcal{S}}(X_{i_1}) - v_{k_1}\|$$
$$\leq 2\gamma\epsilon_0 + \beta, \qquad \text{in Case 2.} \tag{B.33}$$

We put (B.32) and (B.33) together and plug in the value of $\epsilon_0$ in (B.26). It yields:

$$\|X_{i_1} - v_{k_1}\| \leq \beta + 2\gamma\epsilon_0$$
$$\leq \left(1 + \frac{30\gamma}{\sigma_*} \max\left\{1, \frac{d_{\max}}{\sigma_*}\right\}\right)\beta, \qquad \text{for some } k_1. \tag{B.34}$$

**Step 2: Analysis of the second iteration of SPA**.

Let $H_1 = I_d - \frac{1}{\|X_{i_1}\|^2} X_{i_1} X'_{i_1}$ and $\widetilde{X}_i = H_1 X_i$, for $1 \leq i \leq n$. The second iteration operates on the data points $\widetilde{X}_1, \dots, \widetilde{X}_n \in \mathbb{R}^d$. Write

$$\tilde{r}_i = H_1 r_i, \qquad \tilde{\epsilon}_i = H_1 \epsilon_i, \qquad \tilde{v}_k = H_1 v_k, \qquad \widetilde{V} = [\tilde{v}_1, \tilde{v}_2, \dots, \tilde{v}_K].$$

It follows that

$$\widetilde{X}_i = \widetilde{V}\pi_i + \tilde{\epsilon}_i, \qquad 1 \leq i \leq n. \tag{B.35}$$

Let $\widetilde{S} \subset \mathbb{R}^d$ denote the projected simplex, whose vertices are $\tilde{v}_1, \dots, \tilde{v}_K$. Let $\widetilde{F}$ denote the mapping from the standard probability simplex $\mathcal{S}^*$ to the projected simplex $\widetilde{S}$ (note that $\widetilde{F}$ is not necessarily a one-to-one mapping). We consider the neighborhoods of $\widetilde{S}$ using Definition B.1

$$\widetilde{\mathcal{V}}_k(\epsilon) = \left\{\widetilde{F}(\pi) : \pi \in \mathcal{S}^*, \pi_i(k) \geq 1 - \epsilon\right\} \subset \mathbb{R}^d, \qquad 1 \leq k \leq K. \tag{B.36}$$

Let $k_1$ be as in (B.34). Let $\tilde{d}_{\max} := \max_{x \in \widetilde{S}} \|x\|$. The maximum distance $\tilde{d}_{\max}$ is attained at one or multiple vertices. Same as before, let $\widetilde{\mathcal{K}}^*$ be the index set of $k$ at which $\|\tilde{v}_k\| = \tilde{d}_{\max}$. We similarly define

$$\widetilde{\mathcal{K}}(h_0) = \{k : \|\tilde{v}_k\| \geq \tilde{d}_{\max} - h_0\}, \qquad \widetilde{\mathcal{V}}(\epsilon_0, h_0) = \cup_{k \in \widetilde{\mathcal{K}}(h_0)} \widetilde{\mathcal{V}}_k(\epsilon_0). \tag{B.37}$$

At the same time, let $\tilde{\beta} = \beta(\widetilde{X}, \widetilde{V})$. It is easy to see that for any points $x$ and $y$, $\|H_1 x - H_1 y\| \leq \|x - y\|$. Hence, $\tilde{\beta} \leq \beta$. It follows that

$$\max_{1 \leq i \leq n} \mathrm{Dist}(\widetilde{X}_i, \widetilde{S}) \leq \beta, \qquad \max_{1 \leq k \leq K} \min_{i \in J_k} \|\widetilde{X}_i - \tilde{v}_k\| \leq \beta. \tag{B.38}$$

Additionally, we have the following lemma:

**Lemma F.** *Under the conditions of Theorem A, for $\sigma_* = \sigma_{K-1}(V)$, the following claims are true:*

$$\tilde{d}_{\max} \geq \sigma_*/2, \qquad \min_{\substack{(k,\ell):k \neq k_1, \\ \ell \neq k_1, k \neq \ell}} \|\tilde{v}_k - \tilde{v}_\ell\| \geq \sqrt{2}\sigma_*, \quad \text{and} \quad k_1 \notin \widetilde{\mathcal{K}}(h_0). \tag{B.39}$$

Given (B.35)-(B.39), we now apply Lemma E to study the projected simplex $\widetilde{S}$. Similarly as how we obtain (B.27), by choosing

$$h_0 = \sigma_*/3, \qquad \text{and} \qquad \epsilon_1 = 15 \max\{\sigma_*, \sigma_*^{-2}\tilde{d}_{\max}\},$$

we get $\max_{x \in \widetilde{S} \setminus \widetilde{\mathcal{V}}(\epsilon_1, h_0)} \|x\| \leq \tilde{d}_{\max} - 7\beta/3$. Note that $\epsilon_1 \leq \epsilon_0$, and the set $\widetilde{S} \setminus \widetilde{V}(\epsilon, h_0)$ becomes smaller as $\epsilon$ increases. We immediately have

$$\max_{x \in \widetilde{S} \setminus \widetilde{\mathcal{V}}(\epsilon_0, h_0)} \|x\| \leq \tilde{d}_{\max} - 7\beta/3. \tag{B.40}$$

At the same time, by (B.38) and (B.39), it is easy to get (similar to how we obtained (B.28))

$$\|\tilde{X}_{i_2}\| \geq \tilde{d}_{\max} - \beta.$$

We can mimic the analysis between (B.28) and (B.31) to show that one of the two cases happens:

$$\text{Case 1: } \widetilde{X}_{i_2} \in \widetilde{\mathcal{V}}(\epsilon_0, h_0),$$
$$\text{Case 2: } \widetilde{X}_{i_2} \notin \widetilde{\mathcal{S}}, \text{ and } \text{proj}_{\widetilde{\mathcal{S}}}(\widetilde{X}_{i_2}) \in \widetilde{\mathcal{V}}(\epsilon_0, h_0). \tag{B.41}$$

Consider Case 1. Since $H_1$ is a linear projector, $\widetilde{X}_i \in \widetilde{\mathcal{V}}_k(\epsilon_0)$ if and only if $X_i \in \mathcal{V}_k(\epsilon_0)$. Hence,

$$X_{i_2} \in \left( \cup_{k \in \widetilde{\mathcal{K}}(h_0)} \mathcal{V}_k(\epsilon_0) \right).$$

There exists a unique $k_2 \in \widetilde{\mathcal{K}}(h_0)$ such that $X_{i_2} \in \mathcal{V}_{k_2}(\epsilon_0)$. It follows by (B.17) that

$$\|X_{i_2} - v_{k_2}\| \leq 2\gamma\epsilon_0, \qquad \text{in Case 1.}$$

Consider Case 2. Write $\tilde{x} = \text{proj}_{\widetilde{\mathcal{S}}}(\widetilde{X}_{i_2})$ for short, and let $M = \{x \in \mathcal{S} : H_1 x = \tilde{x}\}$. For any $k$, $\tilde{x} \in \widetilde{\mathcal{V}}_k(\epsilon_0)$ implies that $x \in \mathcal{V}_k(\epsilon_0)$ for every $x \in M$. Additionally, $\widetilde{X}_i \in \widetilde{\mathcal{S}}$ if and only if $X_i \in \mathcal{S}$. Hence, it holds in Case 2 that

$$X_{i_2} \notin \mathcal{S}, \text{ and } x \in \left( \cup_{k \in \widetilde{\mathcal{K}}(h_0)} \mathcal{V}_k(\epsilon_0) \right), \text{ for every } x \in M.$$

We pick one $x \in M$. There exists a unique $k_2 \in \widetilde{\mathcal{K}}(h_0)$ such that $x \in \mathcal{V}_{k_2}(\epsilon_0)$. By mimicking the derivation of (B.33), we obtain that

$$\|X_{i_2} - v_{k_2}\| \leq 2\gamma\epsilon_0 + \beta, \qquad \text{in Case 2.}$$

Combining the two cases and using the value of $\epsilon_0$ in (B.26), we have the conclusion as

$$\|X_{i_2} - v_{k_2}\| \leq \left( 1 + \frac{30\gamma}{\sigma_*} \max\left\{1, \frac{d_{\max}}{\sigma_*}\right\} \right) \beta, \qquad \text{for some } k_2 \neq k_1. \tag{B.42}$$

**Step 3: Analysis of the remaining iterations of SPA.**

Fix $3 \leq s \leq K - 1$. We now study the $s$th iteration. Let $i_1, \ldots, i_K$ denote the sequentially selected indices in SPA. We aim to show that there exist distinct $k_1, k_2, \ldots, k_s \in \{1, 2, \ldots, K\}$ such that

$$\|X_{i_s} - v_{k_s}\| \leq \left( 1 + \frac{30\gamma}{\sigma_*} \max\left\{1, \frac{d_{\max}}{\sigma_*}\right\} \right) \beta. \tag{B.43}$$

Let's denote $\mathcal{M}_{s-1} := \{k_1, \ldots, k_{s-1}\}$ for brevity. Suppose we have already shown (B.43) for every index $1, 2, \ldots, s-1$. Our goal is showing that (B.43) continues to hold for $s$ and some $k_s \notin \mathcal{M}_{s-1}$.

Let $X_i^{(1)} = X_i$ and $H_1$ be the same as in Step 1 of this proof. We define $X_i^{(s)}$ and $H_s$ recursively to describe the iterations in SPA:

$$\hat{y}_{s-1} = \frac{X_{i_{s-1}}^{(s-1)}}{\|X_{i_{s-1}}^{(s-1)}\|}, \qquad H_s = (I_d - \hat{y}_{s-1}\hat{y}_{s-1}')H_{s-1}, \qquad X_i^{(s)} = H_s X_i^{(s-1)}. \tag{B.44}$$

It is seen that $H_{s-1} = \prod_{m=1}^{s-1}(I_d - \hat{y}_m \hat{y}_m')$. Note that each $\hat{y}_m$ is orthogonal to $\hat{y}_1, \ldots, \hat{y}_{m-1}$. As a result, $H_{s-1}$ is a projection matrix with rank $(s-1)$. We apply Lemma C to obtain that

$$\sigma_{K-s}(H_{s-1}V) \geq \sigma_{K-1}(V) \geq \sigma_*, \qquad \text{for } 3 \leq s \leq K - 1. \tag{B.45}$$

Write $V^{(s-1)} = H_{s-1}V$ and $V^{(s)} = H_s V$. Using the notations in (B.44), we have

$$X_i^{(s)} = (I_d - \hat{y}_s \hat{y}_s')X_i^{(s-1)}, \qquad V^{(s)} = (I_d - \hat{y}_s \hat{y}_s')V^{(s-1)}.$$

Here, $\Gamma_s := I_d - \hat{y}_s \hat{y}_s'$ is a projection matrix. We observe:

The relationship between $(X_i^{(s-1)}, V^{(s-1)})$ and $(X_i^{(s)}, V^{(s)})$ is similar to the one between $(X_i, V)$ and $(\widetilde{X}_i, \widetilde{V})$ in Step 2, except that $H_1$ is replaced with $\Gamma_s$. $\qquad$ (B.46)

We aim to show that (B.35)-(B.38) still hold when those quantities are defined through $(X_i^{(s)}, V^{(s)})$. Recall that the proofs in Step 2 are inductive, where we actually showed that if (B.35)-(B.38) hold for the corresponding quantities defined through $(X_i, V)$, then they also hold for the same quantities defined through $(\widetilde{X}_i, \widetilde{V})$. Given (B.46), the same is true here.

It remains to develop a counterpart of Lemma F. The following lemma will be in Section B.4.7. It is also an inductive proof, relying on that (B.43) already holds for $1, 2, \ldots, s - 1$. .

**Lemma G.** *Under the conditions of Theorem A, write* $\sigma_* = \sigma_{K-1}(V)$. *Let* $\tilde{v}_k = V^{(s)} e_k$, $\tilde{d}_{\max} = \max_k \|\tilde{v}_k\|$, *and* $\widetilde{\mathcal{K}}(h_0) = \{k : \|\tilde{v}_k\| \geq \tilde{d}_{\max} - h_0\}$. *The following claims are true:*

$$\tilde{d}_{\max} \geq \sigma_*/2, \qquad \min_{\substack{\{k,\ell\} \cap \mathcal{M}_{s-1} = \emptyset, \\ k \neq \ell}} \|\tilde{v}_k - \tilde{v}_\ell\| \geq \sqrt{2}\sigma_*, \quad and \quad \mathcal{M}_{s-1} \cap \widetilde{\mathcal{K}}(h_0) = \emptyset. \qquad (B.47)$$

In Step 2, we have carefully shown how to use (B.35)-(B.39) to get (B.42). Using similar analyses, we can use the counterparts of (B.35)-(B.38), which are defined through $(X_i^{(s)}, V^{(s)})$, and the claim of Lemma G, to obtain (B.43). This completes the proof.

## B.4 PROOF OF THE SUPPLEMENTARY LEMMAS

### B.4.1 PROOF OF LEMMA A

By definition, $F(\pi) = \sum_{k=1}^{K} \pi(k) v_k$. Since $\sum_{k=1}^{K} \pi(k) = 1$, for any $v_0 \in \mathbb{R}^d$, we can re-express $F(\pi)$ as $F(\pi) = v_0 + \sum_{k=1}^{K} \pi(k)(v_k - v_0)$. It follows immediately that

$$\|F(\pi) - F(\tilde{\pi})\| = \left\| \sum_{k=1}^{K} [\pi(k) - \tilde{\pi}(k)](v_k - v_0) \right\| \leq \|\pi - \tilde{\pi}\|_1 \cdot \max_k \|v_k - v_0\|.$$

At the same time, since $\mathbf{1}_K'(\pi - \tilde{\pi}) = 0$, the vector $\pi - \tilde{\pi}$ is an $(K-1)$-dimensional linear subspace. It follows by basic properties of singular values that

$$\|F(\pi) - F(\tilde{\pi})\| = \|V(\pi - \tilde{\pi})\| \geq \sigma_{K-1}(V) \cdot \|\pi - \tilde{\pi}\|.$$

Combining the above gives (B.12).

Suppose there are $1 \leq k_1 < k_2 < \ldots < k_s \leq K$ such that $\pi(k_j) = \tilde{\pi}(k_j)$, for $1 \leq j \leq s$. Then, the vector $\delta = \pi - \tilde{\pi}$ satisfies $(s+1)$ constraints: $\mathbf{1}_K' \delta = 0$, $\delta(k_j) = 0$, for $1 \leq j \leq s$. In other words, $\delta$ lives in a $(K - 1 - s)$-dimensional linear space. It follows by properties of singular values that

$$\|F(\pi) - F(\tilde{\pi})\| = \|V(\pi - \tilde{\pi})\| \geq \sigma_{K-1-s}(V) \cdot \|\pi - \tilde{\pi}\|.$$

This proves (B.13).

### B.4.2 PROOF OF LEMMA B

Write for short $x = \sum_{i=1}^{m} \pi_i x_i \in \mathbb{R}^d$ and $L = \sum_{i=1}^{m} w_i \|x_i\|$. By the triangle inequality,

$$\|x\| \leq L.$$

In this lemma, we would like to get a lower bound for $L - \|x\|$. By definition,

$$\|x\|^2 = \sum_i w_i^2 \|x_i\|^2 + \sum_{i \neq j} w_i w_j x_i' x_j. \qquad (B.48)$$

For any vectors $u, v \in \mathbb{R}^d$, we have a universal equality: $2u'v = 2\|u\|\|v\| + (\|u\| - \|v\|)^2 - \|u - v\|^2$. By our assumption, $\|x_i - x_j\| \geq a$ and $(\|x_i\| - \|x_j\|)^2 \leq b^2$, for all $i \neq j$. It follows that

$$x_i' x_j \leq \|x_i\|\|x_j\| - (a^2 - b^2)/2, \qquad 1 \leq i \neq j \leq m. \qquad (B.49)$$

We plug (B.49) into (B.48) to get

$$\|x\|^2 \leq \sum_i w_i^2 \|x_i\|^2 + \sum_{i \neq j} w_i w_j \|x_i\| \|x_j\| - \frac{1}{2}(a^2 - b^2) \sum_{i \neq j} w_i w_j$$
$$= L^2 - \frac{1}{2}(a^2 - b^2) \sum_{i \neq j} w_i w_j. \tag{B.50}$$

Note that $\sum_{i \neq j} w_i w_j = \sum_i \sum_{j:i \neq j} w_j = \sum_i w_i(1 - w_i)$. Combining it with (B.50) gives

$$\|x\|^2 \leq L^2 - \frac{1}{2}(a^2 - b^2) \sum_i w_i(1 - w_i). \tag{B.51}$$

At the same time, $L + \|x\| \leq 2L$. It follows that

$$L - \|x\| = \frac{L^2 - \|x\|^2}{L + \|x\|} \geq \frac{L^2 - \|x\|^2}{2L} \geq \frac{a^2 - b^2}{4L} \sum_i w_i(1 - w_i). \tag{B.52}$$

This proves the claim.

### B.4.3 PROOF OF LEMMA C

Since $H$ is a projection matrix, there exists $Q_1 \in \mathbb{R}^s$ and $Q_2 \in \mathbb{R}^{d-s}$ such that $Q = [Q_1, Q_2]$ is an orthogonal matrix, $H = Q_1 Q_1'$, and $I_d - H = Q_2 Q_2'$. It follows that

$$(I_d - H)VV'(I_d - H) = Q_2(Q_2'VV'Q_2)Q_2'.$$

Since $Q_2$ has orthonormal columns, for any symmetric matrix $M \in \mathbb{R}^{(d-s) \times (d-s)}$, $M$ and $Q_2 M Q_2'$ have the same set of nonzero eigenvalues. Hence,

$$\sigma_{K-1-s}^2((I_d - H)V) = \lambda_{K-1-s}(Q_2'VV'Q_2).$$

We note that $Q_2'VV'Q_2 \in \mathbb{R}^{(d-s) \times (d-s)}$ is a principal submatrix of $Q'VV'Q \in \mathbb{R}^{d \times d}$. Using the eigenvalue interlacing theorem (Horn & Johnson, 1985, Theorem 4.3.28),

$$\lambda_{K-1-s}(Q_2'VV'Q_2) \geq \lambda_{K-1}(Q'VV'Q).$$

The claim follows immediately by noting that $\lambda_{K-1}(Q'VV'Q) = \lambda_{K-1}(VV') = \sigma_{K-1}^2(V)$.

### B.4.4 PROOF OF LEMMA D

Write $\ell_{\max} = \max_{1 \leq k \leq K} \|v_k\|$. We target to show

$$\ell_{\max}^2 \geq \frac{K - s - 1}{2(K - s)} \sigma_*^2, \qquad \text{with } \sigma_* := \sigma_{K-1-s}(V). \tag{B.53}$$

The right hand side of (B.53) is minimized at $s = K - 2$, at which $\ell_{\max}^2 \geq \sigma_*^2/4$. We now show (B.53). When $s = 0$, it is seen that

$$K\ell_{\max}^2 \geq \sum_k \|v_k\|^2 = \text{trace}(V'V) \geq (K - 1)\sigma_{K-1}^2(V).$$

Therefore, $\ell_{\max}^2 \geq \frac{K-1}{K}\sigma_*^2$, which implies (B.16) for $s = 0$. When $1 \leq s \leq K - 2$, since $\|v_k\| \leq \delta$ for at least $s$ of the vertices,

$$s\delta^2 + (K - s)\ell_{\max}^2 \geq \sum_k \|v_k\|^2 = \text{trace}(V'V) \geq (K - 1 - s)\sigma_{K-1-s}^2(V).$$

As a result, for $\sigma_* = \sigma_{K-1-s}(V)$,

$$\ell_{\max}^2 \geq \frac{(K - s - 1)\sigma_*^2 - s\delta^2}{K - s}. \tag{B.54}$$

Note that $\frac{s}{K-s-1}$ is a monotone increasing function of $s$. Hence, $\frac{s}{K-s-1} \leq K - 2$. The assumption of $2(K - 2)\delta^2 \leq \sigma_*^2$ implies that $\frac{2s}{K-s-1}\delta^2 \leq \sigma_*^2$, or equivalently, $s\delta^2 \leq \frac{K-s-1}{2}\sigma_*^2$. We plug it into (B.54) to get $\ell_{\max}^2 \geq \frac{K-s-1}{2(K-s)}\sigma_*^2$. This proves (B.16) for $1 \leq s \leq K - 2$.

### B.4.5 PROOF OF LEMMA E

Write $\mathcal{K} = \mathcal{K}(h_0)$, $\mathcal{V}_k = \mathcal{V}_k(\epsilon_0)$, and $\mathcal{V} = \mathcal{V}(\epsilon_0, h_0)$ for short. By definition of $\mathcal{K}$,

$$d_{\max} - h_0 \leq \|v_k\| \leq d_{\max}, \text{ for } k \in \mathcal{K}, \quad \|v_k\| \leq d_{\max} - h_0, \text{ for } k \notin \mathcal{K}. \tag{B.55}$$

We shall fix a point $x \in \mathcal{S} \setminus \mathcal{V}$ and derive an upper bound for $\|x\|$.

First, we need some preparation, let $F$ be the mapping in Lemma A. It follows that $\pi = F^{-1}(x)$ is the barycentric coordinate of $x$ in the simplex. By definition of $\mathcal{V}$,

$$\max_{k \in \mathcal{K}} \pi(k) \leq 1 - \epsilon_0, \qquad \text{whenever } x := F(\pi) \text{ is in } \mathcal{S} \setminus \mathcal{V}. \tag{B.56}$$

The $K$ vertices are naturally divided into two groups: those in $\mathcal{K}$ and those not in $\mathcal{K}$. Define

$$\rho := \sum_{k \in \mathcal{K}} \pi(k), \qquad \eta := \begin{cases} \rho^{-1} \sum_{k \in \mathcal{K}} \pi(k) v_k, & \text{if } \rho \neq 0, \\ \mathbf{0}_d, & \text{otherwise.} \end{cases} \tag{B.57}$$

Here, $\rho$ is the total weight $\pi$ puts on those vertices in $\mathcal{K}$, and we can re-write $x$ as

$$x = \rho \eta + \sum_{k \notin \mathcal{K}} \pi(k) v_k.$$

By the triangle inequality,

$$\|x\| = \left\| \rho \eta + \sum_{k \notin \mathcal{K}} \pi(k) v_k \right\| \leq \rho \|\eta\| + \sum_{k \notin \mathcal{K}} \pi(k) \|v_k\|$$
$$\leq \rho \|\eta\| + (1 - \rho)(d_{\max} - h_0). \tag{B.58}$$

Next, we proceed with showing the claim. We consider two cases:

$$1 - \rho \geq \epsilon_0/2 \text{ (Case 1)}, \qquad \text{and} \qquad 1 - \rho < \epsilon_0/2 \text{ (Case 2)}.$$

In Case 1, the total weight that $\pi_i$ puts on those vertices not in $\mathcal{K}$ is at least $\epsilon_0/2$. Since each vertex satisfies that $\|v_k\| \leq d_{\max} - h_0$ (see (B.56)) and $\|\eta\| \leq d_{\max}$, it follows from (B.58) that

$$\|x\| \leq d_{\max} - (1 - \rho) h_0 \leq d_{\max} - \frac{h_0 \epsilon_0}{2}, \qquad \text{in Case 1.} \tag{B.59}$$

In Case 2, if $\mathcal{K} = \{k^*\}$ is a singleton, then $\rho = \pi(k^*)$. By (B.56), $\pi(k^*) \leq 1 - \epsilon_0$, which leads to $1 - \rho = 1 - \pi(k^*) \geq \epsilon_0$. This yields a contradiction to $1 - \rho < \epsilon_0/2$. Hence, it must hold that

$$|\mathcal{K}| \geq 2. \tag{B.60}$$

Now, $\eta$ is a convex combination of more than one point in $\{v_k : k \in \mathcal{K}\}$, for which we hope to apply Lemma B. By (B.55), for each $k \in \mathcal{K}$, $\|v_k\|$ is in the interval $[d_{\max} - h_0, d_{\max}]$. Hence, we can take $b = h_0$ in Lemma B. In addition, from the assumption (B.22), $\|v_k - v_\ell\| \geq \sqrt{2}\sigma_*$ for any $k \neq \ell$. Hence, we set $a = \sqrt{2}\sigma_*$ in Lemma B. We apply this lemma to the vector $\eta$ in (B.57). It yields

$$\|\eta\| \leq L - \frac{(2\sigma_*^2 - h_0^2)}{4L} \sum_{k \in \mathcal{K}} \frac{\pi(k)[\rho - \pi(k)]}{\rho^2}, \qquad \text{with} \quad L := \sum_{k \in \mathcal{K}} \frac{\pi(k)}{\rho} \|v_k\|. \tag{B.61}$$

Since $L \leq d_{\max}$, it follows from (B.61) that

$$\|\eta\| \leq d_{\max} - \frac{2\sigma_*^2 - h_0^2}{4\rho d_{\max}} \sum_{k \in \mathcal{K}} \pi(k)[1 - \rho^{-1}\pi(k)].$$

Additionally, noticing that $\pi(k) \leq 1 - \epsilon_0$ for each $k \in \mathcal{K}$, we have the following inequality:

$$1 - \rho^{-1}\pi(k) = \rho^{-1}[1 - \pi(k)] - \rho^{-1}(1 - \rho) \geq \rho^{-1}[\epsilon_0 - (1 - \rho)].$$

Combining these arguments and using the fact that $\sum_{k \in \mathcal{K}} \pi(k) = \rho$, we have

$$\|\eta\| \leq d_{\max} - \frac{(2\sigma_*^2 - h_0^2)[\epsilon_0 - (1 - \rho)]}{4\rho^2 d_{\max}} \sum_{k \in \mathcal{K}} \pi(k)$$
$$\leq d_{\max} - \frac{(2\sigma_*^2 - h_0^2)[\epsilon_0 - (1 - \rho)]}{4\rho d_{\max}}. \tag{B.62}$$

Since $1 - \rho \leq \epsilon_0/2$, we immediately have $\|\eta\| \leq d_{\max} - \frac{2\sigma_*^2 - h_0^2}{8\rho d_{\max}}$. We plug it into (B.58) to get

$$
\begin{aligned}
\|x\| &\leq \rho\Big(d_{\max} - \frac{2\sigma_*^2 - h_0^2}{8\rho d_{\max}}\Big) + (1 - \rho)(d_{\max} - h_0) \\
&\leq \rho\Big(d_{\max} - \frac{2\sigma_*^2 - h_0^2}{8\rho d_{\max}}\Big) + (1 - \rho)d_{\max} \\
&\leq d_{\max} - \frac{(2\sigma_*^2 - h_0^2)\epsilon_0}{8d_{\max}}, \qquad \text{in Case 2.}
\end{aligned} \tag{B.63}
$$

We now combine (B.59) for Case 1 and (B.63) for Case 2. By setting $h_0 = \sigma_*/3$, we have a unified expression:

$$
\|x\| \leq d_{\max} - \min\Big\{\frac{\sigma_*}{6}, \; \frac{2\sigma_*^2}{9d_{\max}}\Big\}\epsilon_0.
$$

Consequently, a sufficient condition for $\|x\| \leq d_{\max} - t$ to hold is

$$
\min\Big\{\frac{\sigma_*}{6}, \; \frac{\sigma_*^2}{6d_{\max}}\Big\}\epsilon_0 \leq t \qquad \Longleftrightarrow \qquad \epsilon_0 \geq \frac{6}{\sigma^*}\max\Big\{1, \frac{d_{\max}}{\sigma_*}\Big\}t.
$$

This proves the claim.

### B.4.6 PROOF OF LEMMA F

Without loss of generality, we assume $k_1 = 1$.

By definition, $\widetilde{V} = H_1 V$, where $H_1$ is a rank-1 projection matrix. It follows by Lemma C that

$$
\sigma_{K-2}(\widetilde{V}) \geq \sigma_{K-1}(V) = \sigma_*. \tag{B.64}
$$

Note that $\tilde{d}_{\max} \geq \max_{k \neq 1} \|\tilde{v}_k\|$ and $\|\tilde{v}_1\| = 0$. We apply Lemma D with $s = 1$ and $\delta = 0$ to get

$$
\tilde{d}_{\max} \geq \frac{1}{2}\sigma_{K-2}(\widetilde{V}) \geq \frac{1}{2}\sigma_*.
$$

This proves the first claim in (B.39). Note that $\tilde{v}_k = \widetilde{V}e_k$, where $e_k \in \mathbb{R}^K$ is a standard basis vector. For any $2 \leq k \neq \ell \leq K$, $e_k$ and $e_\ell$ both have a zero at the first coordinate; and we apply Lemma A with $s = 1$ to get

$$
\|v_k - v_\ell\| \geq \sigma_{K-2}(\widetilde{V})\|e_k - e_\ell\| \geq \sqrt{2}\sigma_*.
$$

This proves the second claim in (B.39).

Finally, we show the third claim. Note that

$$
\tilde{v}_1 = H_1 v_1 = v_1 - \frac{v_1' X_{i_1}}{\|X_{i_1}\|^2}X_{i_1} = \frac{X_{i_1}'(X_{i_1} - v_1)}{\|X_{i_1}\|^2}v_1 - \frac{v_1' X_{i_1}}{\|X_{i_1}\|^2}(X_{i_1} - v_1). \tag{B.65}
$$

Here, $\|v_1\| \leq d_{\max}$, and by (B.28), $\|X_{i_1}\| \geq d_{\max} - \beta$. Since $|X_{i_1}'(X_{i_1} - v_1)| \leq \|X_{i_1}\| \cdot \|X_{i_1} - v_1\|$, we have

$$
\frac{|X_{i_1}'(X_{i_1} - v_1)|}{\|X_{i_1}\|^2}\|v_1\| \leq \frac{\|v_1\|}{\|X_{i_1}\|}\|X_{i_1} - v_1\| \leq \frac{d_{\max}}{d_{\max} - \beta}\|X_{i_1} - v_1\|,
$$

and

$$
\frac{v_1' X_{i_1}}{\|X_{i_1}\|^2} \leq \frac{\|v_1\|}{\|X_{i_1}\|} \leq \frac{d_{\max}}{d_{\max} - \beta}.
$$

Plugging these inequalities into (B.65) and applying (B.34), we obtain:

$$
\begin{aligned}
\|\tilde{v}_1\| &\leq \frac{2d_{\max}}{d_{\max} - \beta}\|X_{i_1} - r_{i_1}\| \\
&\leq \frac{2d_{\max}}{d_{\max} - \beta}\Big(\beta + \frac{30\gamma}{\sigma_*}\max\{1, \frac{d_{\max}}{\sigma_*}\}\beta\Big). \tag{B.66}
\end{aligned}
$$

By our assumption, $\frac{30 d_{\max}}{\sigma_*} \max\{1, \frac{d_{\max}}{\sigma_*}\}\beta \leq \sigma_*/15$. Moreover, we have shown $d_{\max} \geq \tilde{d}_{\max} \geq \sigma_*/2$. It further implies $\beta \leq \frac{\sigma_*^2}{450 d_{\max}} \leq \frac{1}{225}\sigma_* \leq \frac{1}{100}\tilde{d}_{\max}$. As a result,

$$\|\tilde{v}_1\| \leq \frac{200}{99}(\beta + \frac{\sigma_*}{15}) \leq \frac{3}{10}\tilde{d}_{\max} \leq \tilde{d}_{\max} - \frac{7}{20}\sigma_*. \tag{B.67}$$

At the same time, $h_0 = \sigma_*/3$. Hence,

$$\|\tilde{v}_1\| < \tilde{d}_{\max} - h_0 \qquad \Longrightarrow \qquad 1 \notin \widetilde{\mathcal{K}}(h_0).$$

This proves the third claim in (B.39).

### B.4.7 PROOF OF LEMMA G

Suppose we have already obtained (B.47) and (B.43) for each $1 \leq j \leq s-1$, and we would like to show (B.47) for $s$.

First, consider the second claim in (B.47). For each $k \notin \mathcal{M}_{s-1}$, it has $(s-1)$ zeros in its barycentric coordinate (corresponding to those indices in $\mathcal{M}_{s-1}$). We apply Lemma A to obtain:

$$\|\tilde{v}_k - \tilde{v}_\ell\| \geq \sqrt{2}\sigma_{K-s}(\widetilde{V}) \geq \sqrt{2}\sigma_*, \qquad \text{for all } k \neq \ell \text{ in } \{1, \ldots, K\} \setminus \mathcal{M}_{s-1},$$

where the first inequality is from (B.13) and the second inequality is from (B.45).

Next, consider the third claim in (B.47). Note that $\mathcal{M}_{s-1} = \{k_1, k_2, \ldots, k_{s-1}\}$. For each $1 \leq j \leq s-1$, by definition, $\tilde{v}_{k_j} = \left[\prod_{m \geq j}(I_d - \hat{y}_m \hat{y}_m')\right] \cdot (I_d - \hat{y}_j \hat{y}_j) H_{j-1} v_{k_j}$. It follows that

$$\|\tilde{v}_{k_j}\| \leq \|(I_d - \hat{y}_j \hat{y}_j) H_{j-1} v_{k_j}\|, \qquad \text{where} \quad \hat{y}_j = \frac{H_{j-1}X_{i_j}}{\|H_{j-1}X_{i_j}\|}. \tag{B.68}$$

Here, $\|H_{j-1}X_{i_j}\|$ is the maximum Euclidean distance attained in the $(j-1)$th iteration. Since we have already established (B.47) for $j$, we immediately have

$$\|H_{j-1}X_{i_j}\| \geq \sigma_*/2, \qquad \text{for } 1 \leq j \leq s-1.$$

In addition, we have shown (B.42) for $1 \leq j \leq s-1$, which implies that

$$\|H_{j-1}X_{i_j} - H_{j-1}v_{k_j}\| \leq \left(1 + \frac{30\gamma}{\sigma_*}\max\{1, \frac{d_{\max}}{\sigma_*}\}\right)\beta.$$

Using the above ineqaulities, we can mimic the proof of (B.66) to show that

$$\|(I_d - \hat{y}_j \hat{y}_j) H_{j-1} v_{k_j}\| \leq \left(1 + \frac{30\gamma}{\sigma_*}\max\{1, \frac{d_{\max}}{\sigma_*}\}\right)\beta. \tag{B.69}$$

Write $\Gamma_j = I_d - \hat{y}_j \hat{y}_j'$. It is seen that

$$\|\tilde{v}_{k_j}\| = \left\|\prod_{\ell=j+1}^{s}\Gamma_j H_{j-1}v_{k_j}\right\| \leq \|\Gamma_j H_{j-1}v_{k_j}\| \leq \|(I_d - \hat{y}_j \hat{y}_j) H_{j-1}v_{k_j}\|.$$

Therefore, for $1 \leq j \leq s-1$,

$$\|\tilde{v}_{k_j}\| \leq \left(1 + \frac{30\gamma}{\sigma_*}\max\{1, \frac{d_{\max}}{\sigma_*}\}\right)\beta. \tag{B.70}$$

We further mimic the argument in (B.67) to obtain:

$$\|\tilde{v}_{k_j}\| \leq \tilde{\beta}_{\max} - 7\sigma_*/20 < \tilde{\beta} - h_0, \qquad \text{for all } 1 \leq j \leq s-1.$$

This implies that

$$k_j \notin \widetilde{\mathcal{K}}(h_0) \text{ for } 1 \leq j \leq s-1 \quad \Longrightarrow \quad \mathcal{M}_{s-1} \cap \widetilde{\mathcal{K}}(h_0) = \emptyset. \tag{B.71}$$

Last, consider the first claim in (B.47). Let $\Delta$ denote the right hand side of (B.70) for brevity. We have shown $\|\tilde{v}_k\| \leq \Delta$, for all $k \in \mathcal{M}_{s-1}$. By our assumption, we can easily conclude that $\sigma_*^2 \geq 2(K-2)\Delta$. We then apply Lemma D with $s-1$ and $\delta = \Delta$ to get

$$\tilde{d}_{\max} \geq \frac{1}{2}\sigma_{K-s}(\widetilde{V}) \geq \sigma_*/2, \tag{B.72}$$

where the last inequality is from (B.45).

## C  PROOF OF THE MAIN THEOREMS

We recall our pp-SPA procedure. On the hyperplane, we obtained the projected points

$$\tilde{X}_i := H(X_i - \bar{X}) + \bar{X} = (I_d - H)\bar{X} + Hr_i + H\epsilon_i$$

after rotation by $U$, they become $Y_i = U'\tilde{X}_i = U'r_i + U'\epsilon_i = U'X_i \in \mathbb{R}^{K-1}$. Denote $\tilde{Y}_i = U_0'X_i = U_0'r_i + U_0'\epsilon_i \in \mathbb{R}^{K-1}$. In particular, $U_0'\epsilon_i \sim N(0, \sigma^2 I_{K-1})$. Then, without loss of generality, the vertex hunting analysis on $\tilde{Y}_i$ is equivalent to that of $X_i = r_i + \epsilon_i \in \mathbb{R}^p$, where $\epsilon_i \sim N(0, \sigma^2 I_p)$ with $p = K - 1$. We provide the following theorems for the rate by applying D-SPA on the aforementioned low dimension $p = K - 1$ space. The proof of these two theorems are postponed to Section C.2.

**Theorem B.** *Consider $X_i = r_i + \epsilon_i \in \mathbb{R}^p$, where $\epsilon_i \sim N(0, \sigma^2 I_p)$ for $1 \le i \le n$. Suppose $m \ge c_1 n$ for a constant $c_1 > 0$ and $p \ll \log(n)/\log\log(n)$. Let $p/\log(n) \ll \delta_n \ll 1$. Let $c_2^* = 0.9(2e^2)^{-1/p}\sqrt{(2/p)}(\Gamma(p/2+1))^{1/p}$. Then, $c_2^* \to 0.9e^{-1/2}$ as $p \to \infty$. . We apply D-SPA to $X_1, X_2, \ldots, X_n$ and output $X_1^*, \cdots, X_n^*$ where some $X_i^*$ may be NA owing to the pruning. If we choose $N = \log(n)$ and*

$$\Delta = c_3\sigma\sqrt{p}\left(\frac{\log(n)}{n^{1-\delta_n}}\right)^{1/p} \text{for a constant } c_3 \le c_2^*,$$

*Then,*

$$\beta_{new}(X^*) \le \sqrt{\delta_n} \cdot \sigma \cdot \sqrt{2\log(n)}$$

*If the last inequality of (8 ) and (9 ) hold, then up to a permutation in the columns,*

$$\max_{1 \le k \le K} \|\hat{v}_k - v_k\| \le g_{new}(V) \cdot \sqrt{\delta_n} \cdot \sigma \cdot \sqrt{2\log(n)}.$$

The second theorem discuss the case there a fewer pure nodes.

**Theorem C.** *Consider $X_i = r_i + \epsilon_i \in \mathbb{R}^p$, where $\epsilon_i \sim N(0, \sigma^2 I_p)$ for $1 \le i \le n$. Fix $0 < c_0 < 1$ and assume that $m \ge n^{1-c_0+\delta}$ for a sufficiently small constant $0 < \delta < c_0$. Suppose $p \ll \log(n)/\log\log(n)$. Let $c_2^* = 0.9(2e^{2-c_0})^{-1/p}\sqrt{(2/p)}(\Gamma(p/2+1))^{1/p}$. Then $c_2^* \to 0.9e^{-1/2}$ as $p \to \infty$. Suppose we apply D-SPA to $X_1, X_2, \ldots, X_n$ and output $X_1^*, \cdots, X_n^*$ where some $X_i^*$ may be NA owing to the pruning. If we choose $N = \log(n)$ and*

$$\Delta = c_3\sigma\sqrt{p}\left(\frac{\log(n)}{n^{1-c_0}}\right)^{1/p} \text{for a constant } c_3 \le c_2^*.$$

*Then,*

$$\beta_{new}(X^*) \le \sqrt{c_0} \cdot \sigma \cdot \sqrt{2\log(n)}$$

*If the last inequality of (8 ) and (9 ) hold, then up to a permutation in the columns,*

$$\max_{1 \le k \le K} \|\hat{v}_k - v_k\| \le g_{new}(V) \cdot \sqrt{c_0} \cdot \sigma\sqrt{2\log(n)}.$$

*for any arbitrary small constant $\delta < 0$.*

Based on the above two theorem, we have the results on $\{\tilde{Y}_i\}'s$. However, what we really care about is on $\{Y_i\}'s$ which differ from $\{\tilde{Y}_i\}'s$ by the rotation matrix. To bridge the gap, we need the following Lemma.

**Lemma H.** *Suppose that $s_{K-1}^2(R) \gg \max\{\sqrt{\sigma^2 d/n}, \sigma^2 d/n\}$ and $\sigma = O(1)$. Then, with probability $1 - o(1)$,*

$$\|U - U_0\| \asymp \|H - H_0\| \le \frac{C}{s_{K-1}^2(R)}\max\{\sqrt{\sigma^2 d/n}, \sigma^2 d/n\} \tag{C.73}$$

## C.1 PROOF OF THEOREMS 2 AND 3

With the help of Theorems B, C and Lemma H, we now prove Theorems 2 and 3. We will present the detailed proof for Theorem 3. The proof of Theorem 2 is nearly identical to that of Theorem 3 with the only difference in employing Theorem B, and we refrain ourselves from repeated details.

*Proof of Theorem 3.* Recall that $Y_i = U'X_i = U'r_i + U'\epsilon_i$ and $\tilde{Y}_i = U_0'r_i + U_0'\epsilon_i$. Theorem C indicates that applying D-SPA on $\bar{Y}_i$ improves the rate to $\sigma(1+o(1))\sqrt{2c_0 \log(n)}$. Note that $\|r_i\| \leq 1$. Also, by Lemma 5, $\|\epsilon_i\| \leq (1 + o(1))\sigma(\sqrt{\max\{d, 2\log(n)\}})$ simultaneously for all $i$, with high probability. Under the assumption $\alpha_n = o(1)$ for both cases and $s_{K-1}^2(R) \asymp s_{K-1}^2(\tilde{V})$ by Lemma 4, the first condition in Lemma H is valid. By the last inequality in (8), we have the norm of $r_i$ should be upper bounded for all $1 \leq i \leq n$ and therefore $s_{K-1}(\tilde{V}) \leq C \max_{k \neq l} \|\tilde{v}_k - \tilde{v}_\ell\| \leq C$. Further with the condition (9), we obtain that $\sigma = O(1)$. Therefore, the conditions in Lemma H are both valid. Then by employing Lemma H, we can derive that

$$\|Y_i - \tilde{Y}_i\| = O_{\mathbb{P}}\left(\frac{\sigma\sqrt{d}}{\sqrt{n}s_{K-1}^2(R)}(1 + \sigma\sqrt{\max\{d, 2\log(n)\}})\right) = O_{\mathbb{P}}(\sigma\alpha_n)$$

where the last step is due to Lemma 4 under the condition (8).

Consider the first case that $\alpha_n \ll t_n^*$. We choose $\Delta = c_3 t_n^* \sigma$. It is seen that $\sigma\alpha_n \ll \Delta$. We will prove by contradiction that applying pp-SPA with $(\Delta, \log(n))$ on $\{Y_i\}$, the denoise step can remove outlying points whose distance to the underlying simplex larger than $\sigma[\sqrt{2c_0 \log(n)} + C\alpha_n]$ for some $C > 0$.

First, suppose that with probability $c$ for a small constant $c > 0$, there is one point $Y_{i_0}$ away from the underlying simplex by a distance larger than $\sigma[\sqrt{2c_0 \log(n)} + C\alpha_n]$ and it is not pruned out. Since $\sigma\alpha_n \ll \Delta$, we see that $\tilde{Y}_{i_0}$ is faraway to the simplex with distance $\sigma\sqrt{2c_0 \log(n)}$ for certain large $C$ and it cannot be pruned out by $(1.5\Delta, \log(n))$. Otherwise if it can be pruned out, $\mathcal{B}(Y_{i_0}, \Delta) \subset \mathcal{B}(\tilde{Y}_{i_0}, 1.5\Delta)$ and hence $N(\mathcal{B}(Y_{i_0}, \Delta)) \geq \log(n)$, which means that we can prune out $Y_{i_0}$ with $(\Delta, \log(n))$. This is a contradiction. However, by employing Theorem C on $\{\tilde{Y}_i\}$ with $p = K - 1$ and noticing $c_2^* = 1.8c_2$ with $c_2$ defined in the manuscript, we should be able to prune out $\tilde{Y}_{i_0}$ with high proability. This leads to a contradiction.

Second, suppose that with probability $c$ for a small constant $c > 0$, all outliers can be removed but a vertex $v_1$ is also removed (which means all points near it are removed). Then, $N(\mathcal{B}(v_1, \Delta)) < \log(n)$. For the corresponding vertex for $\{\tilde{Y}_i\}$, denoted by $\tilde{v}_1$, it holds that $N(\mathcal{B}(\tilde{v}_i, \Delta/2)) < \log(n)$ which means the vertex $\tilde{v}_1$ for $\{\tilde{Y}_i\}$ is also pruned. However, again by Theorem C, this can only happen with probability $o(1)$. This leads to another contradiction.

Let us denote by $\beta(Y^*, U_0'V)$ the maximal distance of points in $Y^*$ to the simplex formed by $U_0'V$. By the above two contradictions, we conclude that with high probability,

$$\beta(Y^*, U_0'V) \leq \sigma[\sqrt{2c_0 \log(n)} + C\alpha_n].$$

where $U_0'V$ is the underlying simplex of $\{\tilde{Y}_i\}$. It is worth noting that $\alpha_n = o(1)$. Then, under the assumptions of the theorem, we can apply Theorem A (Theorem 1 in the manuscript). It gives that

$$\max_{1 \leq k \leq K} \|\hat{v}_k^* - U_0'v_k\| \leq \sigma g_{new}(V)[\sqrt{2c_0 \log(n)} + C\alpha_n]$$

where we use $(\hat{v}_1^*, \cdots, \hat{v}_K^*)$ to denote the output vertices by applying SP on $\{Y_i\}$. Eventually, we output each vertex $\hat{v}_k = (I_K - UU')\bar{X} + U\hat{v}_k^*$. It follows that up to a permutation of the $K$ vectors,

$$\max_{1 \leq k \leq K} \|\hat{v}_k - v_k\| \leq \max_{1 \leq k \leq K} \|U\hat{v}_k^* - v_k\| + \|(I_d - UU')\bar{X} - (I_d - U_0U_0')\bar{r}\|$$

$$\leq \max_{1 \leq k \leq K} \|\hat{v}_k^* - U_0'v_k\| + \|U - U_0\| + \|(I_d - UU')\bar{X} - (I_d - U_0U_0')\bar{r}\|$$

Further we can derive

$$\|(I_d - UU')\bar{X} - (I_d - U_0 U_0')\bar{r}\| \leq \|H - H_0\| + \|\bar{X} - \bar{r}\|$$
$$\leq \sigma\alpha_n + \|\bar{\epsilon}\|$$
$$\leq \sigma\alpha_n + \frac{2\sigma\sqrt{\max\{d, 2\log(n)\}}}{\sqrt{n}}$$

this together with Lemma H, give rise to

$$\max_{1 \leq k \leq K} \|\hat{v}_k - v_k\| \leq \sigma g_{new}(V)[\sqrt{2c_0 \log(n)} + C\alpha_n] + \frac{2\sigma\sqrt{\max\{d, 2\log(n)\}}}{\sqrt{n}} .$$

Consider the second case that $\alpha_n \gg t_n^*$ where we choose $\Delta = \sigma\alpha_n$. By Lemma 5, it is observed that with high probability, $\max_{1 \leq i \leq n} d(\tilde{Y}_i, \mathcal{S}) < (1 + o(1))\sigma\sqrt{2\log(n)}$. Notice that $\|Y_i - \tilde{Y}_i\| \leq C\sigma\alpha_n$ with high probability. For $\tilde{Y}_i$, if its distance to the underlying simplex is larger than $\sigma[(1 + o(1))\sqrt{2\log(n)} + C_1\alpha_n]$ for a sufficiently large $C_1 > 3C + 1$, then $d(\tilde{Y}_i, \mathcal{S}) \geq d(Y_i, \mathcal{S}) - C\sigma\alpha_n > \sigma[(1 + o(1))\sqrt{2\log(n)} + (2C + 1)\alpha_n]$. Hence, $\mathbb{B}(\tilde{Y}_i, (2C + 1)\Delta))$ is away from the simplex by a distance larger than $\sigma(1 + o(1))\sqrt{2\log(n)}$. It follows that $N(\mathbb{B}(Y_i, \Delta)) \leq N(\mathbb{B}(\tilde{Y}_i, (2C + 1)\Delta)) < \log(n)$. This is equivalent to say that we prune out the points there. Consequently, with high probability,

$$\beta(Y^*, U_0'V) \leq \sigma[(1 + o_{\mathbb{P}}(1))\sqrt{2\log(n)} + C_1\alpha_n]$$

and further by Theorem A (Theorem 1 in the manuscript),

$$\max_{1 \leq k \leq K} \|\hat{v}_k^* - U_0' v_k\| \leq \sigma g_{new}(V)[\sqrt{2\log(n)} + C_1\alpha_n]$$

Next, replicate the proof for $\max_{1 \leq k \leq K} \|\hat{v}_k - v_k\|$ in the former case, we can conclude that

$$\max_{1 \leq k \leq K} \|\hat{v}_k - v_k\| \leq \sigma g_{new}(V)[(1 + o_{\mathbb{P}}(1))\sqrt{2\log(n)} + C_1\alpha_n] + \frac{2\sigma\sqrt{\max\{d, 2\log(n)\}}}{\sqrt{n}}$$
$$= \sigma g_{new}(V)(1 + o_{\mathbb{P}}(1))\sqrt{2\log(n)}.$$

This concludes our proof.

$\square$

## C.2 PROOF OF THEOREMS B AND C.

In the subsection, we provide the proofs of Theorems B and C. We show the proof of Theorem C in detail and briefly present the proof of Theorems B as it is similar to that of Theorem C.

*Proof of Theorem C.* We first claim the limit of $c_2^* = 0.9(2e^{2-c_0})^{-1/p}\sqrt{(2/p)}(\Gamma(p/2 + 1))^{1/p}$. Note that $\Gamma(p/2 + 1) = (p/2)!$ if $p$ is even and $\Gamma(p/2 + 1) = \sqrt{\pi}(p + 1)!/(2^{p+1}(\frac{p+1}{2})!)$ if $p$ is odd. Using Stirling's approximation, it is elementary to deduce that

$$c_2^* = e^{O(1/p) - (1 - \log(p+1))(p+1)/2p - \log(p)/2} \to e^{-1/2}.$$

Define the radius $\Delta \equiv \Delta_n = c_3\sigma\sqrt{p}\left(\frac{\log(n)}{n^{1-c_0}}\right)^{1/p}$ for a constant $c_3 \leq c_2$. In the sequel, we will prove that applying D-SPA to $X_1, \cdots, X_n$ with $(\Delta, N)$, we can prune out the points whose distance to the underlying true simplex are larger than the rate in the theorem, while the points around vertices are captured.

Denote $d(x, \mathcal{S})$, the distance of $x$ to the simplex $\mathcal{S}$. Let

$$\mathcal{R}_f := \{x \in \mathbb{R}^p : d(x, \mathcal{S}) \geq 2\sigma\sqrt{\log(n)}\}$$

We first claim that the number of points in $\mathcal{R}_f$, denoted by $N(\mathcal{R}_f)$, is bounded with probability $1 - o(1)$. By definition, we deduce

$$N(\mathcal{R}_f) = \sum_{i=1}^{n} \mathbf{1}(x_i \in \mathcal{R}_f) \leq \sum_{i=1}^{n} \mathbf{1}(\|\varepsilon_i\| \geq 2\sigma\sqrt{\log n})$$

The mean on the RHS is given by $n\mathbb{P}(\|\varepsilon_i\| \geq 2\sigma\sqrt{\log n}) = n\mathbb{P}(\chi_p^2 \geq 4\log n) \leq ne^{-1.5\log(n)} = n^{-1/2}$. By similar computations, the order of the variance is again $n^{-1/2}$. By Chebyshev's inequality, we conclude that $N(\mathcal{R}_f) = o_{\mathbb{P}}(1)$.

In the sequel, we use the notation $\mathbb{B}(x, r)$ to represent a ball centered at $x$ with radius $r$ and denote $N(\mathbb{B}(x, r))$ the number of points falling into this ball. And we also denote $\mathcal{S}$ the true underlying simplex.

Based on these notation, we introduce

$$P := \mathbb{P}(\exists\ X_i \text{ satisfying } \sigma\sqrt{2c_0\log(n)} \leq d(X_i, \mathcal{S}) \leq 2\sigma\sqrt{\log(n)} \text{ cannot be pruned out })$$

We aim to show that $P = o(1)$. To see this, we first derive

$$P = \binom{n}{N} N \cdot \mathbb{P}(X_1, \cdots X_N \in B(X_1, \Delta) \text{ s.t. } \sigma\sqrt{2c_0\log(n)} \leq d(X_1, \mathcal{S}) \leq 2\sigma\sqrt{\log(n)})$$

$$\leq \binom{n}{N} N \cdot \int_{a_n \leq d(x, \mathcal{S}) \leq b_n} f_{X_1}(x)\mathbb{P}(X_2, \cdots, X_N \in \mathcal{B}(x, \Delta))\mathrm{d}x$$

$$\leq \binom{n}{N} N \cdot \int_{a_n \leq d(x, \mathcal{S}) \leq b_n} f_{X_1}(x)\prod_{t=2}^{N}\mathbb{P}(X_t \in \mathcal{B}(x, \Delta))\mathrm{d}x$$

where $a_n := \sigma\sqrt{2c_0\log(n)}$ and $b_n := 2\sigma\sqrt{\log(n)}$ for simplicity. We can compute that for any $2 \leq t \leq N$,

$$\mathbb{P}(X_t \in \mathcal{B}(x, \Delta)) = (2\pi\sigma^2)^{-\frac{p}{2}}\int_{\|y-x\| \leq \Delta}\exp\{-\|y - r_t\|^2/2\sigma^2\}\mathrm{d}y$$

$$\leq \frac{(\Delta/\sigma)^p}{2^{p/2}\Gamma(p/2+1)}\exp\left\{-\frac{(\|x - r_t\| - \Delta)^2}{2\sigma^2}\right\}$$

$$\leq (\Delta/\sigma)^p C_p \exp\left\{-\frac{\|x - r_t\|^2}{2(1+\tau_n)\sigma^2}\right\} \tag{C.74}$$

where $\tau_n := C\Delta/\sigma\sqrt{2c_0\log(n)}$ for a large $C > 0$;and we write $C_p := 2^{1-p/2}/\Gamma(p/2+1)$. Here to obtain the last inequality, we used the definition of $\Delta$ and the derivation

$$\frac{\Delta}{\|x - r_t\|} \leq \frac{\Delta}{\sigma\sqrt{2c_0\log(n)}} \leq C\tau_n \leq C\sqrt{p}(\log(n))^{1/p-1/2}/n^{(1-c_0)/p} = o(1)$$

so that

$$(1 - \Delta/\|x - r_t\|)^2 \leq (1+\tau_n)^{-1}$$

by choosing appropriate $C$ in the definition of $\tau_n$. Further, under the condition that $p \ll \log(n)/\log\log(n)$, one can verify that

$$\tau_n \ll 1/\log(n) = o(1).$$

(C.74), together with

$$f_{X_1}(x) = (2\pi\sigma^2)^{-\frac{p}{2}}\exp\{-\|x - r_1\|^2/(2\sigma^2)\} \leq (2\pi\sigma^2)^{-\frac{p}{2}}\exp\{-\|x - r_1\|^2/(2(1+\tau_n)\sigma^2)\},$$

leads to

$$P \leq \binom{n}{N}NC_p^{N-1}(\Delta/\sigma)^{p(k-1)} \cdot \int_{a_n \leq d(x, \mathcal{S}) \leq b_n}(2\pi\sigma^2)^{-\frac{p}{2}}\exp\left\{-\frac{\sum_{t=1}^{N}\|x - r_t\|^2}{2(1+\tau_n)\sigma^2}\right\}\mathrm{d}x$$

Also, notice that $\sum_{t=1}^{N}\|x - r_t\|^2 \geq N\|x - \bar{r}\|^2$ where $\bar{r} = N^{-1}\sum_{t=1}^{N}r_t$. Then,

$$P \leq \binom{n}{N}NC_p^{N-1}(\Delta/\sigma)^{p(N-1)} \cdot \int_{a_n \leq d(x, \mathcal{S}) \leq b_n}(2\pi\sigma^2)^{-\frac{p}{2}}\exp\left\{-\frac{N\|x - \bar{r}\|^2}{2(1+\tau_n)\sigma^2}\right\}\mathrm{d}x$$

$$\leq \binom{n}{N}NC_p^{N-1}(\Delta/\sigma)^{p(N-1)}\int_{\|x-\bar{r}\| \geq a_n}(2\pi\sigma^2)^{-\frac{p}{2}}\exp\left\{-\frac{N\|x - \bar{r}\|^2}{2(1+\tau_n)\sigma^2}\right\}\mathrm{d}x$$

$$\leq \binom{n}{N}NC_p^{N-1}(\Delta/\sigma)^{p(N-1)}N^{-p/2}(1+\tau_n)^{p/2} \cdot \mathbb{P}(\chi_p^2 \geq 2Nc_0\log n/(1+\tau_n))$$

where we used the fact that $\|x - \bar{r}\| \geq d(x, \mathcal{S})$ in the second step and we did change of variables so that the integral reduces to the tail probability of $\chi_p^2$ distribution. By Mills ratio, the tail probability of $\chi_p^2$ is given by

$$\mathbb{P}(\chi_p^2 \geq 2Nc_0 \log n/(1 + \tau_n)) \leq Cn^{-Nc_0/(1+\tau_n)}\big(2Nc_0 \log n/(1 + \tau_n)\big)^{p/2-1},$$

we obtain

$$P \leq C\binom{n}{N}NC_p^{N-1}(\Delta/\sigma)^{p(N-1)}N^{-p/2}n^{-Nc_0/(1+\tau_n)}(2Nc_0 \log n)^{p/2-1}.$$

Using the approximation $\binom{n}{k} \leq C(en/k)^k$, we deduce that

$$P \leq C\left[e(2Nc_0 \log n)^{(p-2)/(2N)}C_p^{1-1/N}N^{(1-p/2)/N} \cdot \frac{n^{1-c_0/(1+\tau_n)}(\Delta/\sigma)^{p(1-1/N)}}{N}\right]^N$$

$$=: C\left[A(n, p, N) \cdot \frac{n^{1-c_0/(1+\tau_n)}(\Delta/\sigma)^{p(1-1/N)}}{N}\right]^N$$

Now we plug in $N = \log(n)$ and $\Delta = c_3\sigma\sqrt{p}\left(\frac{\log(n)}{n^{1-c_0}}\right)^{1/p}$ for a constant $c_3 \leq c_2$ where $c_2 = 0.9(2e^{2-c_0})^{-1/p}\sqrt{(2/p)}(\Gamma(p/2 + 1))^{1/p} = 0.9e^{-(2-c_0)/p}C_p^{-1/p}/\sqrt{p}$ with $C_p = 2^{1-p/2}/\Gamma(p/2 + 1)$. It is straightforward to compute that

$$A(n, p, N) \cdot \frac{n^{1-c_0/(1+\tau_n)}(\Delta/\sigma)^{p(1-1/N)}}{N}$$

$$\leq e^{1-(2-c_0)(1-1/\log(n))}2^{\frac{p-2}{2\log(n)}}(c_0 \log(n))^{\frac{p-2}{2\log(n)}}(0.9)^{p(1-1/\log(n))}n^{\tau_n c_0/(1+\tau_n)}\left(\frac{n^{1-c_0}}{\log(n)}\right)^{1/\log(n)}$$

$$\leq e^{o(1)}(0.9)^p < 1.01 \cdot 0.9 < 1$$

under the condition that $p \ll \log(n)/\log\log(n)$, which also give rise to $\tau_n \log(n) = o(1)$. This implies $P \leq C(0.909)^{\log(n)} = o(1)$.

In the mean time, for each vertex $v_k$, recall that $J_k = \{i : r_i = v_k\}$,

$$N(\mathcal{B}(v_k, \Delta/2)) \geq \sum_{i \in J_k}\mathbf{1}(x_i \in \mathcal{B}(v_k, \Delta/2)) = \sum_{i \in J_k}\mathbf{1}(\|\varepsilon_i\| \leq \Delta/2) \geq mp_\Delta - C\sqrt{mp_\Delta \log\log(n)}.$$

with probability $1 - o(1)$, and

$$p_\Delta := \mathbb{P}(\|\varepsilon_i\| \leq \Delta/2) = \mathbb{P}(\chi_p^2 \leq 4^{-1}(\Delta/\sigma)^2) \geq \frac{e^{-(\Delta/\sigma)^2/8}2^{-p}}{2^{p/2}\Gamma(p/2 + 1)}(\Delta/\sigma)^p$$

Recall the condition that $m \geq n^\delta n^{1-c_0}$. It follows that

$$mp_\Delta \geq n^\delta\frac{e^{-(\Delta/\sigma)^2/8}2^{-p}}{2^{p/2}\Gamma(p/2 + 1)}n^{1-c_0}(\Delta/\sigma)^p = n^\delta\frac{e^{-(\Delta/\sigma)^2/8}}{2^{p/2}\Gamma(p/2 + 1)} \cdot \frac{c \log(n)}{C_p}2^{-p}(c_3/c_2)^p$$

$$\geq cn^\delta 2^{-p}(c_3/c_2)^p \log(n) \gg \log(n)$$

where $c > 0$ is some small constant. The last step is due to the fact that $n^\delta 2^{-p}(c_3/c_2)^p = e^{\delta\log(n)-p\log(2c_2/c_3)} \gg 1$ as $2c_2/c_3 \geq 2$ is a constant and $p \ll \log(n)/\log\log(n)$. Thus, with probability $1 - o(1)$, $N(\mathcal{B}(v_k, \Delta/2)) \gg \log(n)$. Under this event, for any point $X_{i_0} \in \mathcal{B}(v_k, \Delta/2)$, immediately $\mathcal{B}(v_k, \Delta/2) \subset \mathcal{B}(X_{i_0}, \Delta)$ and further $N(\mathcal{B}(X_{i_0}, \Delta)) \gg \log(n)$. Combining this, with $P = o(1)$ and $N(\mathcal{R}_f) = o_\mathbb{P}(1)$, we conclude that we can prune out all points with a distance to the simplex larger than $\sigma\sqrt{2c_0 \log(n)}$ while preserve those points near vertices, with high probability. Thus we finish the claim for $\beta_{new}(X^*)$.

The last claim follows directly from Theorem A (Theorem 1 in the manuscript) under condition (9). We therefore conclude the proof.

$$\square$$

We briefly present the proof of Theorem B below.

*Proof.* The proof strategy is roughly the same as that of Theorem C When $m > c_1 n$, we take $\Delta = c_3 \sigma \sqrt{p} \left( \frac{\log(n)}{n^{1-\delta_n}} \right)^{1/p}$ where $p/\log(n) \ll \delta_n \ll 1$ and $c_3 \leq c_2$, then similarly we can derive that $N(\mathcal{B}(v_k, \Delta/2)) \geq c \log(n) n^{\delta_n} a^p = c \log(n) e^{\delta_n \log(n) - p \log(1/a)} \gg \log(n)$ where $c > 0$ is a small constant and $0 < a \leq 1$. This gives rise to the conclusion that with high probability, $N(\mathcal{B}(X_{i_0}, \Delta)) \gg \log(n)$ for any $X_{i_0} \in N(\mathcal{B}(v_k, \Delta/2))$. Moreover, in the same manner to the above derivations, replacing $c_0$ by $\delta_n$, we can claim again that $N(\mathcal{R}_f) = o_{\mathbb{P}}(1)$ and

$$P \leq C \left( A(n, p, \log(n)) \cdot \frac{n^{1-\delta_n/(1+\tau_n)}(\Delta/\sigma)^{p(1-1/\log(n))}}{\log(n)} \right)^{\log(n)} = o(1).$$

Consequently, all the claims follow from the same reasoning as the proof of Theorem C. We therefore omit the details and conclude the proof . $\square$

### C.3 PROOF OF LEMMA H

Recall that $R = n^{-1/2}[r_1 - \bar{r}, \ldots, r_n - \bar{r}]$. Let $R = U_0 D_0 V_0$ be its singular value decomposition and let $H_0 = U_0 U_0'$. Denote $\epsilon = [\epsilon_1, \ldots, \epsilon_n] \in \mathbb{R}^{d,n}$. We start by analyzing the convergence rate of $\|ZZ' - nRR' - n\sigma^2 I_d\|$. Recall that $\bar{X} = \bar{r} + \bar{\epsilon}$, where $\bar{\epsilon} = n^{-1} \sum_{i=1}^{n} \epsilon_i$. We obtain

$$Z = X_i - \bar{X} = r_i + \epsilon_i - \bar{r} - \bar{\epsilon}, \qquad Z = \sqrt{n}R + \epsilon - \bar{\epsilon}1_n'. \tag{C.75}$$

Observing the fact that $R1_n = 0$, we deduce

$$ZZ' - nRR' - n\sigma^2 I_d = (\sqrt{n}R + \epsilon - \bar{\epsilon}1_n')(\sqrt{n}R + \epsilon - \bar{\epsilon}1_n')' - nRR' - n\sigma^2 I_d$$
$$= \sqrt{n}(\epsilon - \bar{\epsilon}1_n')R' + \sqrt{n}R(\epsilon - 1_n\bar{\epsilon}')' + (\epsilon - \bar{\epsilon}1_n')(\epsilon - \bar{\epsilon}1_n')' - n\sigma^2 I_d$$
$$= \sqrt{n}\epsilon R' + \sqrt{n}R\epsilon' + (\epsilon\epsilon' - n\sigma^2 I_d) - n\bar{\epsilon}\bar{\epsilon}'. \tag{C.76}$$

The above equation implies that

$$\|ZZ' - nRR' - n\sigma^2 I_d\| \leq 2\sqrt{n}\|\epsilon R'\| + \|\epsilon\epsilon' - n\sigma^2 I_d\| + n\|\bar{\epsilon}\|^2. \tag{C.77}$$

We proceed to bound the three terms $\|\epsilon R'\|$, $\|\epsilon\epsilon' - n\sigma^2 I_d\|$ and $n\|\bar{\epsilon}\|^2$ respectively. First, notice that $\epsilon R' \in \mathbb{R}^{d \times d}$ is a Gaussian random matrix with independent rows which follow $N(0, RR')$. By Theorem 5.39 and Remark 5.40 in Vershynin (2010), we can deduce that with probability $1 - o(1)$,

$$n\|R\epsilon'\epsilon R'\| \leq Cnd\sigma^2 s_1^2(R).$$

This, together with the fact that $s_1(R) \leq c$ gives that

$$\sqrt{n}\|\epsilon R' + R\epsilon'\| \leq C\sigma\sqrt{nd}. \tag{C.78}$$

Second, by Bai-Yin law (Bai & Yin (2008)), we can estimate the bound of $\|\mathcal{E}\mathcal{E}' - n\sigma^2 I_d\|$ as follows.

$$\|\epsilon\epsilon' - n\sigma^2 I_d\| \leq n\sigma^2(2\sqrt{d/n} + d/n) \leq \sigma^2(2\sqrt{nd} + d), \tag{C.79}$$

with probability $1 - o(1)$. Third, observe that $\bar{\epsilon} \sim N(0, \sigma^2/n I_d)$. We therefore obtain that with probability $1 - o(1)$,

$$n\|\bar{\epsilon}\|^2 \leq \sigma^2[d + C\sqrt{d\log(n)}].$$

By applying the condition that $\sigma = O(1)$, combining the above equation with (C.77), (C.78) and (C.79) yields that, with probability at least $1 - o(1)$,

$$\|ZZ' - nRR' - n\sigma^2 I_d\| \leq 2\sigma\sqrt{nd} + \sigma^2[d + C\sqrt{d\log(n)}] + \sigma^2(2\sqrt{nd} + d)$$
$$\leq C(\sigma\sqrt{nd} + \sigma^2 d). \tag{C.80}$$

Now, we compute the bound for $\|\widehat{H} - H_0\|$. Let $U^\perp, U_0^\perp \in \mathbb{R}^{d, d-K+1}$ such that their columns are the last $(d - K + 1)$ columns of $U$ and $U_0$, respectively. It follows from direct calculations that

$$\|\widehat{H} - H_0\| = \|U_0 U_0' - UU'\| \leq \|U_0^\perp(U_0^\perp)'(U_0 U_0' - UU')\| + \|U_0 U_0'(U_0 U_0' - UU')\|$$
$$= \|U_0^\perp(U_0^\perp)'UU'\| + \|U_0 U_0'U^\perp(U^\perp)'\| \leq \|(U_0^\perp)'U\| + \|U_0'U^\perp\| = 2\|\sin\Theta(U_0, U)\|.$$

Notably, $U, U^\perp$ is also the eigen-space of $ZZ' - n\sigma^2 I_d$. By Weyl's inequality (see, for example, Horn & Johnson (1985)),

$$\max_{1 \le i \le d} \left| \lambda_i(ZZ' - n\sigma^2 I_d) - \lambda_i(nRR') \right| \le C \|ZZ' - n\sigma^2 I_d - nRR'\|$$

Under the condition that $s_{K-1}^2(R) \gg \max\{\sqrt{\sigma^2 d/n}, \sigma^2 d/n\}$, by Davis-Kahan Theorem (Davis & Kahan (1970)), we deduce that, with probability at least $1 - o(1)$,

$$\|\widehat{H} - H_0\| \le 2\|\sin\Theta(U_0, U)\| \le \frac{2\|ZZ' - nRR' - n\sigma^2 I_d\|}{\lambda_{K-1}(nRR')}$$
$$\le C \frac{\max\{\sqrt{\sigma^2 d/n}, \sigma^2 d/n\}}{s_{K-1}^2(R)}. \tag{C.81}$$

The proof is complete.

## D NUMERICAL SIMULATION FOR THEOREM A

In this short section, we want to provide a better sense of our bound derived in Theorem A and how it compares with the one from the orthodox SPA. To make it easier for the reader to see the difference between the two bounds, we consider toy example where we fix $(K, d) = (3, 3)$ and

$$\widetilde{V} = \{(20, 20, 0), (20, 30, 0), (30, 20, 0)\}$$

while we let

$$V = \widetilde{V} + a \cdot (0, 0, 1).$$

We consider $50$ different values for $a$ ranging from $10$ to $1000$. It is not surprising to see that when $a$ is close to $0$ the bound of the orthodox SPA goes to infinity whereas as the simplex is bounded far away from the origin, the $K^{th}$ singular value will be bounded away from $0$. However, our bound still outperforms the traditional SPA bound even for very large values of $a$. Looking at two specific values of $a$ we have the following. For $a = 10$,

$$\beta_{new} = 0.03, \qquad \beta(V) = 0.05$$

Moreover, as $a$ changes, the Figure 2 below illustrate how much the ratio of

$$\frac{\text{our whole bound}}{\text{Gillis bound}}$$

changes as the parameter $a$ changes. For example, when $a = 10$.

$$\frac{g_{new}(V)}{g(V)} = 0.015,$$

and so

$$\frac{\text{our whole bound}}{\text{Gillis bound}} = 0.009$$

so we reduce the bound by $111$ . Similarly, when $a = 1000$,

$$\frac{g_{new}(V)}{g(V)} = 0.19, \qquad \frac{\text{our whole bound}}{\text{Gillis bound}} = 0.105,$$

so we have reduced the bound by $9.5$.

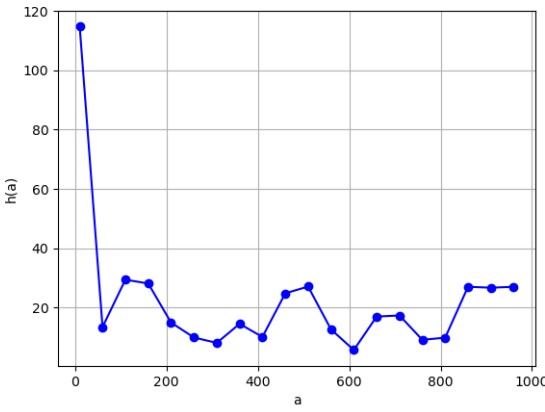

Figure 2: Factor of improvement of our bound over orthodox spa as the true simplex moves away from origin by a distance $a$.