# OpenReview forum: "Improved algorithm and bounds for successive projection"
_ICLR.cc/2024/Conference — ICLR 2024 poster_

### Official Review · Reviewer_WFSc · 2023-10-31

**Soundness:** 2 fair
**Presentation:** 2 fair
**Contribution:** 2 fair
**Rating:** 3
**Confidence:** 3

**Summary:**

This paper consider recovering the k-vertex simplex. Exploiting the idea of K-nearest neighbor, this paper improves the SPA algorithm (Araujo et al. 2001) and propose pp-SPA (shown in Algorithm 2).  Compared with the analysis in Gilis and Vavasis, this paper obtain some improvements as in Theorem 2 and Theorem 3.

**Strengths:**

This paper proposes a novel algorithm to reconstruct the k-vertices and analyze its performance. The analysis seems to be solid. Compared with the prior work (Gillis and Vavasis), improvements are obtained under certain scenarios.

**Weaknesses:**

+ This paper seems to be a incremental work based on Gillis & Vavasis (published in 2013).

+ There is only limited numerical experiments to validate the peformance of the proposed algorithms. (c.f. Figure 2).

**Questions:**

1. Will more samples contribute to the reconstruction performance? My intuition is that the reconstruction performance will improve with more observations, that is, the reconstruction error will decrease with an increasing sample $n$. However, the relation between sample number $n$ and the reconstruction error is largely missing.

2. How is the algorithm compared with the matrix completion solution? To be more specific, we can stack the observations and obtain the sensing relation $X = V\Pi + N$, where observation $X_i$ is the $i$th column of matrix $X$, and $N$ is the sensing noise. Easily, we can verify that $V\Pi$ is a matrix with rank at most $K$. Hence, we can view matrix $X$ as a linear combination of low-rank matrix and noise matrix. Then, we exploit the research on low-rank matrix reconstruction and solve the problem.

3. In comparing the performance between Theorem 1 over Lemma 4, what if we have $\gamma(V) \ll s_{K-1}(V)$? In such case, lemma 4 may be more tight then Theorem 1.

---

> ### Comment · Reviewer_KsqF · 2023-11-10
>
> I have one comment about this review (I'm a reviewer, and not the original author)
>
> ```This paper seems to be a incremental work based on Gillis & Vavasis```
>
> This is a completely incorrect assessment. The proofs are clearly not the same. Working on the same problem/algorithm is not incremental work no matter how you spin it.

---

> > ### Author Response · Authors · 2023-11-10
> > **Comparison with Gillis and Vavasis**
> >
> > Dear All,
> >
> > 1. Reviewer WFSc said "This paper seems to be a incremental work based on Gillis & Vavasis (published in 2013)".
> > 2. A couple of minutes ago,  Review KsqF posted a comment saying that the above comment by WFSc is incorrect, where he/she further said "he proofs are clearly not the same. Working on the same problem/algorithm is not incremental work no matter how you spin it".
> >
> > We thank all reviewers for their valuable time and input, and we especially thank Review KsqF for his/her timely comment. While we are preparing a more thorough response letter, we just wish to point out that
> >
> > -- We have two contributions: a new non-asymptotic bound (which is what review WFSc commented on) and a new algorithm and a set of new results. Gillis and Vavasis's work is only related to the first one (the second contribution goes far beyond Gilis and Vavasis work and is not related.
> >
> > -- Even our first contribution is not an incremental work based on Gillis and Vavasis: you will agree if you read our statement and proofs. Here is a simple way to see it. Our bound says
> >
> > max_{1 leq i \leq K} |\hat{v}_k - v_k| <= C / s_{K-1}(\tilde{V}),
> >
> > while Gillis and Vavasis' bound says
> >
> > max_{1 leq i leq K} |hat{v}_k - v_k| <= C / s_k (V).
> >
> > In the denominator of Gillis and Vavasis bound, s_K(V) does not depend on the volume of the simplex (in fact, it depends on the location of the simplex). Therefore, even when the volume is large (but the location is bad), s_K(V) is 0. In such a case, the RHS of their bound is infinity, and so the bound is not useful. In our bound, s_{K-1}(tilde{V}) can be proved to be lower bounded by a simple function of the volume, so as long as the volume of the simplex is large, s_{K-1}(tilde{V}) is lower bounded from 0. Therefore, our bound is significantly better in this sense. Note that an optimal bound should depend on the volume but not the location of the simplex (a simplex with a small volume is nearly flat so vertex hunting is impossible; however, the location of the simplex should not matter if we have the right algorithm or the right proof).
> >
> > Given such a significant difference, I think you will agree that no one can deduce our results from Gillis and Vavasis (otherwise,
> > I think Gillis and Vavasis will improve their bounds a long time ago).
> >
> > Thanks again for your valuable time and input. We will be in touch soon!

---

> > > ### Comment · Reviewer_KsqF · 2023-11-21
> > >
> > > Hi, did you ever submit your rebuttal? I am not able to see it (in case you did can you check if all reviewers are able to see it?)

---

> > > > ### Author Response · Authors · 2023-11-21
> > > > **Revision submission**
> > > >
> > > > Thanks a lot for the kind reminder. We are still working to polish the rebuttal and revision. We expect to submit it later this evening (or at the latest tomorrow morning).

---

> ### Author Response · Authors · 2023-11-22
> **A more complete response to Reviewer WFSc - Part 1**
>
> Thank you for your comments and for appreciating that "this paper proposes a novel algorithm" and "the analysis seems to be solid". We already posted a quick response earlier. This current response is to supplement the earlier one with more details. We also encourage you to read our Response-to-All-Reviewers for other clarifications.
>
> ### Main Points:
>
> **1. Comparison with [GV13]**: *"This paper seems to be an incremental work based on Gillis & Vavasis (published in 2013)."*
>
> We are sorry to hear this comment, but we believe it is due to some misunderstanding. It is our fault that our initial submission did not explain why the proof is non-trivial. We hope that you will change your mind after seeing the clarification here.
>
> Our paper has 3 major contributions:
> - (a) A new algorithm
> - (b) the non-asymptotic bound for SPA,
> - (c) the error bound for pp-PSA.
>
> Only (b) is related to Gillis & Vavasis (2013) [GV13]. Even without (b), the contributions in (a) and (c) are already significant. For example, our analysis in pp-SPA uses theory of extreme order statistics to analyze the effect of KNN denoise, which is subtle.
>
> Regarding the contribution in (b), we guess you were actually asking: "Is the proof a simple modification of the proof in [GV13]?"
>
> The proof is completely different. The proof in [GV13] is based on controlling matrix norms. Their proof does not use explicitly the geometry behind this problem. To see this, we recall that $\beta(X)$ is equal to the maximum of $ \Vert\epsilon_i\Vert$ among all points,  and $\beta_{\text{new}}(X)$ is roughly the maximum of $\Vert\epsilon_i\Vert$ among those points outside the true simplex. In the proof of [GV13], they frequently control the difference of replacing $X$ by $\mathbb{E}[X]$ in many intertermediate matrices. Such steps are feasible only if one has a bound for the maximum of *all* $\Vert\epsilon_i\Vert$'s - i.e., the quantity $\beta(X)$. However, it is hard to repeat these steps by using $\beta_{\text{new}}(X)$. Similarly, if one has a lower bound for $s_K(V)$, one can control the inverse of $V'V$, but a lower bound for $s_{K-1}(V)$ cannot do so.
> To tackle these difficulties, we use a different proof approach. Our proof relies more on the geometry behind this problem. We define a "simplicial neighborhood" for each true vertex and prove that each estimated vertex must fall into one of these simplicial neighborhoods. This is a new proof strategy.
>
> In addition, we have added a numerical example in this revised draft, where we calculate the two bounds explicitly. In some cases, the new bound is only 0.01 times the old bound in [GV13].
>
> In summary, we agree that [GV13] is an influential work, but our paper is not incremental of [GV13].
>
> **2. More simulations**: *"There is only limited numerical experiments to validate the performance of the proposed algorithms"*
>
> Thanks for your comment. We have significantly expanded our simulations:
> - We have included settings for every $d=1,2,..., 48, 49, 59$.
> - We have added the comparison with robust SPA in all simulation settings.
>
> The updated Figure 3 contains nearly 90 different settings. We hope that the numerical experiments are sufficiently rich now.

---

> ### Author Response · Authors · 2023-11-22
> **A more complete response to Reviewer WFSc - Part 2**
>
> ### Other questions:
>
> 1. The relation between sample number $n$ and the reconstruction error.
>
> The reconstruction error is $O(\sqrt{d+\log(n)})$ for SPA and $O(\sqrt{\log(n)})$ for pp-SPA. In addition, in some special cases (e.g., Theorem 2), the error of pp-SPA can be further reduced to nearly $O(1)$.
>
> At first glance, it is a bit counter-intuitive that increasing $n$ does not decrease the rate. The main reason is that the configuration of population points $r_1, r_2, \ldots, r_n$ can be arbitrary. Here is a toy example. Suppose
> $$
> r_k = v_k, \qquad 1\leq k\leq K, \qquad r_i = \bar{v}:= (1/K)\sum_kv_k, \qquad\mbox{for all the remaining $(n-K)$ points}.
> $$
> In this example, no matter how large $n$ is, there is only one point, $X_k$, that is useful for estimating $v_k$. This is why increasing $n$ does not help reduce the error bound.
>
> This insight is further supported by simulations. On the right panel of Figure 3, we plot the error curve as $n$ increases. It shows that the error does not decrease with $n$, which is consistent with our theory.
>
> 2. How is the algorithm compared with the matrix completion solution?
>
> This is an interesting comment. The main difference is: We are interested in recovering $V$, but the matrix completion literature is interested in recovering $V\Pi$. We can find $\widetilde{V}\in\mathbb{R}^{d\times K}$ and $\widetilde{\Pi}\in\mathbb{R}^{K\times n}$ such that $\widetilde{V}\widetilde{\Pi}=V\Pi$ but $\widetilde{V}\neq V$.
>
> For example, let's consider a special case with $K=3$ and $d=2$. The true simplex is a triangle in $\mathbb{R}^2$. We now consider an arbitrary triangle that contains the true triangle in its interior. Let $\widetilde{V}$ contain vertices of this new triangle, and let $\widetilde{\Pi}$ contain the barycentric coordinates of $r_i$'s in this new triangle. Then, $V\Pi=R=\widetilde{V}\widetilde{\Pi}$. However, $V\neq \widetilde{V}$.
>
> This example suggests that it is impossible to naively apply matrix completion. However, there may be a way to borrow some ideas from matrix completion literature to further improve the current algorithms. We leave this interesting direction to future work.
>
> 3. What if we have $\gamma(V)\ll s_{K-1}(V)$?
>
> Thanks for the comment. This will never happen, as a result of Lemma D in the supplementary material.
>
> For your convenience, we re-prove Lemma D here. By definition,  $\gamma^2(V)$ is the maximum diagonal entry of $V'V$. Note that
> $$
> K\gamma^2(V)=K\max_{1\leq k\leq K}\Vert v_k\Vert^2\geq \sum_{k=1}^K \Vert v_k\Vert^2
> =  \mathrm{trace}(V'V).
> $$
> In addition, let $s_k(V)$ be the $k$th singular value of $V$, so that $s_k^2(V)$ is the $k$th largest eigenvalue of $V'V$. It follows that
> $$
>  \mathrm{trace}(V'V)  = \sum_{k=1}^K s_k^2(V)\geq \sum_{k=1}^{K-1} s_k^2(V)\geq \sum_{k=1}^{K-1} s_{K-1}^2(V) = (K-1) s_{K-1}^2(V).
> $$
> Combining the above, we have
> $
>  \gamma(V) \geq \sqrt{\frac{K-1}{K}} s_{K-1}(V)\geq \frac{1}{\sqrt{2}}s_{K-1}(V).
> $

---

> ### Comment · Reviewer_WFSc · 2023-11-23
> **Concern over the explain of matrix completion**
>
> Thank you for your reply.
>
> First, I am not totally satisfied with the explanation on the matrix completion.
>
> In the field of the matrix completion research, ppl do not discuss whether $V$ equals $\tilde{V}$ but whether they are the same under some rotations. This is because  we always have the relation $VO O^T \Pi = V \Pi$, where $O$ is an orthonormal matrix.
>
> I suspect that this problem can be solved in the framework of matrix completion. All you need is to align the matrix to the vertices. This is not very hard. For example, you can use the fact that the points lies in the vertices, i.e., the linear combination coefficients are sparse.
>
> Also, I am not satisfied with the explanation that sample number $n$ does not play any role. Your paper considers a noisy sensing relation. Even under the duplicate measurements setting, you can improve the performance by averaging the duplicate measurement.
>
> For example, suppose you have $n$ measurements $r_i = v + \epsilon_i$, where $\epsilon_i$ is zero mean and $\sigma^2$ variance. Averaging $n$ cases will reduce the sensing noise variance to $\sigma^2/n$. In this way, I am a little surprised that the measurement plays no role. Or this is an artifact of this line of algorithm?

---

> ### Comment · Reviewer_WFSc · 2023-11-23
>
> Dear reviewer KsqF, I do not want to get into such an argument on the issue such whether I spin this work or not. Still, I believe I am allowed to make an independent decision and am qualified to make one.
>
> I reviewed the paper carefully and acted in a  responsible way. I have provided some unique angles to this paper. Even the authors themselves agree with me, as you can see in their own words.
>
> First, I pointed out that this problem can be studied in the framework of matrix completion. Even the authors themselves agree it is a future research direction they want to explore. I quote their words here "there may be a way to borrow some ideas from matrix completion literature to further improve the current algorithms. We leave this interesting direction to future work."
>
> Second, I asked about the role of the sample number, a question seldom asked. And the author's response is not satisfactory, as I have pointed out in my response. For example, even with duplicate measurements, more measurements can be used to reduce the measurement error. This does not justify the missing role of the sample number in the reconstruction performance. The authors' responses do not address this.
>
> I am open to changing my ideas if some reasonable arguments can be provided. Thank you.

---

### Official Review · Reviewer_xzLC · 2023-10-31

**Soundness:** 3 good
**Presentation:** 3 good
**Contribution:** 3 good
**Rating:** 6
**Confidence:** 2

**Summary:**

The *main problem* tacked by the reviewed paper is the **vertex hunting**, i.e.:  the estimation of $K$ vertices defining a simplex in $\mathbb{R}^d$ given $n$ sampled points belonging to the simplex perturbed by iid zero-centered Gaussian noise with variance $\sigma$.  The discussed problem is essential in many problems in ML/data analysis, e.g.: hypespectral unmixing, archetypal analaysis, and community detection in networks, topic modelling in NLP.

The *main innovation* is the *pseudo-point successive projection algorithm* (pp-SPA), which is the improvement of the well studied SPA method that is simple and effective. The proposed innovation consists of two parts: (i) in applying PCA to estimate the K-1 dimensional linear subspace defining the simplex (ii) to perform K-nearest neighbours averaging on the points. Both of these actions are meant to help with denoising of the given sample points and to help with correct estimation in high-noise settings.

The *main contribution* of the paper, apart from proposing the two denoise methods to SPA, comes from the supplied theoretical analysis. The authors show an upper bound on the error that depends on $K-1$th singular value of the matrix formed by the concatenated sampled points $X = [x_1,x_2, \ldots x_n]$. This is a contrast to the existing bound that includes $K$th singular value in the work of (Gillis & Vavasis, 2013) that proved the first recovery bounds for the basic SPA algorithm. The authors claim that this is a significant difference, especially since the sampled points are guaranteed to lie in a $K-1$ lin. subspace, thus the $K-1$th singular value is bounded away from zero, while the $K$th singular value is not.

The *numerical comparisons* with the basic SPA and the related ablation studies are provided in the text and show benefits of the proposed methods in the specific cases of small $d=2$ and $K=3$, however, other situations of interest when the denoise function can be more pronounced, e.g., when d >> K is not showed.

The proofs seem to be technically involved, but it is difficult for me to check the soudness of the results as I am not an expert in the area.

**Strengths:**

I believe, the paper brings potentially several strong contributions:
1) The authors propose algorithm seems to practically make a lot of sense
2) The theoretical results are novel and non-trivial. The authors exploit the specific Gaussianity of the noise to be able to derive these bounds
3) The experiments are limited, but they support the statements of the paper

**Weaknesses:**

There are some considerable difficulties I have with the paper:
1) The writing and the structure of the text should be improved. For example:
* It is not clear to me what the illustration in Figure 1 shows? Is it obvious that one of these is better and what does ``idea simplex'' denote?
* At times a statement is given without citation, e.g. you say in "Our contributions.", pg 2  that: "since the SPA is greedy algorithm, it tends to be biased outward bound.", but it is not clear where this can be seen.
* It is very difficult to follow the motivation of Theorem 1, and how it shows that the previous results of (Gillis & Vavasis, 2013) is non-satisfactory. It is very difficult to see the comparison with the bound in (5) for $\beta_{new}$.
2) The numerical experiments are done only for very small sizes. I am not sure how big is the effect of denoising using low-dimensional PCA projection in this case.

**Questions:**

* How does the theory differ in the case of 3-simplex in 2D? Is there a difference in the $K$ vs $K-1$ singular value?
* How does the algorithm perform for higher dimensional problems?
* In the second sentence on the first page you state that Gaussian assumption can be relaxed. Can the theory by applied also when the errors are not Gaussian?

---

> ### Author Response · Authors · 2023-11-22
> **Response to Reviewer xzLC - Part 1**
>
> Thank you for your helpful comments. We are glad that you appreciate our contributions in both algorithm and theory. We feel especially encouraged that you agreed that "theoretical results are novel and non-trivial". As you have pointed out, the analysis exploits the tail bounds of Gaussian random vectors and is quite subtle (especially to get the tight constants).
>
> Below is our point-to-point response. We also encourage you to read our Response-to-All-Reviewers for other clarifications.
>
> ### Main comments:
>
> **1. Figure 1**: *"It is not clear to me what the illustration in Figure 1 shows? What does 'idea simplex' denote?"*
>
> Thanks for the comment. In this revision, we have combined two panels of Figure 1 into a single plot, to make it easier to compare the true vertices and the estimates from SPA and pp-SPA. The  "ideal simple" means the ground-truth simplex formed by the true vertices $v_1,v_2,\ldots,v_K$. We have changed the terminology "ideal simplex" to "the simplex formed by the true vertices" to make it clear.
>
> **2. Citations to the biased outward bound**: *"At times a statement is given without citation,... pg 2: 'since the SPA is greedy algorithm, it tends to be biased outward bound.', but it is not clear where this can be seen."*
>
> We realized that since we put SPA and pp-SPA in two separate panels in the old Figure 1, the "biased outward bound" was not easy to see. In the updated Figure 1, the two simplexes are put together, so this phenomenon is more obvious to see.
>
> This phenomenon is caused by the way SPA is designed: At each iteration, it selects $\hat{v}_k$ by finding the point furthest to the original. No matter the original is inside or outside the true simplex, this will yield "biased outward bound". We think this phenomenon was already recognized in the robust SPA literature, although not using the exact phrase. In this revision, we added a reference Gillis (2019) in this paragraph.
>
> **3. The bound in Theorem 1**: *"How it shows that the previous results of (Gillis & Vavasis, 2013) is non-satisfactory. It is very difficult to see the comparison with the bound in (5) for $\beta_{new}$."*
>
> Your first question is about why the bound in [GV13] is non-satisfactory. We now give a toy example where we fix $(K, d) = (3,3)$, $v_1=(20,20,a)$, $v_2=(20,30,a)$, and $v_3=(30,20,a)$, for a parameter $a\in\mathbb{R}$.
> Recall their bound and our bound are  $g(V) \cdot \beta(X)$ and $g_{\text{new}}(V) \cdot \beta_{new}(X,V)$, respectively, where $\beta_{new}(X,V)\leq \beta(X)$ is easy to see. We now compare $g_{\text{new}}(V)$ and $g(V)$. By direct calculations, we get the following table. It demonstrates that the improvement is substantial.
> $$
> \begin{array}{ccccccc}
>     a & 10 & 110 & 210 & 310 & 410 & 510 & 610 & 710 & 810 & 910\cr
>     \frac{g(V)}{g_{\text{new}}(V)} &114.8 & 11.2 & 27.6 &27.2 &14.6 &9.7 &7.9 &14.4 & 10&24.6
>   \end{array}
> $$
> The main reason is that $s_{K-1}(V)$ is driven by the volume of the simplex, but $s_K(V)$ is driven by the location (center) of the simplex. As $a$ varies, the location of the simplex can change significantly, making $g(V)$ unsatisfactory sometimes.
>
> Your second question is about how to interpret $\beta_{\text{new}}(X,V)$. In this revision, we added another toy example and a new figure, Figure 2, to illustrate this quantity. Intuitively speaking,
> - $\beta(X)$ is the maximum noise in all all data points, no matter whether the data points fall inside or outside the true simplex.
> - $\beta_{\text{new}}(X, V)$ only cares about the maximum noise of data points outside the true simplex or near the boundary of the true simplex.
>
> As a result, we can add a large noise to any point deeply in the interior of the true simplex without changing $\beta_{\text{new}}(X,V)$, but $\beta(V)$ will be heavily increased. This improvement of $\beta(V)$ to $\beta_{\text{new}}(X,V)$ is critical in the analysis of the new algorithm, pp-SPA. The KNN denoise idea only helps decrease the noise at those points outside the true simplex. If we stick to the original non-asymptotic bound in [GV13], then we will not be able to derive the sub-sequent bounds for pp-SPA.
>
> **4. Simulations for larger $d$**: *"The numerical experiments are done only for very small sizes. I am not sure how big is the effect of denoising using low-dimensional PCA projection in this case."*
>
> This is a great point. We have expanded Experiment 1 by considering every $d$ from $1$ to $50$. The effect of low-dimensional PCA projection can be seen by comparing pp-SPA and D-SPA (a simplified version with the projection step skipped). We see that the estimation error of D-SPA deteriorates as  $d$ gets larger. However, the estimation error of pp-SPA is almost invariant with respect to the increase of $d$. This shows that the hyperplane projection is crucial for the performance, especially when $d$ is large.

---

> ### Author Response · Authors · 2023-11-22
> **Response to Reviewer xzLC - Part 2**
>
> ### Other questions:
>
> - *"How does the theory differ in the case of 3-simplex in 2D? Is there a difference in the $K$ v.s. $K-1$ singular value?"*
>
> Thanks for the interesting question. Let's consider two cases: (a) The triangle lives in $\mathbb{R}^2$. Then, $V$ is a $2\times 3$ matrix that has only $2$ singular values. The 3rd singular value is exactly zero. (b) The triangle lives in a hyperplane in $\mathbb{R}^3$. In this case, $V$ is a $3\times 3$ matrix, where 2nd singular value is nonzero as long as the triangle is non-degenerate (i.e., it does not collapse to a line segment). However, the 3rd singular value depends on the location of the hyperplane. When the hyperplane crosses the origin, the 3rd singular value can be zero.
>
> The above conclusions can be deduced from Lemma 3 in the revised draft.
>
> - *"How does the algorithm perform for higher dimensional problems?"*
>
> Please see our response to your Main Point 4 (in Part 1 of our responses).
>
> - *"In the second sentence on the first page you state that Gaussian assumption can be relaxed. Can the theory by applied also when the errors are not Gaussian?"*
>
> Yes, the theory can be extended to nonGaussian noise. Take $d=1$ for example. Suppose $\epsilon_i$'s are i.i.d. from a univariate mean-zero distribution $F$. Let  $y_k$ be the $k$-th largest value in $\{|\epsilon_1|, \ldots, |\epsilon_n|\}$. In the Gaussian case of $F = N(0,1)$, the key fact we need for our proof is that
>  $$
>  y_k \approx \sqrt{2 \log(n/k)}, \qquad \mbox{for all }1 \ll k \ll n.
>  $$
> To extend our proofs to a general $F$, we need to replace $\sqrt{2 \log(n/k)}$ by another quantity $c_{n,k}$. Such results for a general $F$ can be found in the literature of order statistics. For example, when $F$ is the Laplace distribution, we have
> $$
> y_k \approx   \log(n/k), \qquad  \mbox{for all }1 \ll k \ll n.
> $$
> Using similar proofs, we can extend our results to Laplace noise. For $d>1$, we change $|\epsilon_k|$ to $\Vert \epsilon_k\Vert $ in the definition of $y_k$. For settings where either $d$ coordinates of $\epsilon_i$ are independent or $\epsilon_i$ follows an elliptically contour distribution, our proofs can be generalized.

---

### Official Review · Reviewer_wP9r · 2023-10-31

**Soundness:** 3 good
**Presentation:** 2 fair
**Contribution:** 3 good
**Rating:** 6
**Confidence:** 2

**Summary:**

The paper describes an algorithm for the vertex hunting problem. In this problem, one is given a set of points from a given simplex that underwent some noise. The goal is to identify the simplex, i.e., to find its vertices. A previously known algorithm for this problem is SPA, which repeatedly chooses the largest sampled point as a vertex, then projects the remaining points to its orthogonal space.

This paper proposes a different algorithm, which is more appropriate for handling noise.
The algorithm first identifies a subspace in which the simplex might exist, and projects the points to this subspace.
Then, a denoising step eliminates "outliers".
Finally, standard SPA is applied to the remaining points.

The paper evaluates this method both theoretically and experimentally.
On the theoretical side, the paper introduces an improved analysis for classic SPA, and shows further improvement is achieved by their method (pp-SPA).
Here, performance is measured in terms of the maximum distance between an estimated vertex and a matched actual vertex (for the best possible matching).
On the experimental side, the paper compares pp-SPA to standard SPA, as well as variants that use only some of the steps in pp-SPA (only projection and only denoising), and shows that pp-SPA achieves the best reconstruction error.

**Strengths:**

The paper gives ample motivating examples for which the problem is relevant. The paper makes attempts to motivate the given bounds. The algorithm itself seems intuitive.

**Weaknesses:**

The bounds given in the paper, e.g. Theorem 2, are very complex; they contain multiple terms and are subject to many conditions. It is hard to understand whether those conditions are restrictive, or whether the bounds are significant.
I think the presentation of the problem could be expanded upon.

**Questions:**

none

---

> ### Author Response · Authors · 2023-11-22
> **Response to Reviewer wP9r**
>
> Thank you for the helpful comments. We are glad that you think our problem is well-motivated and that our algorithm is intuitive. Your comment is mainly on the presentation of theorems. In this revision, we have made several changes to make the theoretical results easy to digest. (PS: We also encourage you to read our Response-to-All-Reviewers for other clarifications.)
>
>  First, we move Table 1 to the beginning of Section 4. This table provides an explicit comparison of bounds for SPA, P-SPA, D-SPA, and pp-SPA. In the paragraph below Table 1, we also explain carefully how to use this table to see (a) the advantage of hyperplane projection, (b) the advantage of KNN denoise, and (c) the advantage of combining both. Long story short: pp-SPA yields a strict improvement over SPA in all cases.
>
> Second, we have followed your suggestion to explain the orders of $(\alpha_n, b_n, t_n)$ in Theorem 2. Take the case where $K=O(1)$ and $d=o(\sqrt{n})$ for example. The orders of $\alpha_n$ and $b_n$ are directly from Equation (11), and the order of $t_n$ is in the theorem body:
> $$
> \alpha_n \asymp \sqrt{[d^2 + d\log(n)]/n}, \qquad  b_n\asymp \sigma\sqrt{[d+ \log(n)]/n}, \qquad t_n\asymp [\log(n)]^{\frac{1}{K-1}}/n^{\frac{1-o(1)}{K-1}}.
> $$
> The order of $t_n$ is only used to determine if it is the first or second case of Theorem 2, but $t_n$ itself does not appear in the bound. In addition, it always holds that $b_n=O(\sigma\alpha_n)$ and $\alpha_n=o(1)$. Therefore, $\alpha_n$ and $b_n$ in the bound will be dominated by $\sqrt{\delta_n\log(n)}$, as long as we choose $\delta_n$ that tends to $0$ properly slow. By choosing $\delta_n=\log\log(n)/\log(n)$, we obtain
> $$
> \max_k \|\hat{v}\_k - v\_k\| \leq  \sigma g_{\text{new}}(V)\cdot
> \begin{cases}
> \sqrt{\log\log(n)} & \text{ if $d/n$ is properly small};\cr
> \sqrt{[2+o(1)] \log (n)} & \text{ if $d/n$ is properly large}.
> \end{cases}
> $$
> This is how we get simplified bounds, which are now stated in Equation (12). The above derivation is also in the new paragraph below Theorem 2. At the same time, the bound for SPA is in Equation (14):
> $$
> \max_k \|\hat{v}\_k^{\text{spa}} - v\_k\| \leq  \sigma g_{\text{new}}(V)\cdot
> \begin{cases}
> \sqrt{\max\{d, \; 2 \log(n)\}} & \text{ if $d\ll \log(n)$ or $d\gg \log (n)$};\cr
> \sqrt{a_1 \log (n)} & \text{ if $d= a_0 \log(n)$, where $a_1>2$ is a constant}.
> \end{cases}
> $$
> It is now easy to compare pp-SPA and SPA directly from Equation (12) and Equation (14).
>
> Finally, we want to explain why we did not present these simplified bounds in Theorem 2 but the more complicated ones. The main reason is that we want to cover as broad settings as possible, including finite or diverging $K$, finite or diverging $d$, and fixed or diverging or diminishing $\sigma$. With the general form in Theorem 2, we are able to get different simplified bounds in different cases. Another reason is that we seek for not only the sharp order but also the tight constant. The "constant" comes from properties of the extreme order statistics of chi-square random variables, which  analysis is very subtle. We need to specify $(\alpha_n, b_n, t_n)$ precisely to guarantee that the statements are all rigorous.

---

### Official Review · Reviewer_KsqF · 2023-11-02

**Soundness:** 3 good
**Presentation:** 3 good
**Contribution:** 3 good
**Rating:** 8
**Confidence:** 3

**Summary:**

Fix a dimension $d$, and let $r_1, ..., r_n$ be $n$ vectors that lie on the same $K$ dimension simplex with extremal vertices $v_1, ..., v_K$, where $1\leq K \leq d+1$. The authors consider the (practical) problem of having noisy realisations $X_i = r_i + \epsilon_i$ where $\epsilon_i$ is some Gaussian noise.  This Gaussian noise assumption can be relaxed; the authors don't explain how, but looking at the proofs, I don't doubt that claim.

We are interested in writing each $r_i$ as a linear combination of $v_1, .., v_K$. This would ofcourse be trivial if we have access to $r_1, ..., r_n$, because we can find the extremal vertices $v_1, ..., v_K$, and then the problem amounts to solving a linear system of equation. Unfortunately, we only have access to the noisy $X_1, ..., X_n$, so finding the extremal vertices is not possible. Intuitively, we want to solve a linear system "approximately".

The existing algorithm for this problem is called SPA, or the successive projection algorithm. The algorithm starts with an empty extremal set K. at iteration $k$, given the current residual space, the algorithm projects all the points to the compliment of the previous residual space, then adds the vertex that maximizes the Euclidean norm greedily to the extremal set.

The authors claim that the current SPA algorithm has an issue of being biased outward bound. To counter this, they do a few contributions (in the paper):

1) They introduce a new practical variant of the classical SPA algorithm, pp-SPA. This algorithm has two steps. First, it uses the fact that the points $r_1, ..., r_n$ lie on the same $K-1$ dimensional hyperplane $H$. So they project the noisy readings $X_1, ..., X_n$ onto the plane that minimizes the least square error with respect to the noisy realizations. Next, it "averages" each point using KNN to create more robustness and crease "pseudo-points". This is where the pp term in the algorithm name comes from. The algorithm seems to do better in practice.

2) The authors tighten the non-asymptotic bound for the classical SPA result from depending on $O(1/s_k^2)$ to $O(1/s_{k-1}^2)$ where $s_k$ is the $k$-th largest Eigen value of the extremal vertices $v_1, ..., v_K$. They also a similar bound for pp-SPA.

**Strengths:**

I like the pp-SPA algorithm a lot, and it feels like a very natural idea. It's an added bonus that the authors were able to get tighter theoretical bounds, which while admittedly are too complicated sometimes to compare fairly to classical SPA, are clearly tighter. It's not clear to me at all when the bounds for pp-SPA beat the bounds for classical SPA, simply because of how complicated the bounds are.

I was able to follow most proofs as a non-expert, so the **proofs** are well written (the actual writing is a whole other story, see below). The analysis for the SPA algorithm is a bit gross; I spent several days following the proof, it felt like a nightmare. I don't claim I understand all steps there, but otherwise the proof looked kosher to me. Perhaps spend some time on simplifying it, but that's easier said than done.

Overall, I think that together with the new practical and nice pp-SPA algorithm that clearly does better in practice from the experiments, and the new non trivial theory to back it up, the paper is above acceptance threshold.

**Weaknesses:**

1) For some reason, the authors do not compare pp-SPA with any robust SPA. I found many results in the literature on robust SPA, so it seems bizare it is not included in the experimental section here. Please try to include at least one robust variant of SPA/similar algorithm in the experimental section.

2) The writing can be dramatically improved. See below minor points, but if this is accepted, please spent a few iterations on improving the writing. It's extremely terse at time.

3) You reprove several "elementary" results that I am almost certain must exist in the literature. Please don't do this, and spend some time to search for the result you need and cite appropriately. You don't need to prove everything from first principles! For example, Lemma 1 can be given as a HW question for an honors linear algebra class....

4) The statements of Theorem 2/3 are an absolute nightmare, and I struggle to see how anyone would cite/use your result without a ton of heavy lifting in understanding your proofs.

----------------------------------------------------------------
Minor comments:

Page 2 "and so on and so forth" Akward, please change.

"However, since the simplex lies on a
hyperplane of (K − 1)-dimension, it is inefficient if we directly apply SPA to X1, X2, . . . , Xn"  This needs more explanation

"The results suggest that the SPA may be significantly
biased outward bound, and there is a large room for improvement" No! This result suggests that SPA is biased outward bound for this example! Why would this suggest it is a general phenomenon?! Actually, how do you know your own algorithm doesn't display this phenomenon for other examples?


"he triangles in black, green, and cyan are the true simplex (which is a triangle since (K −1) = 2), the simplex estimated by SPA, and the simplex estimated by D-SPA (D- SPA is a special case of pp-SPA where the projection step is skipped and pp-SPA is a new approach to be introduced; see below), respectively." Rewrite this sentence!


Page 2 "Roughly say" --> Informally, ...


"successive projection algorithm (SPA) (Ara´ujo et al., 2001)" The "term" successive projection algorithm might've originated from the Araujo paper from 2001, but I can't believe that this algorithm was first discovered in 2001. To start off, it has similarities to the Frank-Wolfe minimum norm point algorithm, and several polyhedral algorithms. Can you please verify where the original **idea** was derived, to keep historical accuracy?

Lemma 1 (Best-fit hyperplane) There is no way this result isn't known. I can literally prove this result without even looking at your Appendix with Lagrangian multipliers and straightforward linear algebra identities. Please search for where this result was first proved, and don't reprove old things unless your proof is starkly different (which is absolutely not the case here)

**Questions:**

My questions are listed in the Weakness section if any.

---

> ### Author Response · Authors · 2023-11-22
> **Response to Reviewer KsqF**
>
> Thank you for your positive assessment of our paper and constructive comments. We are very encouraged that you "like the pp-SPA algorithm a lot" and you think it is "an added bonus ... to get tighter theoretical bounds".
>
> Below is a point-to-point response to your comments. We also encourage you to read our Response-to-All-Reviewers.
>
> ### Main comments:
>
> **1. Robust SPA**: *"Please try to include at least one robust variant of SPA/similar algorithm in the experimental section."*
>
> This is a great point. In the revision we have added the comparison with robust-SPA (Gillis, 2019) in all simulations. We used the Matlab code downloaded from $\texttt{bit.ly/robustSPA}$.
> The results are included in the updated Figure 3. We observe that pp-SPA outperforms robust-SPA, especially when $d$ is large.
>
> We also added references to recent variants of SPA, including robust SPA, pre-conditioned SPA, rank-K approximate SPA, etc.. The hyperplane projection and KNN denoise idea have never been used in these algorithms. Furthermore, none of existing works has got as tight bounds as in our work.
>
> **2. Writing**: *"If this is accepted, please spent a few iterations on improving the writing. It's extremely terse at time."*
>
> In this revision, we have already re-written some sections (a summary of changes is in our response-to-all-reviewers). We tried to use figures/examples/words to explain the math and notations. Below are a few examples.
> - New figure 2 and text therein: They provide an intuitive explanation of $\beta_{\text{new}}(X,V)$ in the non-asymptotic bound.
> - The numerical example below Theorem~1: We explicitly calculate the bound in Gillis and Vavasis (2013) and our new bound in a numerical example, to show why the improvement can be substantial in some cases.
> - Table 1: This table is a summary of rate comparison and has been moved to the beginning of Section 4.
>
> If the paper is accepted, we are willing to spend more time improving the writing.
>
> **3. Whether to include the proof of Lemma 1**: *"You don't need to prove everything from first principles! For example, Lemma 1 can be given as a HW question for an honors linear algebra class."*
>
> We agree that Lemma 1 is a standard result regarding subspace approximation via SVD, which appears in tutorial materials. However, we hope to prove this lemma in the supplementary materials for completion, and we think doing so will increase the readability of our paper. We did not intend to have any credit for such results.
>
> In this revision, we added a sentence in the first line of proof saying that "this is a quite standard result, which can be found at tutorial materials (e.g., https://people.math.wisc.edu/~roch/mmids/roch-mmids-llssvd-6svd.pdf). We include a proof here only for convenience of readers."
>
> **4. Statements of Theorems 2-3**: *"I struggle to see how anyone would cite/use your result without a ton of heavy lifting in understanding your proofs."*
>
> We recognized where the confusion came from. In order to make the results as general as possible, we use $\alpha_n$, $b_n$, and $t_n$ in the theorem statements. To get the simple expressions of rates, one still needs to plug in the orders of $(\alpha_n, b_n, t_n)$ from the theorem body. We agree that this is not reader-friendly.
>
> In the revision, we have explicitly explained how to plug in the order of $(\alpha_n, b_n, t_n)$ to get simplified bounds. For example, in the newly added paragraph below Theorem 2, we showed carefully how to use Theorem 2 to get a bound like
> $$
> \max_k \|\hat{v}\_k -v_k\|\leq \sigma g_{\text{new}}(V)\cdot  \begin{cases}
> \sqrt{\log\log(n)} & \text{ if $d/n$ is properly small};\cr
> \sqrt{[2+o(1)] \log (n)} & \text{ if $d/n$ is properly large}.
> \end{cases}
> $$
> In addition, we have included a table in the beginning of Section 4 to summarize all the bounds.
>
> ### Minor comments:
>
> - Editing of text: We have made all the changes as you suggested.
> - Lemma 1: Please see our response to your Main Point 3.
> - An explanation of "it is inefficient if we directly apply SPA":  We added more explanations on page 2 as follows: "However, since the true simplex vertices $v_1,\ldots,v_K$ lie on a hyperplane of $(K-1)$-dimension, if we directly apply SPA to $X_1, X_2, \ldots, X_n$, the resultant hyperplane formed by the estimated simplex vertices is very likely to deviate from the true hyperplane, due to noise corruption. This will cause inefficiency of SPA."
> - If bias outward of SPA is a general phenomenon: This is indeed a general phenomenon (e.g., see Gillis (2019)), and it is caused by the way SPA is designed.  At each iteration, the estimated vertex is (by definition) the point in the projected data cloud that is *furthest away* from the origin. Therefore,  the output vertices are likely those points that lie on the furthest positions away from the center of the data cloud. This is why we always have the biased outward bound phenomenon. It is now seen more clearly in the updated Figure 1.

---

> > ### Author Response · Authors · 2023-11-23
> > **Response to Reviewer KsqF (an additional minor point)**
> >
> > We realize that we did not respond to your minor point about Frank-Wolfe minimum norm point (MNP) algorithm, and we add it here.
> >
> > The Frank-Wolfe (Rank and Wolfe, 1956) and MNP (Wolfe, 1976) seemed to be two algorithms. The MNP algorithm is more relevant, which aims to find the smallest Euclidean norm in a polytope. The major difference is however, in Wolf's setting, the polytope is known but in our setting, the simplex is unknown. In a later round, we are happy to add some references of Wolfe's works and connect it to the history of SPA.

---

### Author Response · Authors · 2023-11-22
**Response to All Reviews**

We thank all reviews for their helpful comments. We are especially glad that the reviewers appreciate our contributions in both algorithm and theory. In this response, we hope to clarify some misunderstandings and also outline the major changes in the revision.

## 1. Rate comparison between SPA and pp-SPA.

Reviewers mentioned that it is not easy to find the explicit rate comparison. This was originally shown in a table after Theorems 2-3. In this revision, we have moved this table to the beginning of Section 4, so that the readers see the comparison before reading the detailed statements. This table compares bounds of SPA, P-SPA, D-SPA, and pp-SPA. For your convenience, we include here the two rows corresponding to SPA and pp-SPA (please see the caption of Table~1 about the notations).
$$
\begin{array}{l  cccc}
& d\ll \log(n) & d= a_0\log (n) & \log(n)\ll d\ll n^{1-\frac{2(1-c_0)}{K-1}} & d\gg n^{1-\frac{2(1-c_0)}{K-1}}\cr
\text{SPA} & \sqrt{2\log(n)} & \sqrt{a_1\log (n)} & \sqrt{d} & \sqrt{d}\cr
\text{pp-SPA} &  \sqrt{2c_0\log(n)} &\sqrt{2c_0\log(n)} &  \sqrt{2c_0\log(n)} & \sqrt{2\log(n)}
\end{array}
$$
In summary: (a) pp-SPA  always has a better error bound than SPA. (b) When $d\gg\log(n)$, the improvement is a factor of $o(1)$. (c) When $d=O(\log(n))$, the improvement is a constant factor that is strictly smaller than $1$.

## 2. Significance of our results compared to Gillis and Vavasis (2013).

We clarify that the improvement of $s_{K}(V)$ to $s_{K-1}(V)$ cannot be achieved by an extension of [GV13]'s proof (we thank one reviewer for helping point out that "the proof is completely different"). The proof in [GV13] is driven by **matrix norm inequalities** and does not use any geometry.  However, quantities such as $s_{K-1}(V)$ and $\beta_{\text{new}}(X,V)$ are insufficient to provide strong matrix norm inequalities, making it very difficult (if not impossible) to extend [GV13]'s proofs to get our theorem. We used a very different proof idea based on **geometric insights**. We construct a *simplicial neighborhood* near each true vertex and show that the $\hat{v}_k$ in each step of SPA must fall into one of these simplicial neighborhoods.

We also want to point out that the improvement of the bound is substantial in some cases. Note that $s_{K-1}(V)$ is driven by the volume of the simplex, but $s_K(V)$ is driven by the location (center) of the simplex. In Section D of the supplementary material, we added a toy example, where we fix a simplex and gradually move it close to the origin. Since the shape and volume of this simplex do not change, our bound always remains the same. However, as the simplex is moved closer to the origin, the bound in [GV13] gets larger and larger.

In addition, our paper proposes a new algorithm, pp-SPA, and contains a whole section of theoretical analysis of pp-SPA, where we have used random matrix theory and extreme value theory. Even if we ignore the non-asymptotic bound, these contributions alone are significant.

## 3. New simulation results.

-  *Higher dimension*:
Reviewers suggested to add simulations for larger $d$. We have followed this suggestion and included experiments for $d=1,2,3, \ldots, 48,49, 50$.

- *Comparison with robust SPA*: One reviewer suggested to compare robust SPA (Gillis, 2019), which is a great idea. We have added robust SPA to all simulations.
We observe that pp-SPA outperforms robust-SPA. Especially, the error of robust SPA increases with $d$, but the error of pp-SPA stays rather flat as $d$ increases.

These results are in the updated Figure~3. In addition, we added references to robust SPA and other variants of SPA in the introduction.

## 4. Writing.

We thank all reviewers for the great suggestions on writing. In this revision, we significantly re-wrote some sections and added illustrating examples and figures. Below is a summary of the changes:

- Figure 1 in the introduction: Following the reviewers' suggestions, we combined two plots into a single plot, so that it is easier to compare the simplex estimated by SPA and the one estimated by pp-SPA.
- Splitting Section 3 into two sections: We separate out the part about non-asymptotic bound for SPA and make it the new Section 3. The analysis of pp-SPA is put in the new Section 4.
- In the new Section 3, we added (a) a new paragraph and a new figure to explain the meaning of $\beta_{\mathrm{new}}(X,V)$ and (b) a toy example where we give explicit values of the old and new bounds.
- In the new Section 4, we list the rate comparison in a table before stating the theorems.  We also added text below Theorem 2 and Theorem 3 to explain how to interpret the results.

We hope that these changes help make the paper reader-friendlier.

---

### Meta-Review · Area_Chair_Cchv · 2023-12-07

**Metareview:**

This paper considers the vertex hunting problem, which finds many important applications. There are three main contributions: (i) A new algorithm (pp-SPA) which improves the performance of a widely used existing method SPA; (ii) An improved error bound for the SPA method; (iii) A rate for pp-SPA, which is much faster than the rate of SPA. Numerical results confirmed these contribution. Overall, this paper provide some important new results for a fundamental problem.

**Justification For Why Not Higher Score:**

Results not significant enough.

**Justification For Why Not Lower Score:**

Provides solid results for an important problem. Should be accepted.

---

### Decision · Program_Chairs · 2024-01-16

Accept (poster)